# On Regret Bounds of Thompson Sampling for Bayesian Optimization

**Shion Takeno** [1]   **Shogo Iwazaki** [2]

## Abstract

We study a widely used Bayesian optimization method, Gaussian process Thompson sampling (GP-TS), under the assumption that the objective function is a sample path from a GP. Compared with the GP upper confidence bound (GP-UCB) with established high-probability and expected regret bounds, most analyses of GP-TS have been limited to expected regret. Moreover, whether the recent analyses of GP-UCB for the lenient regret and the improved cumulative regret upper bound can be applied to GP-TS remains unclear. To fill these gaps, this paper shows several regret bounds: (i) a regret lower bound for GP-TS, which implies that GP-TS suffers from a polynomial dependence on $1/\delta$ with probability $\delta$, (ii) an upper bound of the second moment of cumulative regret, which directly suggests an improved regret upper bound on $\delta$, (iii) expected lenient regret upper bounds, and (iv) an improved cumulative regret upper bound on the time horizon $T$. Along the way, we provide several useful lemmas, including a relaxation of the necessary condition from recent analysis to obtain improved regret upper bounds on $T$.

## 1. Introduction

Bayesian optimization (BO) (Kushner, 1964; Mockus et al., 1978) is a powerful framework for black-box optimization problems. BO aims to optimize an expensive-to-evaluate black-box function using a small number of input-output pairs by adaptively querying input points based on a Bayesian model, typically a Gaussian process (GP) model. BO has been applied to a wide range of applications, such as materials informatics (Ueno et al., 2016), AutoML (Snoek et al., 2012), and drug discovery (Korovina et al., 2020). Alongside these applications and algorithmic developments,

theoretical properties of BO have also been studied. This paper focuses on regret analysis under the assumption that the objective function is a sample path from a GP.

Gaussian process upper confidence bound (GP-UCB) (Kushner, 1964; Srinivas et al., 2010) is a BO algorithm with well-established theoretical guarantees. The theoretical performance of BO methods is often measured by cumulative regret (Srinivas et al., 2010), for which both high-probability and expected upper bounds of GP-UCB have been derived (Srinivas et al., 2010; Takeno et al., 2023). High-probability upper bound of an alternative criterion called *lenient regret* (Merlis & Mannor, 2021), which counts a regret exceeding a given tolerance, has also been obtained by Cai et al. (2021); Iwazaki (2025b). Furthermore, Iwazaki (2025b) has shown a tighter high-probability upper bound for the cumulative regret. These results position GP-UCB as the most extensively analyzed algorithm in the BO literature.

GP Thompson sampling (GP-TS) (Russo & Van Roy, 2014) is another well-known BO algorithm with regret guarantees. GP-TS achieves a similar expected cumulative regret upper bound as GP-UCB (Kandasamy et al., 2018; Russo & Van Roy, 2014; Takeno et al., 2024). However, its high-probability regret upper bound is limited to the direct consequence of the expected regret bound, leading to worse dependence on probability $\delta$ than GP-UCB. In addition, neither high-probability lenient regret bounds nor tighter high-probability cumulative regret bounds have been established for GP-TS, which has been explicitly described as an open problem in (Iwazaki, 2025b). Despite these limitations, GP-TS remains a promising alternative among BO methods with regret guarantees, since GP-UCB often relies on a carefully tuned confidence width parameter that lacks such guarantees in practice. Therefore, more refined regret analyses of GP-TS are needed to support its empirical effectiveness.

This paper aims to bridge the gap between the regret analyses of GP-UCB and GP-TS discussed above. First, we investigate the dependence on probability $\delta$ in high-probability regret bounds. While GP-UCB enjoys logarithmic dependence on $\delta$, existing analyses of GP-TS require polynomial dependence of $1/\delta$, raising the question of whether logarithmic dependence is achievable by GP-TS. For this question, we construct a problem instance in which GP-TS incurs

---

[1]Department of Engineering, Nagoya University, Aichi, Japan [2]MI-6 Ltd., Tokyo, Japan. Correspondence to: Shion Takeno <takeno.s.mllab.nit@gmail.com>.

*Proceedings of the 43$^{rd}$ International Conference on Machine Learning*, Seoul, South Korea. PMLR 306, 2026. Copyright 2026 by the author(s).

$\Omega(1/\delta^c)$ regret, which implies a purely theoretical limitation that GP-TS cannot obtain $O(\log(1/\delta))$ regret upper bounds in general. Second, we derive an upper bound on the second moment of the cumulative regret, which directly leads to an improved cumulative regret upper bound on $\delta$. Third, we show that GP-TS achieves polylogarithmic expected upper bounds on lenient regrets, the first such bounds for any algorithm. Our proof for the lenient regret is different from (Cai et al., 2021; Iwazaki, 2025b) and readily suggests polylogarithmic expected lenient regret upper bounds for GP-UCB by combining the known proof techniques (Russo & Van Roy, 2014; Takeno et al., 2023). Finally, by adapting the recent analysis of GP-UCB by Iwazaki (2025b) and our lenient regret upper bounds, we obtain an improved cumulative regret bound on the time horizon $T$ for GP-TS. Our refined analysis, applicable to both GP-UCB and GP-TS, relaxes the conditions on Matérn kernels.

Our contributions are summarized as follows:

1. Theorem 3.1 provides the problem instance in which GP-TS incurs $\Omega(1/\delta^c)$ regret with probability $\delta$. This result implies a purely theoretical limitation of GP-TS that the regret upper bounds of GP-TS cannot be bounded from above by $O\big(\log(1/\delta)\big)$.

2. In Theorem 3.2, we derive the upper bound on the second moment of the cumulative regret and the high-probability cumulative regret bound that improves dependence on probability $\delta$ by a factor of $1/\sqrt{\delta}$.

3. In Theorem 3.3, we obtain the expected lenient regret upper bound for GP-TS, which is polylogarithmic on the time horizon $T$ as with the high-probability bound of GP-UCB.

4. Theorem 3.5 presents the $\tilde{O}(\sqrt{T})$ high-probability cumulative regret upper bound, where $\tilde{O}$ hides polylogarithmic factors on $T$, improved with resect to $T$ for squared exponential and Matérn kernels under a looser condition on Matérn kernels by the refined analysis shown in Lemma 3.4.

Along the way, we obtain useful lemmas, which are discussed alongside the most relevant theorems.

## 1.1. Related Work

Common theoretical assumptions for BO are categorized into the frequentist setting (Chowdhury & Gopalan, 2017; Iwazaki, 2025a; Iwazaki & Takeno, 2025; Janz et al., 2020; Srinivas et al., 2010) and the Bayesian setting (Iwazaki, 2025b; Scarlett, 2018; Srinivas et al., 2010; Takeno et al., 2023; 2024). In the frequentist setting, the objective function is assumed to be an element of a reproducing kernel Hilbert space endowed by a predefined kernel function. In the Bayesian setting, the objective function is assumed to be a sample path from a GP with a predefined kernel function. Note that neither setting encompasses the other as discussed in Section 4 of (Srinivas et al., 2010). As already discussed, this paper focuses on the Bayesian setting. In addition, we focus on the noisy setting, where observations are contaminated by noise, since a noise-free setting has been considered separately using different proof techniques (De Freitas et al., 2012; Iwazaki, 2025a).

GP-TS has also been considered in the frequentist setting (Chowdhury & Gopalan, 2017; Vakili et al., 2021b). Although we will not consider the frequentist setting, the proof technique in (Chowdhury & Gopalan, 2017) could potentially be leveraged in the Bayesian setting, as discussed in Section 4. Furthermore, GP-TS has also been extended to various problem settings, such as parallel BO (Hernández-Lobato et al., 2017; Kandasamy et al., 2018; Nava et al., 2022), constrained BO (Eriksson & Poloczek, 2021), multi-objective BO (Bradford et al., 2018; Paria et al., 2020; Renganathan & Carlson, 2025), preferential BO (Sui et al., 2017), and reinforcement learning (Bayrooti et al., 2025a;b). Thus, fundamental refinement of analysis for GP-TS is needed. Note that, for (Bayrooti et al., 2025a;b), we will discuss their result that appears to contradict our Theorem 3.1 in Section 3.1.

Since the posterior sampling of GPs cannot be conducted analytically, we require an approximation, for example, proposed by Wilson et al. (2020; 2021) based on random Fourier feature (Rahimi & Recht, 2008). In our analysis, we ignore this approximation error, though it may be addressed by high-accuracy approximations as with (Mutny & Krause, 2018).

In the BO literature, many other algorithms have been studied. For example, probability of improvement (Kushner, 1964), expected improvement (Mockus et al., 1978), knowledge gradient (Frazier et al., 2009), entropy search (Hennig & Schuler, 2012; Villemonteix et al., 2009), predictive entropy search (Hernández-Lobato et al., 2014) have been extensively studied. However, regret guarantees for these algorithms remain an open problem.

Except for GP-UCB and GP-TS, several BO algorithms with regret guarantees have been proposed recently (Takeno et al., 2023; 2024; 2025a;b). These existing methods are randomized algorithms, as with GP-TS, and their regret analysis follows a similar approach to GP-TS. Therefore, our proof technique may be useful for analyzing those algorithms.

Scarlett (2018) has shown the algorithm-independent regret lower bound in the Bayesian setting. Specifically, Scarlett (2018) has shown the $\Omega(\sqrt{T})$ regret lower bound for conditional expected regret with one-dimensional objective functions. Therefore, rigorous high-probability regret lower

bounds have not been established. Thus, as discussed in Section 4, proving the optimality of BO algorithms rigorously remains an open problem.

TS (Thompson, 1933) has been extensively studied in the bandit literature, and many regret bounds of TS have been provided, as reviewed in (Russo et al., 2018). In particular, Agrawal & Goyal (2017) have shown the expected regret lower bound for (independent) Gaussian multi-armed bandit. However, to our knowledge, regret lower bounds focusing on probability, particularly applicable to GP-TS, have not been established.

We analyze lenient regret, recently introduced by Merlis & Mannor (2021) in the bandit literature. Cai et al. (2021) have derived the high-probability lenient regret upper bound of GP-UCB in the frequentist setting. Iwazaki (2025b) has extended this analysis to the Bayesian setting. We show the Bayesian expected lenient regret upper bound of GP-TS, in which we employ a different proof from (Cai et al., 2021; Iwazaki, 2025b).

# 2. Preliminaries

This section provides problem setup, regularity assumptions, and background knowledge for GP-TS.

## 2.1. Problem Setup

We consider a sequential decision-making problem to optimize an unknown black-box function $f$:

$$\boldsymbol{x}^* = \arg\max_{\boldsymbol{x} \in \mathcal{X}} f(\boldsymbol{x}),$$

where $\mathcal{X} \subset \mathbb{R}^d$ is an input domain and $d$ is an input dimension. Here, "black-box" means that we can obtain only (contaminated) function values, and other information, such as derivatives, is unknown. Furthermore, we generally assume that $f$ is expensive to evaluate. Therefore, our goal is to optimize $f$ with fewer function evaluations.

In this problem, BO sequentially evaluates the maximizer of an acquisition function (AF), which is designed so that beneficial inputs are chosen based on a Bayesian model. That is, for all iterations $t \geq 1$, BO algorithms sequentially evaluate $f(\boldsymbol{x}_t)$, where $\boldsymbol{x}_t = \arg\max_{\boldsymbol{x} \in \mathcal{X}} \alpha_t(\boldsymbol{x})$ with some AF $\alpha_t : \mathcal{X} \to \mathbb{R}$ based on the Bayesian model. By this procedure, BO aims to achieve a sample-efficient optimization.

In the BO literature, the performance of algorithms is often measured by cumulative regret (Srinivas et al., 2010):

$$R_T = \sum_{t=1}^{T} f(\boldsymbol{x}^*) - f(\boldsymbol{x}_t).$$

In particular, if the cumulative regret is sublinear $R_T = o(T)$, then we can see that the solution of the BO algorithm converges to the optimal solution. This is because we can choose the solution $\hat{\boldsymbol{x}}_t$ so that $f(\boldsymbol{x}^*) - f(\hat{\boldsymbol{x}}_t) \leq \frac{1}{T} R_T = o(1)$. Hence, we focus on sublinear cumulative regret bounds.

In addition, we consider the performance measure called lenient regret (Cai et al., 2021; Merlis & Mannor, 2021):

$$LR_T = \sum_{t \in \mathcal{T}} f(\boldsymbol{x}^*) - f(\boldsymbol{x}_t),$$

where $\mathcal{T} = \{t \in [T] \mid f(\boldsymbol{x}^*) - f(\boldsymbol{x}_t) \geq \Delta\}$ and $\Delta > 0$ is a predefined constant. The lenient regret measures regret under a given tolerance $\Delta$, where the input $\boldsymbol{x}$ that satisfies $f(\boldsymbol{x}^*) - f(\boldsymbol{x}) \leq \Delta$ is regarded as a sufficiently good action. We further analyze upper bounds of $|\mathcal{T}|$ as with (Cai et al., 2021; Merlis & Mannor, 2021).

## 2.2. Gaussian Process Regression

In this paper, we assume the following regularity condition on the objective function $f$:

**Assumption 2.1.** The objective function $f : \mathcal{X} \to \mathbb{R}$ follows $\mathcal{GP}(0, k)$, where $\mathcal{X} \subset \mathbb{R}^d$, $k : \mathcal{X} \times \mathcal{X} \to \mathbb{R}$ is a kernel function, and $\mathcal{GP}(0, k)$ denotes a zero-mean GP with a covariance function $k$. Furthermore, the kernel function satisfies $k(\boldsymbol{x}, \boldsymbol{x}') \leq 1$ for all $\boldsymbol{x}, \boldsymbol{x}' \in \mathcal{X}$. Observation noise $\{\epsilon_i\}_{i \in \mathbb{N}}$ are mutually independent and $\epsilon_i \sim \mathcal{N}(0, \sigma^2)$ with $\sigma^2 > 0$ for all $i \in \mathbb{N}$. The noise contaminates observations as $y_i = f(\boldsymbol{x}_i) + \epsilon_i$.

Let the training dataset $\mathcal{D}_t = \{(\boldsymbol{x}_i, y_i)\}_{i=1}^{t}$. Under Assumption 2.1, the posterior distribution of $f$ given $\mathcal{D}_t$ is also a GP, whose mean and variance are derived as follows:

$$\mu_t(\boldsymbol{x}) = \boldsymbol{k}_t(\boldsymbol{x})^\top \left(\boldsymbol{K}_t + \sigma^2 \boldsymbol{I}_t\right)^{-1} \boldsymbol{y}_t,$$
$$\sigma_t^2(\boldsymbol{x}) = k(\boldsymbol{x}, \boldsymbol{x}) - \boldsymbol{k}_t(\boldsymbol{x})^\top \left(\boldsymbol{K}_t + \sigma^2 \boldsymbol{I}_t\right)^{-1} \boldsymbol{k}_t(\boldsymbol{x}),$$

where $\boldsymbol{k}_t(\boldsymbol{x}) := \left(k(\boldsymbol{x}, \boldsymbol{x}_1), \dots, k(\boldsymbol{x}, \boldsymbol{x}_t)\right)^\top \in \mathbb{R}^t$, $\boldsymbol{K}_t \in \mathbb{R}^{t \times t}$ is the kernel matrix whose $(i, j)$-element is $k(\boldsymbol{x}_i, \boldsymbol{x}_j)$, $\boldsymbol{I}_t \in \mathbb{R}^{t \times t}$ is the identity matrix, and $\boldsymbol{y}_t := (y_1, \dots, y_t)^\top \in \mathbb{R}^t$ (Rasmussen & Williams, 2005).

For continuous input domains, we consider the following additional assumption (Kandasamy et al., 2018; Srinivas et al., 2010; Takeno et al., 2024; 2025c):

**Assumption 2.2.** Let $\mathcal{X} \subset [0, r]^d$ be a compact set with some $r > 0$. There exist constants $a, b > 0$ such that the kernel $k$ satisfies the following condition on the derivatives of a sample path $f$:

$$\Pr\left(\sup_{\boldsymbol{x} \in \mathcal{X}} \left|\frac{\partial f}{\partial \boldsymbol{x}_j}\right| > L\right) \leq a \exp\left(-\left(\frac{L}{b}\right)^2\right), \text{ for } j \in [d],$$

where $[d] = \{1, \dots, d\}$.

This condition is satisfied by the common kernel functions (Srinivas et al., 2010), such as linear kernels $k_{\mathrm{Lin}}(\boldsymbol{x}, \boldsymbol{x}') = \boldsymbol{x}^\top \boldsymbol{x}'$, squared exponential (SE) kernels $k_{\mathrm{SE}}(\boldsymbol{x}, \boldsymbol{x}') = \exp\left(-\|\boldsymbol{x} - \boldsymbol{x}'\|_2^2 / (2\ell^2)\right)$, and Matérn kernels $k_{\mathrm{Mat}}(\boldsymbol{x}, \boldsymbol{x}') = \frac{2^{1-\nu}}{\Gamma(\nu)} \left(\frac{\sqrt{2\nu}\|\boldsymbol{x}-\boldsymbol{x}'\|_2}{\ell}\right)^\nu J_\nu\left(\frac{\sqrt{2\nu}\|\boldsymbol{x}-\boldsymbol{x}'\|_2}{\ell}\right)$ with $\nu > 2$, where $\ell, \nu > 0$ are the lengthscale and smoothness parameter, respectively, and $\Gamma(\cdot)$ and $J_\nu$ are Gamma and modified Bessel functions.

In addition, we leverage the following result (Theorem E.4 of Kusakawa et al., 2022) for continuous input domains:

**Lemma 2.3** (Lipschitz constants for posterior standard deviation). *Let $k : \mathbb{R}^d \times \mathbb{R}^d \to \mathbb{R}$ be linear, SE, or Matérn kernel with $\nu > 1$ and $k(\boldsymbol{x}, \boldsymbol{x}) \leq 1$ for all $\boldsymbol{x} \in \mathbb{R}^d$. Moreover, assume that a noise variance $\sigma^2$ is positive. Then, for any $t \geq 1$ and $\mathcal{D}_{t-1}$, the posterior standard deviation $\sigma_{t-1}(\boldsymbol{x})$ satisfies that*

$$\forall \boldsymbol{x}, \boldsymbol{x}' \in \mathbb{R}^d, \ |\sigma_{t-1}(\boldsymbol{x}) - \sigma_{t-1}(\boldsymbol{x}')| \leq L_\sigma \|\boldsymbol{x} - \boldsymbol{x}'\|_1,$$

*where $L_\sigma$ is a positive constant given by*

$$L_\sigma = \begin{cases} 1 & \text{if } k = k_{\mathrm{Lin}}, \\ \frac{\sqrt{2}}{\ell} & \text{if } k = k_{\mathrm{SE}}, \\ \frac{\sqrt{2}}{\ell}\sqrt{\frac{\nu}{\nu-1}} & \text{if } k = k_{\mathrm{Mat}} \text{ with } \nu > 1. \end{cases}$$

We use this lemma to handle discretization error, as in the existing analysis of GP-TS (Takeno et al., 2024).

Finally, to obtain a tighter regret upper bound on $T$, we further leverage the following lemma (De Freitas et al., 2012; Iwazaki, 2025b; Scarlett, 2018):

**Lemma 2.4** (Conditions on the global maximizer of sample path). *Let $\mathcal{X} = [0, r]^d$. Suppose $k = k_{\mathrm{SE}}$ or $k = k_{\mathrm{Mat}}$ with $\nu > 2$ and Assumption 2.1 holds. Then, for any $\delta_{\mathrm{GP}} \in (0, 1)$, there exist strictly positive constants $c_{\mathrm{gap}}, c_{\mathrm{sup}}, c_{\mathrm{quad}}, \rho_{\mathrm{quad}} > 0$ such that the following statements hold simultaneously with probability at least $1 - \delta_{\mathrm{GP}}$:*

1. *The function $f$ has a unique maximizer $\boldsymbol{x}^* \in \mathcal{X}$ such that $f(\boldsymbol{x}^*) > f(\tilde{\boldsymbol{x}}^*) + c_{\mathrm{gap}}$ holds for any local maximizer $\tilde{\boldsymbol{x}}^* \in \mathcal{X}$ of $f$.*

2. *The sup-norm of the sample path is bounded as $\|f\|_\infty \leq c_{\mathrm{sup}}$.*

3. *The function $f$ satisfies $\forall \boldsymbol{x} \in \mathcal{B}_2(\rho_{\mathrm{quad}}; \boldsymbol{x}^*), f(\boldsymbol{x}^*) - c_{\mathrm{quad}}\|\boldsymbol{x}^* - \boldsymbol{x}\|_2^2 \geq f(\boldsymbol{x})$, where $\mathcal{B}_2(\rho; \boldsymbol{x}^*) = \{\boldsymbol{x} \in \mathcal{X} \mid \|\boldsymbol{x}^* - \boldsymbol{x}\|_2 \leq \rho\}$ is the L2-ball on $\mathcal{X}$, whose radius and center are $\rho$ and $\boldsymbol{x}^*$, respectively.*

We use this lemma to derive a similar proof as that of (Iwazaki, 2025b). For more details, see (De Freitas et al., 2012; Iwazaki, 2025b; Scarlett, 2018).

---

**Algorithm 1** Thompson Sampling

**Require:** Domain $\mathcal{X}$, kernel function $k$
1: $\mathcal{D}_0 \leftarrow \emptyset$
2: **for** $t = 1, \ldots, T$ **do**
3:      Update GP posterior $p(f \mid \mathcal{D}_{t-1})$
4:      Generate sample path $g_t \sim p(f \mid \mathcal{D}_{t-1})$
5:      $\boldsymbol{x}_t \leftarrow \arg\max_{\boldsymbol{x} \in \mathcal{X}} g_t(\boldsymbol{x})$
6:      Observe $y_t$ and $\mathcal{D}_t \leftarrow \mathcal{D}_{t-1} \cup (\boldsymbol{x}_t, y_t)$
7: **end for**
8: **return** Some recommended input $\hat{\boldsymbol{x}}_T$

---

### 2.3. Maximum Information Gain (MIG)

The complexity of BO problems is commonly quantified by the MIG (Srinivas et al., 2010) defined below:

**Definition 2.5** (Maximum information gain). Let $f \sim \mathcal{GP}(0, k)$ over $\mathcal{X} \subset [0, r]^d$. Let $A = \{\boldsymbol{a}_i\}_{i=1}^T$, where $\boldsymbol{a}_i \in \mathcal{X}$ for all $i \in [T]$. Let $\boldsymbol{f}_A = \left(f(\boldsymbol{a}_i)\right)_{i=1}^T$, $\boldsymbol{\epsilon}_A = \left(\epsilon_i\right)_{i=1}^T$, where $\forall i, \epsilon_i \sim \mathcal{N}(0, \sigma^2)$, and $\boldsymbol{y}_A = \boldsymbol{f}_A + \boldsymbol{\epsilon}_A \in \mathbb{R}^T$. Then, MIG $\gamma_T$ is defined as follows:

$$\gamma_T := \max_A I(\boldsymbol{y}_A; \boldsymbol{f}_A),$$

where $I$ is the Shannon mutual information.

The MIGs of several commonly used kernel functions are known to be sublinear in terms of $T$. For example, $\gamma_T = O(d \log T)$ for linear kernels, $\gamma_T = O\left((\log T)^{d+1}\right)$ for SE kernels, and $\gamma_T = O\left(T^{\frac{d}{2\nu+d}}(\log T)^{\frac{4\nu+d}{2\nu+d}}\right)$ for Matérn-$\nu$ kernels, respectively (Iwazaki, 2025b; 2026; Srinivas et al., 2010; Vakili et al., 2021a)[1]. For convenience, we define the above known upper bound of MIG as $\tilde{\gamma}_t : \mathbb{R}_{\geq 0} \to \mathbb{R}_{\geq 0}$, which is a concave function with respect to $t \geq 1$ (Iwazaki, 2025b; 2026; Srinivas et al., 2010; Vakili et al., 2021a). We will use this concavity particularly to show Lemma E.7.

### 2.4. Gaussian Process Thompson Sampling

GP-TS (Kandasamy et al., 2018; Russo & Van Roy, 2014; Takeno et al., 2024) is the BO method that sequentially evaluates the optimal point of a posterior sample path. Therefore, the AF of GP-TS is formalized as

$$\boldsymbol{x}_t = \arg\max_{\boldsymbol{x} \in \mathcal{X}} g_t(\boldsymbol{x}),$$

where $g_t \sim p(f \mid \mathcal{D}_{t-1})$. Algorithm 1 provides a pseudocode of GP-TS.

The expected regret upper bounds under Assumption 2.1 have been shown as (Kandasamy et al., 2018; Russo &

---

[1] Although Vakili et al. (2021a) claimed the tighter MIG upper bound for Matérn kernels, we use the bound from Iwazaki (2025b) due to the issue pointed out by Iwazaki (2025b); Janz (2021).

Van Roy, 2014; Takeno et al., 2024):

$$\mathbb{E}\left[R_T\right] = \tilde{O}(\sqrt{T\gamma_T}),$$

where $\tilde{O}$ hides polylogarithmic factors in terms of $T$. Therefore, for any $\delta \in (0, 1)$, the following inequality holds with probability at least $1 - \delta$:

$$R_T = \tilde{O}\left(\frac{\sqrt{T\gamma_T}}{\delta}\right),$$

as a direct consequence of Markov's inequality (Russo & Van Roy, 2014).

## 3. Regret Analysis

This section describes our regret analyses.

### 3.1. Regret Lower Bound

First, we show the simple two-armed problem instance where GP-TS must incur $(1/\delta)^c$ cumulative regret, where $c \in (0, 1)$, with probability at least $\delta \in (0, 1)$.

**Theorem 3.1.** *Assume that $\mathcal{X} = \{\boldsymbol{x}^{(1)}, \boldsymbol{x}^{(2)}\}$, $\epsilon_t \sim \mathcal{N}(0, 1)$ for all $t \in [T]$, and*

$$\left(\begin{array}{c} f(\boldsymbol{x}^{(1)}) \\ f(\boldsymbol{x}^{(2)}) \end{array}\right) \sim \mathcal{N}\left(\left(\begin{array}{c} 0 \\ 0 \end{array}\right), \left(\begin{array}{cc} 1 & 1/2 \\ 1/2 & 1 \end{array}\right)\right).$$

*Then, if Algorithm 1 runs and $T > e$, the following holds:*

$$\Pr\left(R_T \geq T/2\right) \geq \frac{c_1}{T^{c_2}},$$

*where $c_1$ and $c_2$ are strictly positive constants.*

This theorem implies a purely theoretical limitation that we cannot obtain $O(\log(1/\delta))$ cumulative regret upper bounds for GP-TS in general. This is because we can show a contradiction from Theorem 3.1 if we set $\delta = c_1/T^{c_2}$. On the other hand, in practice, GP-TS often shows superior performance. Indeed, since the constant $c_2 = 17$ in Theorem 3.1, shown in Appendix A, is highly large, Theorem 3.1 does not necessarily suggest the practical instability of GP-TS. This result motivates us to improve the exponent of $1/\delta$ in the regret upper bound. See the following proof sketch and Appendix A for the proof.

**Proof sketch.** We derive a lower bound on probability

$$\Pr(E_f \wedge E_\epsilon \wedge E_{\mathrm{TS}}) \geq c_1/T^{c_2},$$

where the events are defined as follows:

$$E_f = \left\{f(\boldsymbol{x}^{(1)}) \geq 4\sqrt{\log T} \text{ and } f(\boldsymbol{x}^{(2)}) \geq f(\boldsymbol{x}^{(1)}) + 1\right\},$$

$$E_\epsilon = \left\{\forall t \in [\lceil T/2 \rceil], \frac{1}{t}\sum_{i=1}^{t} \epsilon_i \geq -1\right\},$$

$$E_{\mathrm{TS}} = \left\{\forall t \in [\lceil T/2 \rceil], \boldsymbol{x}_t = \boldsymbol{x}^{(1)}\right\},$$

for which we leverage Lemma 5.1 of (Takeno et al., 2025a) shown in Lemma A.1. Obviously, under these events, GP-TS suffers from $T/2$ regret by querying the suboptimal arm $\boldsymbol{x}^{(1)}$ for $T/2$ times.

**Extendability of Theorem 3.1.** Extending Theorem 3.1 to arbitrary algorithms is difficult. On the other hand, we expect similar results to hold for algorithms without tuning parameters that promote exploration, such as the confidence width parameter in GP-UCB. Indeed, the proof of Theorem 3.1 is inspired by Theorem 4.6 in (Takeno et al., 2025a), which analyzes the regret lower bound of GP-UCB with a fixed (non-increasing) confidence width parameter. Therefore, we conjecture that, without some tuning parameter or modification of the algorithm to promote exploration, the BO algorithms cannot achieve $O(\log(1/\delta))$ cumulative regret upper bound in general, though solving it rigorously still remains open.

**Contradictions from the existing study.** Bayrooti et al. (2025a;b) (for example, Remark 4 of (Bayrooti et al., 2025a)) have reported the $O\left(\sqrt{T\gamma_T}\log(T/\delta)\right)$ regret upper bound that appears to contradict Theorem 3.1. Their analysis in the Bayesian setting closely follows the proof technique of (Chowdhury & Gopalan, 2017), originally developed for the frequentist setting. We conjecture that this transfer may not be fully justified, as the randomness of $f$ could affect the validity of the argument. In particular, Eq. (11) in (Bayrooti et al., 2025b) and Eq. (18) in (Bayrooti et al., 2025a) may fail to hold in general. This is because, following the definitions in (Bayrooti et al., 2025a), the realization $\xi_{k-1}(\boldsymbol{s}_{h,k}, \boldsymbol{b}_{h,k})$ is not necessarily smaller than the conditional expectation $\mathbb{E}\left[\xi_{k-1}(\boldsymbol{s}_{h,k}, \boldsymbol{a}_{h,k}) \mid \boldsymbol{a}_{h,k} \notin \mathcal{S}_{h,k}\right]$ since the saturated set $\mathcal{S}_{h,k}$ and $\boldsymbol{b}_{h,k} = \arg\min_{\boldsymbol{a} \in \mathcal{A}\backslash\mathcal{S}_{h,k}} \xi_{k-1}(\boldsymbol{s}_{h,k}, \boldsymbol{a})$ are random. On the other hand, we believe this issue can be addressed through suitable modifications to the analysis and the algorithm, such as variance inflation, as in (Chowdhury & Gopalan, 2017), at least for our problem setup, though it remains necessary to verify whether the proof technique of (Chowdhury & Gopalan, 2017) extends consistently for the problem setup in (Bayrooti et al., 2025a;b).

### 3.2. Improved Regret Upper Bounds Regarding $\delta$

We confirmed that GP-TS cannot achieve $O(\log(1/\delta))$ upper bounds with respect to $\delta$ in general by Theorem 3.1. The best known dependence on $\delta$ in the sublinear cumulative regret upper bound for GP-TS is $O(\sqrt{T\gamma_T}\log T/\delta)$ (Russo & Van Roy, 2014; Takeno et al., 2024). The following theorem tightens the dependence on $\delta$ by showing the upper bound of the second moment of cumulative regret:

**Theorem 3.2** (Informal). *Suppose that Assumption 2.1 and $|\mathcal{X}| < \infty$ hold or Assumptions 2.1 and 2.2 hold. Then, if*

*Algorithm 1 runs, the following holds:*

$$\mathbb{E}\left[R_T^2\right] = O(T\gamma_T \log T).$$

*Thus, from Markov's inequality, we have*

$$\Pr\left(R_T = O\left(\sqrt{\frac{T\gamma_T \log T}{\delta}}\right)\right) \geq 1 - \delta,$$

*for any $\delta \in (0, 1)$.*

Theorem 3.2 tightens the term $1/\sqrt{\delta}$ from the existing result $\tilde{O}(\sqrt{T\gamma_T}/\delta)$ (Russo & Van Roy, 2014; Takeno et al., 2024) though the additional $\log T$ factor is required in the case of $|\mathcal{X}| < \infty$ compared with (Takeno et al., 2024). Furthermore, the upper bound of the second moment itself implies the concentration property of the cumulative regret incurred by GP-TS. On the other hand, we have not revealed the tightness for $\delta$. Showing the optimal bound for $\delta$, which maintains the sublinearity in terms of $T$, is an interesting future work. See the following proof sketch and Appendix B for the formal statement of Theorem 3.2 and the proof.

**Proof sketch.** The high-probability result follows from the upper bound on the second moment and Markov's inequality. The proof for the upper bound of the second moment mostly follows the existing analysis (Russo & Van Roy, 2014; Takeno et al., 2024). For simplicity, we first focus on the case of $|\mathcal{X}| < \infty$. From Cauchy–Schwarz inequality and the properties of GP-TS that $(\boldsymbol{x}^* \mid \mathcal{D}_{t-1})$ and $(\boldsymbol{x}_t \mid \mathcal{D}_{t-1})$ are identically distributed and $\boldsymbol{x}^* \perp\!\!\!\perp \boldsymbol{x}_t \mid \mathcal{D}_{t-1}$, we can decompose the regret similarly to (Russo & Van Roy, 2014):

$$\mathbb{E}[R_T^2] \lesssim T\sum_{t=1}^{T} \mathbb{E}\left[\left(f(\boldsymbol{x}^*) - U_t(\boldsymbol{x}^*)\right)_+^2 + \beta_t \sigma_{t-1}^2(\boldsymbol{x}_t)\right],$$

where $\lesssim$ hides $O(1)$ factors, $(c)_+ = \max\{0, c\}$, and $U_t(\boldsymbol{x}) = \mu_{t-1}(\boldsymbol{x}) + \beta_t^{1/2}\sigma_{t-1}(\boldsymbol{x})$ with $\beta_t = O(\log t)$. The sum of posterior variances can be bounded from above by $\gamma_T$ (Srinivas et al., 2010). On the other hand, although the upper bound $\sum_{t=1}^{T} \mathbb{E}\left[\left(f(\boldsymbol{x}^*) - U_t(\boldsymbol{x}^*)\right)_+\right] = O(1)$ is known (Russo & Van Roy, 2014), we need an upper bound of $\mathbb{E}\left[\left(f(\boldsymbol{x}^*) - U_t(\boldsymbol{x}^*)\right)_+^2\right]$ to obtain the upper bound of $R_T^2$. Thus, we provide its upper bound in Lemma E.5. For continuous input domains, for a similar reason, we provide the upper bound of squared discretization error in Lemma E.6.

### 3.3. Lenient Regret Upper Bounds

The following theorem provides the expected lenient regret upper bound for GP-TS:

**Theorem 3.3.** *Fix $\Delta > 0$ and $\delta \in (0, 1)$ and let $\mathcal{T} = \{t \mid f(\boldsymbol{x}^*) - f(\boldsymbol{x}_t) \geq \Delta\}$. Suppose that Assumption 2.1 and $|\mathcal{X}| < \infty$ hold or Assumptions 2.1 and 2.2 hold. Then, if*

*Algorithm 1 runs, the following inequalities hold:*

$$\mathbb{E}[|\mathcal{T}|] \leq T_{\max},$$
$$\mathbb{E}[LR_T] = O\left(\sqrt{\beta_T T_{\max}\tilde{\gamma}_{T_{\max}}}\right).$$

*Here, $T_{\max}$ satisfies:*

$$T_{\max} = \begin{cases} O\left(\frac{\beta_T d \log T}{\Delta^2}\right) & \text{if } k = k_{\mathrm{Lin}}, \\ O\left(\frac{\beta_T \log^{d+1} T}{\Delta^2}\right) & \text{if } k = k_{\mathrm{SE}}, \\ O\left(\left(\frac{\beta_T \log^{\frac{4\nu+d}{2\nu+d}} T}{\Delta^2}\right)^{1+\frac{d}{2\nu}}\right) & \text{if } k = k_{\mathrm{Mat}}, \end{cases}$$

*where $\beta_T = O\left(\log(|\mathcal{X}|T)\right)$ and $\nu > 1/2$ for the case of $|\mathcal{X}| < \infty$ and $\beta_T = O\left(d\log(dT)\right)$ and $\nu > 1$ for the case of continuous $\mathcal{X}$.*

Theorem 3.3 shows that GP-TS attains a polylogarithmic upper bound on the expected lenient regret, matching the order of the existing high-probability upper bounds for GP-UCB (Cai et al., 2021; Iwazaki, 2025b) in terms of $T$. In the next section, we leverage Theorem 3.3 to derive an improved upper bound on the cumulative regret with respect to $T$. We employ a different proof technique than that of (Cai et al., 2021; Iwazaki, 2025b) to obtain a bound on the expectation, as an analysis of expected lenient regret has not previously appeared in the BO literature. We expect that our proof technique can be extended to obtain the Bayesian expected lenient regret upper bound for GP-UCB. See the following proof sketch and Appendix C for the proof.

**Proof sketch.** For simplicity, we here denote the case of $|\mathcal{X}| < \infty$. We employ a proof inspired by that of the elliptical potential count lemma (Flynn & Reeb, 2025; Iwazaki & Takeno, 2025). That is, we leverage the inequality

$$|\mathcal{T}| = \sum_{t\in\mathcal{T}} \min\left\{1, \frac{f(\boldsymbol{x}^*) - f(\boldsymbol{x}_t)}{\Delta}\right\}$$
$$\leq \sum_{t\in\mathcal{T}} \frac{\left(f(\boldsymbol{x}^*) - f(\boldsymbol{x}_t)\right)^2}{\Delta^2},$$

which holds because $\frac{f(\boldsymbol{x}^*) - f(\boldsymbol{x}_t)}{\Delta} \geq 1$ for all $t \in \mathcal{T}$. Then, by applying the properties of the expectation, GP-TS, and GPs, we can obtain

$$\mathbb{E}[|\mathcal{T}|] \lesssim \frac{\beta_T}{\Delta^2}\left(\mathbb{E}\left[\sum_{t\in\mathcal{T}} \sigma_{t-1}^2(\boldsymbol{x}^*)\right] + \mathbb{E}\left[\sum_{t\in\mathcal{T}} \sigma_{t-1}^2(\boldsymbol{x}_t)\right]\right),$$

where $\lesssim$ hides $O(1)$ factors. Thus, we need to obtain an upper bound of $\mathbb{E}\left[\sum_{t\in\mathcal{T}} \sigma_{t-1}^2(\boldsymbol{x}^*)\right] + \mathbb{E}\left[\sum_{t\in\mathcal{T}} \sigma_{t-1}^2(\boldsymbol{x}_t)\right]$, which is nontrivial, especially because $\boldsymbol{x}_t$ and $\boldsymbol{x}^*$ are not necessarily identically distributed given the condition $\mathcal{D}_{t-1}$

and $t \in \mathcal{T}$. Hence, we show Lemma E.7, which states

$$\mathbb{E}\left[\sum_{t\in\mathcal{T}} \sigma_{t-1}^2(\boldsymbol{x}^*)\right] \lesssim \tilde{\gamma}_{\mathbb{E}[|\mathcal{T}|]},$$

$$\mathbb{E}\left[\sum_{t\in\mathcal{T}} \sigma_{t-1}^2(\boldsymbol{x}_t)\right] \lesssim \tilde{\gamma}_{\mathbb{E}[|\mathcal{T}|]},$$

from which we obtain the upper bound of $\mathbb{E}[|\mathcal{T}|]$. Regarding the upper bound of $\mathbb{E}[LR_T]$, we can obtain $\mathbb{E}[LR_T] \lesssim \beta_T^{1/2}\left(\mathbb{E}\left[\sum_{t\in\mathcal{T}} \sigma_{t-1}(\boldsymbol{x}^*)\right] + \mathbb{E}\left[\sum_{t\in\mathcal{T}} \sigma_{t-1}(\boldsymbol{x}_t)\right]\right)$ by the similar proof as that of (Russo & Van Roy, 2014; Takeno et al., 2024). Then, by adopting the following inequalities shown in Lemma E.7,

$$\mathbb{E}\left[\sum_{t\in\mathcal{T}} \sigma_{t-1}(\boldsymbol{x}^*)\right] \lesssim \sqrt{\mathbb{E}[|\mathcal{T}|\tilde{\gamma}_{\mathbb{E}[|\mathcal{T}|]}},$$

$$\mathbb{E}\left[\sum_{t\in\mathcal{T}} \sigma_{t-1}(\boldsymbol{x}_t)\right] \lesssim \sqrt{\mathbb{E}[|\mathcal{T}|\tilde{\gamma}_{\mathbb{E}[|\mathcal{T}|]}},$$

we see that $\mathbb{E}[LR_T] \lesssim \sqrt{\beta_T \mathbb{E}[|\mathcal{T}|\tilde{\gamma}_{\mathbb{E}[|\mathcal{T}|]}}$. Therefore, combining the upper bound of $\mathbb{E}[|\mathcal{T}|]$ concludes the proof. See Appendix E for the details of Lemma E.7.

### 3.4. Improved Regret Upper Bound Regarding $T$

We show the following generalized result before obtaining the regret upper bound specific to GP-TS:

**Lemma 3.4.** *Let $\mathcal{X} = [0, r]^d$. Suppose Assumptions 2.1 and 2.2 hold. In addition, assume the three conditions in Lemma 2.4 and the following event $E$ are true:*

$$E = \left\{\forall t \in \mathbb{N}, f(\boldsymbol{x}^*) - f(\boldsymbol{x}_t) \leq 2\beta_t^{1/2}\sigma_{t-1}(\boldsymbol{x}_t) + \frac{1}{t^2}\right\},$$

*where $\{\beta_t\}_{t\in\mathbb{N}}$ is some monotonically increasing sequence with $\beta_t = O(\log t)$. Then, the following holds:*

$$R_T = \begin{cases} O(\sqrt{T}\log T) & \text{if } k = k_{\mathrm{SE}}, \\ \tilde{O}(\sqrt{T}) & \text{if } k = k_{\mathrm{Mat}} \text{ with } \nu > 2, \end{cases}$$

*where the hidden constants may depend on $d, \nu, \ell, r, \sigma^2$, and the constants $c_{\sup}, c_{\mathrm{gap}}, \rho_{\mathrm{quad}}, c_{\mathrm{quad}}$ in Lemma 2.4.*

Since the assumption in Lemma 3.4 holds for GP-UCB, Lemma 3.4 can be applied to the analysis for GP-UCB. Notably, our result relaxes the condition on Matérn kernels from $2\nu + d \leq \nu^2$ (Iwazaki, 2025b) to $\nu > 2$ by the refined analysis, which is discussed as an open question in (Iwazaki, 2025b). Fortunately, although Iwazaki (2025b) conjecture that this relaxation of the condition requires stronger regularity conditions than those in Lemma 2.4, we did not require such conditions. See Appendix D for the detailed proof, though we provide a short description about the difference from the proof of (Iwazaki, 2025b) and the proof sketch hereafter.

**Difference from the proof of (Iwazaki, 2025b).** Iwazaki (2025b) showed the proof that (i) derive $\tilde{O}(1)$ regret upper bound over the input sets distant from $\boldsymbol{x}^*$, that is, $\mathcal{X}\backslash\mathcal{B}(\rho_{\mathrm{quad}}; \boldsymbol{x}^*)$, by the lenient regret upper bound with fixed $\Delta$ and (ii) obtain the upper bound of regret incurred in $\mathcal{B}(\rho_{\mathrm{quad}}; \boldsymbol{x}^*)$ by utilizing the fact that the inputs $\{\boldsymbol{x}_t\}_{t\in\mathbb{N}}$ concentrate around $\boldsymbol{x}^*$ and the MIG over $\mathcal{B}(\rho; \boldsymbol{x}^*)$ becomes small in proportion to $\rho$. The first difference is that we control $\Delta$ based on $T$ so that the resulting sum of regret is $\tilde{O}(\sqrt{T})$. The second difference, particularly for the case of Matérn kernels, is that we repeatedly apply the arguments like the lenient regret upper bound and the property that the MIG over $\mathcal{B}(\rho; \boldsymbol{x}^*)$ becomes small in proportion to $\rho$. From these differences, we can tighten the regret upper bounds in the derivation, thereby simplifying the analysis for SE kernels and relaxing the required condition for Matérn kernels.

**Proof sketch.** We here concentrate on the case of Matérn kernels, since SE kernels can be handled by a simpler analysis. We divide the index set $[T]$ as

$$\mathcal{T}_0 = \{t \in [T] \mid f(\boldsymbol{x}^*) - f(\boldsymbol{x}_t) \geq \Delta_1\},$$
$$\mathcal{T}_i = \{t \in [T] \mid \Delta_i \geq f(\boldsymbol{x}^*) - f(\boldsymbol{x}_t) \geq \Delta_{i+1}\}, \forall i \in [\bar{i} - 1],$$
$$\mathcal{T}_{\bar{i}} = \{t \in [T] \mid \Delta_{\bar{i}} \geq f(\boldsymbol{x}^*) - f(\boldsymbol{x}_t)\},$$

where we discuss the condition of $\bar{i} \in \mathbb{N}$ and monotonically decreasing $\Delta_2, \ldots, \Delta_{\bar{i}} > 0$ later. Roughly speaking, we tune $\Delta_i$ so that the cumulative regret over the index set $\mathcal{T}_i$ can be bounded from above by $\tilde{O}(\sqrt{T})$ and confirm that $\bar{i}$ can be set to a finite integer even for such $\Delta_i$. If $\sum_{t\in\mathcal{T}_i} f(\boldsymbol{x}^*) - f(\boldsymbol{x}_t) = \tilde{O}(\sqrt{T})$ for all $i = 0, \ldots, \bar{i}$ with $\tilde{i} = O(1)$, then $R_T = \tilde{O}(\bar{i}\sqrt{T}) = \tilde{O}(\sqrt{T})$. First, we obtain the regret bounds for $\mathcal{T}_0$ as $\tilde{O}(\sqrt{T})$ with $\Delta_1 = T^{-\frac{\nu}{2(\nu+d)}}$ by the similar argument as the lenient regret bound. Next, for the sufficiently small $\Delta_1$, we can derive $\boldsymbol{x}_t \in \mathcal{B}\left(\sqrt{c_{\mathrm{quad}}^{-1}\Delta_i}; \boldsymbol{x}^*\right)$ for all $t \in \mathcal{T}_i$ and $i \in [\bar{i}]$ by the condition 3 of Lemma 2.4 and Lemmas 4 and 20 of (Iwazaki, 2025b). This fact suggests that the MIG for $\mathcal{T}_i$ for all $i \in [\bar{i}]$ can be bounded from above in proportion to $\sqrt{c_{\mathrm{quad}}^{-1}\Delta_i}$ based on Corollary 8 of (Iwazaki, 2025b). Therefore, we can tighten the upper bound of $|\mathcal{T}_i|$ for all $i \in [\bar{i}-1]$. In addition, we can tighten the upper bound of $\sum_{t\in\mathcal{T}_i} f(\boldsymbol{x}^*) - f(\boldsymbol{x}_t)$ for all $i \in [\bar{i}]$. From these tightened upper bounds, we can see that $\sum_{t\in\mathcal{T}_i} f(\boldsymbol{x}^*) - f(\boldsymbol{x}_t) = \tilde{O}(\sqrt{T})$ for all $i \in [\bar{i}]$ if

$$\Delta_{\bar{i}} = O\left(T^{-\frac{1}{\nu}}\right),$$
$$\Delta_{i+1} = \tilde{O}\left(T^{-\frac{\nu}{2(\nu+d)}}\Delta_i^{\frac{\nu d}{2(\nu+d)}}\right) \quad \text{for all } i \in [\bar{i} - 1].$$

Finally, we show that the above condition can be satisfied with $\tilde{i} = O(1)$ if $\nu > 2$ by the arguments based on the formulas for the sum of a geometric series.

From Lemma 3.4, we obtain the following improved cumulative regret upper bound of GP-TS:

**Theorem 3.5.** *Fix $\delta \in (0, 1)$ and $\delta_{\mathrm{GP}} \in (0, 1)$. Assume the same premise as in Lemma 2.4. Suppose Assumption 2.2 holds. Then, if Algorithm 1 runs, the following holds with probability at least $1 - \delta - \delta_{\mathrm{GP}}$:*

$$
R_T = \begin{cases} O(\sqrt{T} \log T) & \text{if } k = k_{\mathrm{SE}}, \\ \tilde{O}(\sqrt{T}) & \text{if } k = k_{\mathrm{Mat}} \text{ with } \nu > 2, \end{cases}
$$

*where the hidden constants may depend on $1/\delta, d, \nu, \ell, r, \sigma^2$, and the constants $c_{\mathrm{sup}}, c_{\mathrm{gap}}, \rho_{\mathrm{quad}}, c_{\mathrm{quad}}$ corresponding with $\delta_{\mathrm{GP}}$.*

Theorem 3.5 shows that GP-TS achieves $\tilde{O}(\sqrt{T})$ cumulative regret upper bound as with GP-UCB, which is discussed as an open question in (Iwazaki, 2025b). Thus, compared with the known-best high probability cumulative regret upper bound for GP-TS $O(\sqrt{\gamma_T T \log T}/\delta)$ (Russo & Van Roy, 2014; Takeno et al., 2024) (or $O(\sqrt{\gamma_T T \log T/\delta})$ in Theorem 3.2), Theorem 3.5 shows the improved result with respect to $T$. Note that, although the dependence on $\delta$ is worse than that of GP-UCB, the dependence on $\delta_{\mathrm{GP}}$ is almost the same. Thus, the resulting dependence on probability of GP-TS and GP-UCB may be the same since the dependence on $\delta_{\mathrm{GP}}$ in $c_{\mathrm{sup}}, c_{\mathrm{gap}}, \rho_{\mathrm{quad}}$ and $c_{\mathrm{quad}}$ have not been explicitly provided. Although Lemma 3.4 can be applied to GP-UCB directly, its application to GP-TS requires a careful derivation since we have only shown the expected regret bounds without conditioning on the conditions in Lemma 2.4. See the following proof sketch and Appendix D for the proof.

**Proof sketch.** Under the three conditions in Lemma 2.4, from the similar arguments as that of Lemma 3.4, we can decompose the cumulative regret as

$$
R_T \leq \sum_{i=0}^{\bar{i}} \sum_{t \in \widetilde{\mathcal{T}}_i} f(\boldsymbol{x}^*) - f(\boldsymbol{x}_t),
$$

where the index sets $\widetilde{\mathcal{T}}_i$ are defined as

$$
\widetilde{\mathcal{T}}_0 = \{t \in [T] \mid f(\boldsymbol{x}^*) - f(\boldsymbol{x}_t) \geq \Delta_1\},
$$

$$
\widetilde{\mathcal{T}}_i = \left\{ t \in [T] \ \middle| \ \begin{array}{c} f(\boldsymbol{x}^*) - f(\boldsymbol{x}_t) \geq \Delta_{i+1} \\ \wedge \boldsymbol{x}_t \in \mathcal{B}\left(\sqrt{c_{\mathrm{quad}}^{-1}\Delta_i}; \boldsymbol{x}^*\right) \end{array} \right\}, \forall i \in [\bar{i} - 1],
$$

$$
\widetilde{\mathcal{T}}_{\bar{i}} = \left\{ t \in [T] \mid \boldsymbol{x}_t \in \mathcal{B}\left(\sqrt{c_{\mathrm{quad}}^{-1}\Delta_{\bar{i}}}; \boldsymbol{x}^*\right) \right\}.
$$

Therefore, if we can show the expected regret upper bound $\mathbb{E}\left[\sum_{t \in \widetilde{\mathcal{T}}_i} f(\boldsymbol{x}^*) - f(\boldsymbol{x}_t)\right] = \tilde{O}(\sqrt{T})$ for all $i = 0, \ldots, \bar{i}$ with $\bar{i} = O(1)$, we can obtain $R_T = \tilde{O}(\bar{i}^2\sqrt{T}) = \tilde{O}(\sqrt{T})$ by the union bound and almost the same proof as that of Lemma 3.4. We can show $\mathbb{E}\left[\sum_{t \in \widetilde{\mathcal{T}}_i} f(\boldsymbol{x}^*) - f(\boldsymbol{x}_t)\right] =$

$\tilde{O}(\sqrt{T})$ for all $i = 0, \ldots, \bar{i}$ by the similar proof as those of Theorems 3.2 and 3.3 and the same $\{\Delta_i\}_{i=1}^{\bar{i}}$ and $\bar{i}$ as in the proof of Lemma 3.4.

## 4. Conclusion and Discussion

In this paper, we showed several regret bounds of GP-TS in the Bayesian setting. First, we constructed the simple two-armed problem instance where GP-TS suffers from $1/\delta^c$ regret with probability $\delta$, which implies that $O(\log(1/\delta))$ regret upper bounds cannot be obtained by GP-TS in general. Second, we showed the $\tilde{O}(\sqrt{T\gamma_T/\delta})$ cumulative regret upper bound tighter by a $1/\sqrt{\delta}$ factor than the existing result. Third, we proved the polylogarithmic expected lenient regret upper bounds by the different proof from (Cai et al., 2021; Iwazaki, 2025b). Lastly, we showed the $\tilde{O}(\sqrt{T})$ high-probability cumulative regret upper bound by extending the analysis by Iwazaki (2025b), in which we relaxed the condition of $\nu$ and $d$ for Matérn kernels to only $\nu > 2$, which matches the condition needed for Lemma 2.4. Note that this refined analysis can readily be adapted to that of GP-UCB. Hence, Lemma 3.4 and Theorem 3.5 partially address the limitations discussed as the "Smoothness condition" and the "Extension to other algorithms" in (Iwazaki, 2025b).

However, our analyses still have the limitations listed below:

- **Optimality for $\delta$.** Although we show the regret lower bound for GP-TS in Theorem 3.1, it does not imply the tightness of our Theorem 3.2 in terms of $\delta$. In addition, whether a similar result can be obtained for continuous $\mathcal{X}$ is unclear. Therefore, showing the tightness on $\delta$ and $T$ or a sharper concentration property of $R_T$ incurred by GP-TS for both finite and continuous input domains remains as future work.

- **Analysis of GP-TS with variance inflation.** We conjecture that GP-TS with variance inflation (Chowdhury & Gopalan, 2017) can attain an $O(\sqrt{T\gamma_T \log(T/\delta)})$ cumulative regret upper bound by almost the same proof as that in the frequentist setting (Chowdhury & Gopalan, 2017). However, we expect that extending the analysis by Chowdhury & Gopalan (2017) to the polylogarithmic lenient regret upper bound is not straightforward because of the use of the Azuma–Hoeffding inequality that causes $\tilde{O}(\sqrt{T})$ dependence. This difficulty also hinders the application of the analysis of the improved regret upper bound on $T$. An analysis of such a modified algorithm may be an important direction for achieving a regret upper bound with improved dependence on $\delta$ of GP-TS.

- **Condition on $\nu$ and $d$ for Matérn kernels.** We have relaxed the condition for Matérn kernels on $\nu$ and $d$ from $2\nu + d \leq \nu^2$ (Iwazaki, 2025b) to $\nu > 2$ in

Lemma 3.4 and Theorem 3.5. This condition $\nu > 2$ is also currently needed as the sufficient condition of Assumption 2.2 and Lemma 2.4, which is leveraged in most prior works (Iwazaki, 2025b; Scarlett, 2018; Srinivas et al., 2010). However, this condition $\nu > 2$ cannot be satisfied by the widely used value $\nu = 1/2$ or $3/2$. Therefore, it is crucial to determine whether the condition $\nu > 2$ cannot be avoided or if this condition can be further relaxed.

- **Extension to other algorithms.** As discussed in Section 1.1, there are several algorithms except for GP-UCB and GP-TS with the $\tilde{O}(\sqrt{T\gamma_T})$ expected cumulative regret upper bounds (Takeno et al., 2023; 2024; 2025a;b). Therefore, extending our analyses to those algorithms would be an intriguing future direction.

- **Extension to other problem setup.** Although we and Iwazaki (2025b) focused on the analysis of vanilla BO algorithms, BO has been extended to various problem settings, including multi-fidelity (Kandasamy et al., 2016; 2017; Takeno et al., 2020; 2022), multi-objective (Inatsu et al., 2024; Paria et al., 2020; Zuluaga et al., 2016), constrained (Eriksson & Poloczek, 2021), and parallel BO (Kandasamy et al., 2018; Nava et al., 2022). Thus, extending our analysis to these settings would be of interest to investigate the theoretical performance of BO algorithms.

- **Limitations discussed in prior work.** The limitations named as "Optimality," "Extension to the expected regret," and "Instance dependence analysis in the frequentist setting" discussed in Section 4 of (Iwazaki, 2025b) still remain as interesting directions for future work.

## Acknowledgements

This work was supported by JST PRESTO Grant Number JPMJPR24J6 and JSPS KAKENHI Grant Number JP24K20847.

## Impact Statement

This paper focuses on the theoretical foundation of a machine learning method. Hence, we believe that there is no potential negative societal impact.

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

# A. Proof of Theorem 3.1

**Theorem 3.1.** *Assume that $\mathcal{X} = \{\boldsymbol{x}^{(1)}, \boldsymbol{x}^{(2)}\}$, $\epsilon_t \sim \mathcal{N}(0, 1)$ for all $t \in [T]$, and*

$$\begin{pmatrix} f(\boldsymbol{x}^{(1)}) \\ f(\boldsymbol{x}^{(2)}) \end{pmatrix} \sim \mathcal{N}\left( \begin{pmatrix} 0 \\ 0 \end{pmatrix}, \begin{pmatrix} 1 & 1/2 \\ 1/2 & 1 \end{pmatrix} \right). \tag{1}$$

*Then, if Algorithm 1 runs and $T > e$, the following holds:*

$$\Pr\left( R_T \geq T/2 \right) \geq \frac{c_1}{T^{c_2}}, \tag{2}$$

*where $c_1$ and $c_2$ are strictly positive constants.*

*Proof.* We show that the probability that the following events simultaneously hold can be bounded from below by $\frac{c_1}{T^{17}}$:

$$E_f = \left\{ f(\boldsymbol{x}^{(1)}) \geq 4\sqrt{\log T} \text{ and } f(\boldsymbol{x}^{(2)}) \geq f(\boldsymbol{x}^{(1)}) + 1 \right\}, \tag{3}$$

$$E_\epsilon = \left\{ \forall t \in [[T/2]], \frac{1}{t} \sum_{i=1}^{t} \epsilon_i \geq -1 \right\}, \tag{4}$$

$$E_{\mathrm{TS}} = \left\{ \forall t \in [[T/2]], \boldsymbol{x}_t = \boldsymbol{x}^{(1)} \right\}. \tag{5}$$

If these events simultaneously hold, we can see that $\sum_{t=1}^{T} f(\boldsymbol{x}^*) - f(\boldsymbol{x}_t) \geq T/2$, which concludes the proof.

**Probability of $E_f$:**  From the assumption, we can see that $\left( f(\boldsymbol{x}^{(1)}), f(\boldsymbol{x}^{(2)}) \right)$ and $(Z_0 + Z_1, Z_0 + Z_2)$ are identically distributed, where $Z_0, Z_1$, and $Z_2$ are independent and identically distributed as $\mathcal{N}(0, 1/2)$. Thus, we can rephrase the probability as

$$\Pr\left( f(\boldsymbol{x}^{(1)}) \geq 4\sqrt{\log T} \text{ and } f(\boldsymbol{x}^{(2)}) \geq f(\boldsymbol{x}^{(1)}) + 1 \right) = \Pr\left( Z_0 + Z_1 \geq 4\sqrt{\log T} \text{ and } Z_2 - Z_1 \geq 1 \right) \tag{6}$$

$$\geq \Pr\left( Z_0 \geq 4\sqrt{\log T} \right) \Pr\left( Z_2 - Z_1 \geq 1 \text{ and } Z_1 \geq 0 \right). \tag{7}$$

Since $\Pr\left( Z_2 - Z_1 \geq 1 \text{ and } Z_1 \geq 0 \right)$ does not depend on any variables, we treat it as a constant. Furthermore, using Gaussian anti-concentration inequality $1 - \Phi(c) \geq \frac{1}{\sqrt{2\pi}} \frac{c}{c^2 + 1} \exp(-c^2/2)$, we can obtain the lower bound of $\Pr\left( \sqrt{2}Z_0 \geq 4\sqrt{2\log T} \right)$:

$$\Pr\left( Z_0 \geq 4\sqrt{\log(T)} \right) \geq \frac{1}{\sqrt{2\pi}} \frac{4\sqrt{2\log T}}{32\log T + 1} \exp(-16\log T) \tag{8}$$

$$\geq \frac{1}{\sqrt{2\pi}} \frac{4\sqrt{2\log T}}{32\log T + 1} T^{-16}. \tag{9}$$

Hence, by choosing sufficiently small $c_f > 0$, we can obtain

$$\Pr\left( f(\boldsymbol{x}^{(1)}) \geq 4\sqrt{\log T} \text{ and } f(\boldsymbol{x}^{(2)}) \geq f(\boldsymbol{x}^{(1)}) + 1 \right) \geq c_f T^{-17}. \tag{10}$$

**Probability of $E_\epsilon$:**  We apply Lemma D.1 of (Takeno et al., 2025a):

**Lemma A.1** (Lemma D.1 of (Takeno et al., 2025a)). *Let $\{\epsilon_i\}_{i \geq 1}$ be a mutually independent sequence of standard normal random variables, that is, $\epsilon_i \sim \mathcal{N}(0, 1)$ for all $i \geq 1$. Then, there exists a positive constant $C$ such that*

$$\Pr\left( \forall t \leq T, \frac{\sum_{i=1}^{t} \epsilon_i}{t} \geq -1 \right) \geq C. \tag{11}$$

Therefore, $\Pr(E_\epsilon) \geq c_\epsilon$ with some absolute constant $c_\epsilon > 0$.

**Probability of $E_{\mathrm{TS}}$:** For the first iteration, $\Pr(\boldsymbol{x}_1 = \boldsymbol{x}^{(1)} \mid E_f \wedge E_\epsilon) = \Pr(\boldsymbol{x}_1 = \boldsymbol{x}^{(1)}) = 1/2$ since $\mathcal{D}_0 = \emptyset$, the prior mean is zero, and the algorithm is independent of $f$ and $\epsilon$. We will show

$$\Pr(\boldsymbol{x}_{t+1} = \boldsymbol{x}^{(1)} \mid E_f \wedge E_\epsilon \wedge E_t) \geq 1/2 \qquad \text{for all } t \in [9], \tag{12}$$

$$\Pr(\boldsymbol{x}_{t+1} = \boldsymbol{x}^{(1)} \mid E_f \wedge E_\epsilon \wedge E_t) \geq 1 - 1/T \qquad \text{for all } t \in \{10, \ldots, \lceil T/2 \rceil\}, \tag{13}$$

where the event $E_t = \{\forall i \in [t], \boldsymbol{x}_i = \boldsymbol{x}^{(1)}\}$ and 10 is chosen for simplicity. Then, we can show the probability that $\boldsymbol{x}^{(1)}$ is chosen successively $\lceil T/2 \rceil$ times can be bounded from below by an absolute constant.

First, we show $\Pr(\boldsymbol{x}_{t+1} = \boldsymbol{x}^{(1)} \mid E_f \wedge E_\epsilon \wedge E_t) \geq 1/2$ for $t \in [9]$. If $E_t$ is true, the posterior distribution of $\big(f(\boldsymbol{x}^{(1)}), f(\boldsymbol{x}^{(2)})\big)$ given $\mathcal{D}_t$ can be obtained as

$$\mathcal{N}\left( \begin{pmatrix} \frac{t}{t+1}\bar{y}_t \\ \frac{t}{2(t+1)}\bar{y}_t \end{pmatrix}, \begin{pmatrix} \frac{1}{t+1} & \frac{1}{2(t+1)} \\ \frac{1}{2(t+1)} & \frac{3t/4+1}{t+1} \end{pmatrix} \right), \tag{14}$$

where $\bar{y}_t = \frac{1}{t}\sum_{i=1}^t y_i = f(\boldsymbol{x}^{(1)}) + \frac{1}{t}\sum_{i=1}^t \epsilon_i$. Note that we use the Woodbury formula $(\mathbf{1}\mathbf{1}^\top + \boldsymbol{I}_t)^{-1} = \boldsymbol{I}_t - \frac{\mathbf{1}\mathbf{1}^\top}{t+1}$ as with the proof of Theorem 4.6 in (Takeno et al., 2025a). Therefore, we can obtain the posterior distribution of $f(\boldsymbol{x}^{(2)}) - f(\boldsymbol{x}^{(1)})$ as

$$\mathcal{N}\left( -\frac{t}{2(t+1)}\bar{y}_t, \frac{3t/4+1}{t+1} \right). \tag{15}$$

Hence, for any $\mathcal{D}_t$ that satisfies $E_t$, we can obtain

$$\Pr(\boldsymbol{x}_{t+1} = \boldsymbol{x}^{(1)} \mid \mathcal{D}_t) = \Phi\left( \frac{t\bar{y}_t}{2\sqrt{t+1}\sqrt{3t/4+1}} \right). \tag{16}$$

If $E_f$ and $E_\epsilon$ are true, then $\bar{y}_t \geq 0$ because $f(\boldsymbol{x}^{(1)}) \geq 4\sqrt{\log T}$, $\frac{1}{t}\sum_{i=1}^T \epsilon_i \geq -1$ and $T > e$. Hence, we see that

$$\Pr\left( \boldsymbol{x}_{t+1} = \boldsymbol{x}^{(1)} \mid E_f \wedge E_\epsilon \wedge E_t \right) \tag{17}$$

$$= \mathbb{E}_{\mathcal{D}_t \mid E_f \wedge E_\epsilon \wedge E_t}\left[ \Pr\left( \boldsymbol{x}_{t+1} = \boldsymbol{x}^{(1)} \mid \mathcal{D}_t \wedge E_f \wedge E_\epsilon \wedge E_t \right) \right] \tag{18}$$

$$\stackrel{(a)}{=} \mathbb{E}_{\mathcal{D}_t \mid E_f \wedge E_\epsilon \wedge E_t}\left[ \Pr\left( \boldsymbol{x}_{t+1} = \boldsymbol{x}^{(1)} \mid \mathcal{D}_t \right) \right] \tag{19}$$

$$\stackrel{(b)}{\geq} 1/2, \tag{20}$$

which holds because (a) $\{\boldsymbol{x}_i\}_{i \leq t}$ is fixed given $\mathcal{D}_t$ and $\boldsymbol{x}_{t+1} \perp\!\!\!\perp f, \{\epsilon_t\}_{t \geq 1} \mid \mathcal{D}_t$ and (b) $\bar{y}_t \geq 0$ under the conditions $E_f, E_\epsilon$, and $E_t$.

Next, we obtain the lower bound $\Pr(\boldsymbol{x}_{t+1} = \boldsymbol{x}^{(1)} \mid E_f, E_\epsilon, E_t) \geq 1 - 1/T$ for all $t \geq 10$. If $E_f, E_\epsilon$, and $E_t$ are true, then we see that

$$\frac{1}{t}\sum_{i=1}^t \epsilon_i \stackrel{(a)}{\geq} -1 \tag{21}$$

$$\stackrel{(b)}{\geq} 2\sqrt{2\log\left(\frac{T}{2}\right)} - 4\sqrt{\log T} \tag{22}$$

$$\stackrel{(c)}{\geq} 2\sqrt{2\log\left(\frac{T}{2}\right)} - f(\boldsymbol{x}^{(1)}) \tag{23}$$

$$\stackrel{(d)}{\geq} \frac{2\sqrt{t+1}\sqrt{3t/4+1}}{t}\sqrt{2\log\left(\frac{T}{2}\right)} - f(\boldsymbol{x}^{(1)}), \tag{24}$$

where the inequalities hold because (a) $E_\epsilon$ is true, (b) $g(T) = 2\sqrt{2\log\left(\frac{T}{2}\right)} - 4\sqrt{\log T}$ has the maximum $g(4) \approx -2.354$ at $T = 4$ since $g'(T) = \frac{2}{T}\left(\frac{1}{\sqrt{2\log(T/2)}} - \frac{1}{\sqrt{\log T}}\right)$, (c) $E_f$ is true, and (d) $g(t) = \frac{2\sqrt{t+1}\sqrt{3t/4+1}}{t} = \frac{\sqrt{t+1}\sqrt{3t+4}}{t}$ is monotonically decreasing for $t > 0$ and $g(t) \le g(10) \approx 1.934 \le 2$ for $t \ge 10$ (note that $\log(T/2) > 0$ due to $T > e$). Therefore, we obtain

$$\bar{y}_t = f(\boldsymbol{x}^{(1)}) + \frac{1}{t}\sum_{i=1}^{t}\epsilon_i \ge \frac{2\sqrt{t+1}\sqrt{3t/4+1}}{t}\sqrt{2\log\left(\frac{T}{2}\right)} \tag{25}$$

$$\Leftrightarrow \frac{t\bar{y}_t}{2\sqrt{t+1}\sqrt{3t/4+1}} \ge \sqrt{2\log\left(\frac{T}{2}\right)}. \tag{26}$$

Hence, as with the case of $t \in [9]$, for any $\mathcal{D}_t$ that satisfies $E_f$, $E_\epsilon$, and $E_t$, we see that

$$\Pr(\boldsymbol{x}_{t+1} = \boldsymbol{x}^{(1)} \mid \mathcal{D}_t) = \Phi\left(\frac{t\bar{y}_t}{2\sqrt{t+1}\sqrt{3t/4+1}}\right) \tag{27}$$

$$\ge \Phi\left(\sqrt{2\log\left(\frac{T}{2}\right)}\right) \tag{28}$$

$$\ge 1 - \frac{1}{T}, \tag{29}$$

where we use the inequality $\Phi(c) \ge 1 - \frac{1}{2}e^{-c^2/2}$ (for example, Lemma 5.1 of Srinivas et al., 2010). Thus, for any $t \ge 10$, we derive

$$\Pr\left(\boldsymbol{x}_{t+1} = \boldsymbol{x}^{(1)} \mid E_f \wedge E_\epsilon \wedge E_t\right) \tag{30}$$

$$= \mathbb{E}_{\mathcal{D}_t \mid E_f \wedge E_\epsilon \wedge E_t}\left[\Pr\left(\boldsymbol{x}_{t+1} = \boldsymbol{x}^{(1)} \mid \mathcal{D}_t \wedge E_f \wedge E_\epsilon \wedge E_t\right)\right] \tag{31}$$

$$\overset{(a)}{=} \mathbb{E}_{\mathcal{D}_t \mid E_f \wedge E_\epsilon \wedge E_t}\left[\Pr\left(\boldsymbol{x}_{t+1} = \boldsymbol{x}^{(1)} \mid \mathcal{D}_t\right)\right] \tag{32}$$

$$\ge 1 - 1/T, \tag{33}$$

where the inequality (a) holds because $\{\boldsymbol{x}_i\}_{i \le t}$ is fixed given $\mathcal{D}_t$ and $\boldsymbol{x}_{t+1} \perp\!\!\!\perp f, \{\epsilon_t\}_{t \ge 1} \mid \mathcal{D}_t$.

Consequently, we see that

$$\Pr(E_{\text{TS}} \mid E_f \wedge E_\epsilon) = \Pr\left(\forall t \in [\lceil T/2 \rceil], \boldsymbol{x}_t = \boldsymbol{x}^{(1)} \mid E_f \wedge E_\epsilon\right) \tag{34}$$

$$= \Pr(\boldsymbol{x}_1 = \boldsymbol{x}^{(1)} \mid E_f \wedge E_\epsilon)\prod_{t=1}^{\lceil T/2 \rceil}\Pr\left(\boldsymbol{x}_{t+1} = \boldsymbol{x}^{(1)} \mid E_f \wedge E_\epsilon \wedge E_t\right) \tag{35}$$

$$\ge \frac{1}{2^{10}}\left(1 - \frac{1}{T}\right)^{\max\{\lceil T/2 \rceil - 10, 0\}} \tag{36}$$

$$\ge \frac{1}{2^{10}}\left(1 - \frac{1}{T}\right)^{T/2} \tag{37}$$

$$\ge \frac{1}{2^{11}} =: c_{\text{TS}}, \tag{38}$$

where the second equality follows by the repeated applications of Bayes-rule, and the last inequality follows from Bernoulli's inequality $(1 + c)^r \ge 1 + rc$ for all integer $r \ge 1$ and $c \ge -1$.

**Resulting lower bound:** Combining all the lower bounds for the probability, since $E_f \wedge E_\epsilon \wedge E_{\text{TS}} \Rightarrow \sum_{t=1}^{T} f(\boldsymbol{x}^*) - f(\boldsymbol{x}_t) \geq T/2$, we can see that

$$\Pr\left(\sum_{t=1}^{T} f(\boldsymbol{x}^*) - f(\boldsymbol{x}_t) \geq T/2\right) \geq \Pr(E_f \wedge E_\epsilon \wedge E_{\text{TS}}) \tag{39}$$

$$= \Pr(E_{\text{TS}} \mid E_f \wedge E_\epsilon) \Pr(E_f) \Pr(E_\epsilon) \tag{40}$$

$$\geq \frac{c_f c_\epsilon c_{\text{TS}}}{T^{17}}. \tag{41}$$

$\square$

## B. Proof of Theorem 3.2

Here, we show the proof of the formal version of Theorem 3.2:

**Theorem B.1.** *If Algorithm 1 runs, the following inequality holds:*

- *If Assumption 2.1 holds and $|\mathcal{X}| < \infty$,*

$$\mathbb{E}\left[R_T^2\right] \leq 12C_1 T(\beta_T + 1)\gamma_T + \beta_T^{1/2} = O(T\beta_T\gamma_T), \tag{42}$$

  *where $C_1 = 2/\log(1 + \sigma^{-2})$ and $\beta_t = \max\{1, 2\log(24|\mathcal{X}|T^2/\sqrt{2\pi})\}$ for all $t \in [T]$.*

- *If Assumptions 2.1 and 2.2 hold and $\nu > 1$ for Matérn kernels,*

$$\mathbb{E}[R_T^2] \leq 27 + \beta_T^{1/2} + 18T(\beta_T + 1)(2 + C_1\gamma_T) = O(T\beta_T\gamma_T), \tag{43}$$

  *where $C_1 = 2/\log(1 + \sigma^{-2})$, $\beta_t = \max\{1, 2\log\big(36\lceil drLT\rceil^d T^2/\sqrt{2\pi})\big)\}$ for all $t \in [T]$, $L = \max\{L_\sigma, b\sqrt{\log(ad) + 1}\}$ and $L_\sigma$ is defined as in Lemma 2.3.*

*Proof.* Let $U_t(\boldsymbol{x}) = \mu_{t-1}(\boldsymbol{x}) + \beta_t^{1/2}\sigma_{t-1}(\boldsymbol{x})$ and $(a)_+ = \max\{0, a\}$.

First, we consider the case of $|\mathcal{X}| < \infty$. We can obtain

$$\mathbb{E}\left[R_T^2\right] \leq \mathbb{E}\left[\left(\sum_{t=1}^{T} f(\boldsymbol{x}^*) - U_t(\boldsymbol{x}^*) + U_t(\boldsymbol{x}^*) - g_t(\boldsymbol{x}^*) + g_t(\boldsymbol{x}_t) - U_t(\boldsymbol{x}_t) + U_t(\boldsymbol{x}_t) - f(\boldsymbol{x}_t)\right)^2\right] \tag{44}$$

$$\leq \mathbb{E}\left[\left(\sum_{t=1}^{T}\left(f(\boldsymbol{x}^*) - U_t(\boldsymbol{x}^*)\right)_+ + U_t(\boldsymbol{x}^*) - g_t(\boldsymbol{x}^*) + \left(g_t(\boldsymbol{x}_t) - U_t(\boldsymbol{x}_t)\right)_+ + U_t(\boldsymbol{x}_t) - f(\boldsymbol{x}_t)\right)^2\right], \tag{45}$$

which holds because $g_t(\boldsymbol{x}_t) \geq g_t(\boldsymbol{x}^*)$ from the nature of GP-TS and $f(\boldsymbol{x}^*) - f(\boldsymbol{x}_t) \geq 0$ for all $t \in [T]$. Furthermore, by the Cauchy-Schwarz inequality, we obtain

$$\mathbb{E}\left[R_T^2\right] \leq 6T \sum_{t=1}^{T} \mathbb{E}\left[\left(f(\boldsymbol{x}^*) - U_t(\boldsymbol{x}^*)\right)_+^2 + \beta_t\sigma_{t-1}^2(\boldsymbol{x}^*) + \left(\mu_{t-1}(\boldsymbol{x}^*) - g_t(\boldsymbol{x}^*)\right)^2 \right. \tag{46}$$

$$\left. + \left(g_t(\boldsymbol{x}_t) - U_t(\boldsymbol{x}_t)\right)_+^2 + \beta_t\sigma_{t-1}^2(\boldsymbol{x}_t) + \left(\mu_{t-1}(\boldsymbol{x}_t) - f(\boldsymbol{x}_t)\right)^2\right]. \tag{47}$$

From the independence $\boldsymbol{x}_t \perp\!\!\!\perp f \mid \mathcal{D}_{t-1}$ and $\boldsymbol{x}^* \perp\!\!\!\perp g_t \mid \mathcal{D}_{t-1}$ for all $t \in [T]$, we see that

$$\mathbb{E}\left[\left(\mu_{t-1}(\boldsymbol{x}^*) - g_t(\boldsymbol{x}^*)\right)^2\right] = \mathbb{E}\left[\mathbb{E}\left[\left(\mu_{t-1}(\boldsymbol{x}^*) - g_t(\boldsymbol{x}^*)\right)^2 \mid \boldsymbol{x}^*, \mathcal{D}_{t-1}\right]\right] = \mathbb{E}\left[\sigma_{t-1}^2(\boldsymbol{x}^*)\right], \tag{48}$$

$$\mathbb{E}\left[\left(\mu_{t-1}(\boldsymbol{x}_t) - f(\boldsymbol{x}_t)\right)^2\right] = \mathbb{E}\left[\mathbb{E}\left[\left(\mu_{t-1}(\boldsymbol{x}_t) - f(\boldsymbol{x}_t)\right)^2 \mid \boldsymbol{x}_t, \mathcal{D}_{t-1}\right]\right] = \mathbb{E}\left[\sigma_{t-1}^2(\boldsymbol{x}_t)\right]. \tag{49}$$

In addition, since $\boldsymbol{x}_t \mid \mathcal{D}_{t-1}$ and $\boldsymbol{x}^* \mid \mathcal{D}_{t-1}$ are identically distributed from the nature of GP-TS, we obtain $\mathbb{E}\left[\sigma_{t-1}^2(\boldsymbol{x}^*)\right] = \mathbb{E}\left[\sigma_{t-1}^2(\boldsymbol{x}_t)\right]$. Hence, we derive

$$\mathbb{E}\left[R_T^2\right] \leq 6T \sum_{t=1}^T \mathbb{E}\left[\underbrace{\left(f(\boldsymbol{x}^*) - U_t(\boldsymbol{x}^*)\right)_+^2 + \left(g_t(\boldsymbol{x}_t) - U_t(\boldsymbol{x}_t)\right)_+^2}_{A_1} + \underbrace{2(\beta_t + 1)\sigma_{t-1}^2(\boldsymbol{x}_t)}_{A_2}\right]. \tag{50}$$

Regarding the term $A_1$, from Lemma E.5, we see that

$$6T \sum_{t=1}^T \mathbb{E}\left[\left(f(\boldsymbol{x}^*) - U_t(\boldsymbol{x}^*)\right)_+^2 + \left(g_t(\boldsymbol{x}_t) - U_t(\boldsymbol{x}_t)\right)_+^2\right] \leq 12T \sum_{t=1}^T \frac{2\beta_t^{1/2}}{24T^2} = \beta_T^{1/2}. \tag{51}$$

Regarding the term $A_2$, since $\sum_{t=1}^T \sigma_{t-1}^2(\boldsymbol{x}_t) \leq C_1 \gamma_T$ (Srinivas et al., 2010), we obtain

$$6T \sum_{t=1}^T \mathbb{E}\left[2(\beta_t + 1)\sigma_{t-1}^2(\boldsymbol{x}_t)\right] \leq 12 C_1 T(\beta_T + 1)\gamma_T. \tag{52}$$

Consequently, we derive the following desired result:

$$\mathbb{E}\left[R_T^2\right] \leq 12 C_1 T(\beta_T + 1)\gamma_T + \beta_T^{1/2} = O(T\beta_T\gamma_T). \tag{53}$$

Second, we consider the case of $\mathcal{X} \subset [0, r]^d$. Purely for the sake of analysis, we consider equally divided grid points $\mathcal{X}_t$, where the grid size for each dimension is $\tau_t = \lceil drLT \rceil$, $|\mathcal{X}_t| = \tau_t^d$, and $L = \max\{b(\sqrt{\log(ad)} + 1), L_\sigma\}$ with $L_\sigma$ defined in Lemma 2.3. We denote $[\boldsymbol{x}]_t = \arg\min_{\boldsymbol{x}' \in \mathcal{X}_t} \|\boldsymbol{x} - \boldsymbol{x}'\|_1$. Therefore, $\sup_{\boldsymbol{x} \in \mathcal{X}} \|\boldsymbol{x} - [\boldsymbol{x}]_t\|_1 \leq \frac{1}{LT}$.

Since $g_t(\boldsymbol{x}_t) \geq g_t([\boldsymbol{x}^*]_t)$, we can obtain

$$\mathbb{E}\left[R_T^2\right] \leq \mathbb{E}\left[\left(\sum_{t=1}^T f(\boldsymbol{x}^*) - f([\boldsymbol{x}^*]_t) + f([\boldsymbol{x}^*]_t) - U_t([\boldsymbol{x}^*]_t) + U_t([\boldsymbol{x}^*]_t) - g_t([\boldsymbol{x}^*]_t) + g_t(\boldsymbol{x}_t) - g_t([\boldsymbol{x}_t]_t)\right.\right. \tag{54}$$

$$\left.\left. + g_t([\boldsymbol{x}_t]_t) - U_t([\boldsymbol{x}_t]_t) + U_t([\boldsymbol{x}_t]_t) - f([\boldsymbol{x}_t]_t) + f([\boldsymbol{x}_t]_t) - f(\boldsymbol{x}_t)\right)^2\right] \tag{55}$$

$$\leq 9T \sum_{t=1}^T \mathbb{E}\left[\underbrace{\left(f(\boldsymbol{x}^*) - f([\boldsymbol{x}^*]_t)\right)^2 + \left(g_t(\boldsymbol{x}_t) - g_t([\boldsymbol{x}_t]_t)\right)^2 + \left(f([\boldsymbol{x}_t]_t) - f(\boldsymbol{x}_t)\right)^2}_{A_1}\right. \tag{56}$$

$$\left. + \underbrace{\left(f([\boldsymbol{x}^*]_t) - U_t([\boldsymbol{x}^*]_t)\right)_+^2 + \left(g_t([\boldsymbol{x}_t]_t) - U_t([\boldsymbol{x}_t]_t)\right)_+^2}_{A_2}\right. \tag{57}$$

$$\left. + \underbrace{\left(\mu_{t-1}([\boldsymbol{x}^*]_t) - g_t([\boldsymbol{x}^*]_t)\right)^2 + \left(\mu_{t-1}([\boldsymbol{x}_t]_t) - f([\boldsymbol{x}_t]_t)\right)^2 + \beta_t \sigma_{t-1}^2([\boldsymbol{x}^*]_t) + \beta_t \sigma_{t-1}^2([\boldsymbol{x}_t]_t)}_{A_3}\right]. \tag{58}$$

For the term $A_1$, from Lemma E.6,

$$9T \sum_{t=1}^T \mathbb{E}\left[\left(f(\boldsymbol{x}^*) - f([\boldsymbol{x}^*]_t)\right)^2 + \left(g_t(\boldsymbol{x}_t) - g_t([\boldsymbol{x}_t]_t)\right)^2 + \left(f([\boldsymbol{x}_t]_t) - f(\boldsymbol{x}_t)\right)^2\right] \leq 27T \sum_{t=1}^T \frac{1}{T^2} = 27. \tag{59}$$

Note that, since $g_t$ and $f$ are identically distributed, we can apply Lemma E.6 to $g_t$. For the term $A_2$, as with the case of $|\mathcal{X}| < \infty$, from Lemma E.5

$$9T \sum_{t=1}^T \mathbb{E}\left[\left(f([\boldsymbol{x}^*]_t) - U_t([\boldsymbol{x}^*]_t)\right)_+^2 + \left(g_t([\boldsymbol{x}_t]_t) - U_t([\boldsymbol{x}_t]_t)\right)_+^2\right] \leq 18T \sum_{t=1}^T \frac{2\beta_t^{1/2}}{36T^2} = \beta_T^{1/2}. \tag{60}$$

For the term $A_3$, as with the case of $|\mathcal{X}| < \infty$

$$9T \sum_{t=1}^{T} \mathbb{E}\left[ \left( \mu_{t-1}([\boldsymbol{x}^*]_t) - g_t([\boldsymbol{x}^*]_t) \right)^2 + \left( \mu_{t-1}([\boldsymbol{x}_t]_t) - f([\boldsymbol{x}_t]_t) \right)^2 + \beta_t \sigma_{t-1}^2([\boldsymbol{x}^*]_t) + \beta_t \sigma_{t-1}^2([\boldsymbol{x}_t]_t) \right] \tag{61}$$

$$\leq 18T(\beta_T + 1) \sum_{t=1}^{T} \mathbb{E}\left[ \sigma_{t-1}^2([\boldsymbol{x}_t]_t) \right]. \tag{62}$$

From Lemma 2.3 and $\sigma_{t-1}(\boldsymbol{x}) \leq 1$ for all $\boldsymbol{x} \in \mathcal{X}$, we see that for any $t \geq 1$ and $\mathcal{D}_{t-1}$,

$$\forall \boldsymbol{x}, \boldsymbol{x}' \in \mathcal{X}, |\sigma_{t-1}^2(\boldsymbol{x}) - \sigma_{t-1}^2(\boldsymbol{x}')| \leq |\sigma_{t-1}(\boldsymbol{x}) + \sigma_{t-1}(\boldsymbol{x}')||\sigma_{t-1}(\boldsymbol{x}) - \sigma_{t-1}(\boldsymbol{x}')| \leq 2L_\sigma \|\boldsymbol{x} - \boldsymbol{x}'\|_1. \tag{63}$$

Thus, from $\sum_{t=1}^{T} \sigma_{t-1}^2(\boldsymbol{x}_t) \leq C_1 \gamma_T$ (Srinivas et al., 2010) and the definition of $\mathcal{X}_t$, we obtain

$$\sum_{t=1}^{T} \mathbb{E}\left[ \sigma_{t-1}^2([\boldsymbol{x}_t]_t) \right] \leq \sum_{t=1}^{T} \mathbb{E}\left[ |\sigma_{t-1}^2([\boldsymbol{x}_t]_t) - \sigma_{t-1}^2(\boldsymbol{x}_t)| + \sigma_{t-1}^2(\boldsymbol{x}_t) \right] \tag{64}$$

$$\leq \sum_{t=1}^{T} \frac{2}{T} + \sum_{t=1}^{T} \mathbb{E}\left[ \sigma_{t-1}^2(\boldsymbol{x}_t) \right] \tag{65}$$

$$\leq 2 + C_1 \gamma_T. \tag{66}$$

Combining all the results, we see that

$$\mathbb{E}[R_T^2] \leq 27 + \beta_T^{1/2} + 18T(\beta_T + 1)(2 + C_1 \gamma_T) = O(T\beta_T \gamma_T). \tag{67}$$

$\square$

## C. Proof of Theorem 3.3

First, we show the following general theorem to show the kernel-specific results in Theorem 3.3.

**Theorem C.1.** *Fix $\Delta > 0$ and let $\mathcal{T} = \{t \mid f(\boldsymbol{x}^*) - f(\boldsymbol{x}_t) \geq \Delta\}$. Then, if Algorithm 1 runs, the following inequality holds for any $T \geq 2$:*

- *If Assumption 2.1 holds and $|\mathcal{X}| < \infty$,*

$$\mathbb{E}[|\mathcal{T}|] \leq T_{\max} \leq \frac{32C_1 \beta_T \tilde{\gamma}_{T_{\max}}}{\Delta^2} + \frac{1}{T}, \tag{68}$$

  *where $C_1 = 2/\log(1 + \sigma^{-2})$, $T_{\max} = \max\{t \in \mathbb{R}_{\geq 0} \mid t \leq \frac{32C_1 \beta_T \tilde{\gamma}_t}{\Delta^2} + \frac{1}{T}\}$, and $\beta_t = 2\log(2|\mathcal{X}|T^2 t(t+1))$*

- *If Assumptions 2.1 and 2.2 hold,*

$$\mathbb{E}[|\mathcal{T}|] \leq T_{\max} \leq \frac{32C_1 \beta_T \tilde{\gamma}_{T_{\max}}}{\Delta^2} + \frac{8\pi^2(\beta_T^{1/2} + 1)^2}{\Delta^2} + \frac{1}{T}, \tag{69}$$

  *where $C_1 = 2/\log(1 + \sigma^{-2})$, $T_{\max} = \max\{t \in \mathbb{R}_{\geq 0} \mid t \leq \frac{32C_1 \beta_T \tilde{\gamma}_t}{\Delta^2} + \frac{8\pi^2(\beta_T^{1/2} + 1)^2}{\Delta^2} + \frac{1}{T}\}$, and $\beta_t = 2\log(2\lceil drLt \rceil^d T^2 t(t+1))$, and $L = \max\{b\sqrt{\log(2adT^2)}, L_\sigma\}$ with $L_\sigma$ defined in Lemma 2.3.*

*Proof.* First, we consider the case of $|\mathcal{X}| < \infty$. We define the event for the credible intervals of $f$ and $g_t$:

$$E = \left\{ \forall t \geq 1, \forall \boldsymbol{x} \in \mathcal{X}, |f(\boldsymbol{x}) - \mu_{t-1}(\boldsymbol{x})| \leq \beta_t^{1/2} \sigma_{t-1}(\boldsymbol{x}), |g_t(\boldsymbol{x}) - \mu_{t-1}(\boldsymbol{x})| \leq \beta_t^{1/2} \sigma_{t-1}(\boldsymbol{x}) \right\}, \tag{70}$$

where $\beta_t = 2\log(2|\mathcal{X}|T^2 t(t+1))$. We denote a complementary event of $E$ as $E^c$. Then, using Lemma E.1 (Lemma 5.1 of (Srinivas et al., 2010)) and the union bound, $\Pr(E^c) \leq 1/T^2$. Therefore, we can see that

$$\mathbb{E}[|\mathcal{T}|] = \mathbb{E}[|\mathcal{T}| \mid E]\Pr(E) + \mathbb{E}[|\mathcal{T}| \mid E^c]\Pr(E^c) \tag{71}$$

$$\leq \mathbb{E}[|\mathcal{T}| \mid E] + \frac{1}{T}, \tag{72}$$

where the inequality holds because $\Pr(E) \leq 1$, $\mathbb{E}[|\mathcal{T}| \mid E^c] \leq T$, and $\Pr(E^c) \leq 1/T^2$.

Regarding $\mathbb{E}[|\mathcal{T}| \mid E]$, as with the proof of the elliptical potential count lemma (Flynn & Reeb, 2025; Iwazaki & Takeno, 2025), we can obtain the upper bound as follows:

$$\mathbb{E}[|\mathcal{T}| \mid E] = \mathbb{E}\left[\sum_{t \in \mathcal{T}} \min\left\{1, \frac{f(\boldsymbol{x}^*) - f(\boldsymbol{x}_t)}{\Delta}\right\} \,\middle|\, E\right] \tag{73}$$

$$\overset{(a)}{\leq} \mathbb{E}\left[\sum_{t \in \mathcal{T}} \min\left\{1, \frac{(f(\boldsymbol{x}^*) - f(\boldsymbol{x}_t))^2}{\Delta^2}\right\} \,\middle|\, E\right] \tag{74}$$

$$\leq \mathbb{E}\left[\sum_{t \in \mathcal{T}} \frac{(f(\boldsymbol{x}^*) - f(\boldsymbol{x}_t))^2}{\Delta^2} \,\middle|\, E\right] \tag{75}$$

$$\overset{(b)}{\leq} \frac{1}{\Delta^2}\mathbb{E}\left[\sum_{t \in \mathcal{T}} 4\beta_t\big(\sigma_{t-1}(\boldsymbol{x}^*) + \sigma_{t-1}(\boldsymbol{x}_t)\big)^2 \,\middle|\, E\right] \tag{76}$$

$$\overset{(c)}{\leq} \frac{8\beta_T}{\Delta^2}\left(\mathbb{E}\left[\sum_{t \in \mathcal{T}} \sigma_{t-1}^2(\boldsymbol{x}^*) \,\middle|\, E\right] + \mathbb{E}\left[\sum_{t \in \mathcal{T}} \sigma_{t-1}^2(\boldsymbol{x}_t)) \,\middle|\, E\right]\right) \tag{77}$$

$$\overset{(d)}{\leq} \frac{16\beta_T}{\Delta^2}\left(\mathbb{E}\left[\sum_{t \in \mathcal{T}} \sigma_{t-1}^2(\boldsymbol{x}^*)\right] + \mathbb{E}\left[\sum_{t \in \mathcal{T}} \sigma_{t-1}^2(\boldsymbol{x}_t)\right]\right), \tag{78}$$

where the inequalities hold because of (a) $\frac{f(\boldsymbol{x}^*) - f(\boldsymbol{x}_t)}{\Delta} \geq 1$ for $t \in \mathcal{T}$, (b) $f(\boldsymbol{x}_t) + 2\beta_t^{1/2}\sigma_{t-1}(\boldsymbol{x}_t) \geq g_t(\boldsymbol{x}_t) \geq g_t(\boldsymbol{x}^*) \geq f(\boldsymbol{x}^*) - 2\beta_t^{1/2}\sigma_{t-1}(\boldsymbol{x}^*)$ due to the conditioning by $E$ and the nature of GP-TS, (c) $(a+b)^2 \leq 2(a^2 + b^2)$ for any $a, b \in \mathbb{R}$, and (d) the definition of the conditional expectation $\mathbb{E}[X|E] = \mathbb{E}[X\mathbb{1}_E]/\Pr(E)$ (Example 34.1 of (Billingsley, 2013)), $\Pr(E) \geq 1 - 1/T^2 \geq 1/2$ due to $T \geq 2$, and $\sigma_{t-1}^2(\boldsymbol{x}_t) \geq 0$.

Therefore, combining Lemma E.7, we obtain

$$\mathbb{E}[|\mathcal{T}| \mid E] \leq \frac{32 C_1 \beta_T \tilde{\gamma}_{\mathbb{E}[|\mathcal{T}|]}}{\Delta^2}. \tag{79}$$

Thus, we see that

$$\mathbb{E}[|\mathcal{T}|] \leq \frac{32 C_1 \beta_T \tilde{\gamma}_{\mathbb{E}[|\mathcal{T}|]}}{\Delta^2} + \frac{1}{T} \tag{80}$$

Hence, $\mathbb{E}[|\mathcal{T}|] \in \{t \in \mathbb{R}_{\geq 0} \mid t \leq \frac{32 C_1 \beta_T \tilde{\gamma}_t}{\Delta^2} + \frac{1}{T}\}$ and $\mathbb{E}[|\mathcal{T}|] \leq T_{\max}$ from the definition of $T_{\max}$.

Next, we consider the case of $\mathcal{X} \subset [0, r]^d$. Purely for the sake of analysis, we consider equally divided grid points $\mathcal{X}_t$, where the grid size for each dimension is $\tau_t = \lceil drLt \rceil$ and $|\mathcal{X}_t| = \tau_t^d$, where $L = \max\{b\sqrt{\log(2adT^2)}, L_\sigma\}$ with $L_\sigma$ defined in Lemma 2.3. We denote $[\boldsymbol{x}]_t = \arg\min_{\boldsymbol{x}' \in \mathcal{X}_t} \|\boldsymbol{x} - \boldsymbol{x}'\|_1$. Therefore, $\sup_{\boldsymbol{x} \in \mathcal{X}} \|\boldsymbol{x} - [\boldsymbol{x}]_t\|_1 \leq 1/(Lt)$.

We define

$$E_{\mathrm{Lip}} = \left\{\|f\|_{\mathrm{Lip}} \leq b\sqrt{\log(2adT^2)}\right\}, \tag{81}$$

where $\|f\|_{\mathrm{Lip}}$ is the Lipschitz constant of $f$. From Assumption 2.2, $\Pr(E_{\mathrm{Lip}}^c) \leq 1/(2T^2)$. In addition, we define the event for the credible intervals of $f$ and $g_t$:

$$E = \left\{\forall t \geq 1, \forall \boldsymbol{x} \in \mathcal{X}_t, |f(\boldsymbol{x}) - \mu_{t-1}(\boldsymbol{x})| \leq \beta_t^{1/2}\sigma_{t-1}(\boldsymbol{x}), |g_t(\boldsymbol{x}) - \mu_{t-1}(\boldsymbol{x})| \leq \beta_t^{1/2}\sigma_{t-1}(\boldsymbol{x})\right\}, \tag{82}$$

where $\beta_t = 2\log(2|\mathcal{X}_t|T^2t(t+1)) = 2\log(2\tau_t^d T^2 t(t+1))$. From Lemma E.1 and the union bound, $\Pr(E^c) \le 1/(2T^2)$. Hence, we see $\Pr(E^c \vee E_{\mathrm{Lip}}^c) \le 1/T^2$.

Therefore, we can see that

$$\mathbb{E}[|\mathcal{T}|] = \mathbb{E}[|\mathcal{T}| \mid E \wedge E_{\mathrm{Lip}}]\Pr(E \wedge E_{\mathrm{Lip}}) + \mathbb{E}[|\mathcal{T}| \mid E^c \vee E_{\mathrm{Lip}}^c]\Pr(E^c \vee E_{\mathrm{Lip}}^c) \tag{83}$$

$$\le \mathbb{E}[|\mathcal{T}| \mid E \wedge E_{\mathrm{Lip}}] + \frac{1}{T}, \tag{84}$$

where the inequality holds because $\Pr(E \wedge E_{\mathrm{Lip}}) \le 1$, $|\mathcal{T}| \le T$ a.s., and $\Pr(E^c \vee E_{\mathrm{Lip}}^c) \le 1/T^2$. Then, as with the case of discrete $\mathcal{X}$, we obtain

$$\mathbb{E}[|\mathcal{T}| \mid E \wedge E_{\mathrm{Lip}}] = \mathbb{E}\left[\sum_{t \in \mathcal{T}} \min\left\{1, \frac{f(\boldsymbol{x}^*) - f(\boldsymbol{x}_t)}{\Delta}\right\} \,\middle|\, E \wedge E_{\mathrm{Lip}}\right] \tag{85}$$

$$\le \mathbb{E}\left[\sum_{t \in \mathcal{T}} \min\left\{1, \frac{\left(f(\boldsymbol{x}^*) - f(\boldsymbol{x}_t)\right)^2}{\Delta^2}\right\} \,\middle|\, E \wedge E_{\mathrm{Lip}}\right] \tag{86}$$

$$\le \mathbb{E}\left[\sum_{t \in \mathcal{T}} \frac{\left(f(\boldsymbol{x}^*) - f(\boldsymbol{x}_t)\right)^2}{\Delta^2} \,\middle|\, E \wedge E_{\mathrm{Lip}}\right] \tag{87}$$

where the first inequalities hold because $1 \le \frac{f(\boldsymbol{x}^*) - f(\boldsymbol{x}_t)}{\Delta}$ due to the definition of $\mathcal{T}$.

Since the event $E$ and $E_{\mathrm{Lip}}$ holds and discretization error can be bounded from above by $1/t$ from Lemma E.4, for all $t \ge 1$,

$$f(\boldsymbol{x}^*) - f(\boldsymbol{x}_t) \le |f(\boldsymbol{x}^*) - f([\boldsymbol{x}^*]_t)| + |f([\boldsymbol{x}^*]_t) - \mu_{t-1}([\boldsymbol{x}^*]_t)| + \mu_{t-1}([\boldsymbol{x}^*]_t) - \mu_{t-1}([\boldsymbol{x}_t]_t) \tag{88}$$

$$+ |f([\boldsymbol{x}_t]_t) - \mu_{t-1}([\boldsymbol{x}_t]_t)| + |f(\boldsymbol{x}_t) - f([\boldsymbol{x}_t]_t)| \tag{89}$$

$$\le \mu_{t-1}([\boldsymbol{x}^*]_t) - \mu_{t-1}([\boldsymbol{x}_t]_t) + \beta_t^{1/2}\left(\sigma_{t-1}([\boldsymbol{x}^*]_t) + \sigma_{t-1}([\boldsymbol{x}_t]_t)\right) + \frac{2}{t}. \tag{90}$$

Furthermore, we can obtain

$$f(\boldsymbol{x}^*) - f(\boldsymbol{x}_t) \le 2\beta_t^{1/2}\left(\sigma_{t-1}([\boldsymbol{x}^*]_t) + \sigma_{t-1}([\boldsymbol{x}_t]_t)\right) + \frac{4}{t}, \tag{91}$$

because

$$g_t(\boldsymbol{x}_t) \ge g_t(\boldsymbol{x}^*) \quad \text{(from the nature of GP-TS)} \tag{92}$$

$$\Rightarrow \mu_{t-1}([\boldsymbol{x}_t]_t) + \beta_t^{1/2}\sigma_{t-1}([\boldsymbol{x}_t]_t) + \frac{1}{t} \ge \mu_{t-1}([\boldsymbol{x}^*]_t) - \beta_t^{1/2}\sigma_{t-1}([\boldsymbol{x}^*]_t) - \frac{1}{t} \tag{93}$$

$$\Rightarrow \mu_{t-1}([\boldsymbol{x}^*]_t) - \mu_{t-1}([\boldsymbol{x}_t]_t) \le \beta_t^{1/2}\left(\sigma_{t-1}([\boldsymbol{x}^*]_t) + \sigma_{t-1}([\boldsymbol{x}_t]_t)\right) + \frac{2}{t}. \tag{94}$$

Finally, we see that

$$f(\boldsymbol{x}^*) - f(\boldsymbol{x}_t) \le 2\beta_t^{1/2}\left(\sigma_{t-1}(\boldsymbol{x}^*) + \sigma_{t-1}(\boldsymbol{x}_t)\right) + \frac{4(\beta_t^{1/2} + 1)}{t}, \tag{95}$$

because

$$\sup_{\boldsymbol{x} \in \mathcal{X}} |\sigma_{t-1}(\boldsymbol{x}) - \sigma_{t-1}([\boldsymbol{x}]_t)| \le L_\sigma \|\boldsymbol{x} - [\boldsymbol{x}]_t\|_1 \le \frac{1}{t}, \tag{96}$$

due to Lemma 2.3.

Hence, noting $f(\boldsymbol{x}^*) - f(\boldsymbol{x}_t) \geq 0$ for all $t \geq 1$, we obtain

$$\mathbb{E}[|\mathcal{T}| \mid E \wedge E_{\text{Lip}}] \leq \frac{1}{\Delta^2}\mathbb{E}\left[\sum_{t\in\mathcal{T}}\left(2\beta_t^{1/2}\big(\sigma_{t-1}(\boldsymbol{x}^*)+\sigma_{t-1}(\boldsymbol{x}_t)\big)+\frac{4(\beta_t^{1/2}+1)}{t}\right)^2 \,\middle|\, E \wedge E_{\text{Lip}}\right] \tag{97}$$

$$\overset{(a)}{\leq} \frac{12\beta_T}{\Delta^2}\mathbb{E}\left[\sum_{t\in\mathcal{T}}\big\{\sigma_{t-1}^2(\boldsymbol{x}^*)+\sigma_{t-1}^2(\boldsymbol{x}_t)\big\} \,\middle|\, E \wedge E_{\text{Lip}}\right] + \frac{48}{\Delta^2}\mathbb{E}\left[\sum_{t\in\mathcal{T}}\frac{(\beta_t^{1/2}+1)^2}{t^2} \,\middle|\, E \wedge E_{\text{Lip}}\right] \tag{98}$$

$$\overset{(b)}{\leq} \frac{12\beta_T}{\Delta^2}\mathbb{E}\left[\sum_{t\in\mathcal{T}}\big\{\sigma_{t-1}^2(\boldsymbol{x}^*)+\sigma_{t-1}^2(\boldsymbol{x}_t)\big\} \,\middle|\, E \wedge E_{\text{Lip}}\right] + \frac{8\pi^2(\beta_T^{1/2}+1)^2}{\Delta^2}, \tag{99}$$

where the inequalities hold because of the monotonicity of $\beta_t$, (a) the Cauchy–Schwarz inequality, and (b) $\sum_{t=1}^T 1/t^2 \leq \pi^2/6$.

Finally, for the term $\mathbb{E}\left[\sum_{t\in\mathcal{T}}\big\{\sigma_{t-1}^2(\boldsymbol{x}^*)+\sigma_{t-1}^2(\boldsymbol{x}_t)\big\} \,\middle|\, E \wedge E_{\text{Lip}}\right]$, since we can apply the same proof as the case of $|\mathcal{X}| < \infty$ (with $\Pr(E \wedge E_{\text{Lip}}) \geq 1 - 1/T^2 \geq 3/4$ due to $T \geq 2$), we obtain

$$\mathbb{E}[|\mathcal{T}| \mid E \wedge E_{\text{Lip}}] \leq \frac{32C_1\beta_T\tilde{\gamma}_{\mathbb{E}[|\mathcal{T}|]}}{\Delta^2} + \frac{8\pi^2(\beta_T^{1/2}+1)^2}{\Delta^2}, \tag{100}$$

and

$$\mathbb{E}[|\mathcal{T}|] \leq \frac{32C_1\beta_T\tilde{\gamma}_{\mathbb{E}[|\mathcal{T}|]}}{\Delta^2} + \frac{8\pi^2(\beta_T^{1/2}+1)^2}{\Delta^2} + \frac{1}{T}. \tag{101}$$

Hence, $\mathbb{E}[|\mathcal{T}|] \in \{t \in \mathbb{R}_{\geq 0} \mid t \leq \frac{32C_1\beta_T\tilde{\gamma}_{\mathbb{E}[|\mathcal{T}|]}}{\Delta^2} + \frac{8\pi^2(\beta_T^{1/2}+1)^2}{\Delta^2} + \frac{1}{T}\}$ and $\mathbb{E}[|\mathcal{T}|] \leq T_{\max}$ from the definition of $T_{\max}$. $\qquad\square$

Next, we show the general lenient regret upper bounds for $\sum_{t\in\mathcal{T}} f(\boldsymbol{x}^*) - f(\boldsymbol{x}_t)$.

**Theorem C.2.** *Fix $\Delta > 0$ and $\delta \in (0,1)$ and let $\mathcal{T} = \{t \mid f(\boldsymbol{x}^*) - f(\boldsymbol{x}_t) \geq \Delta\}$. Then, if Algorithm 1 runs, the following inequality holds for any $T \geq 2$:*

- *If Assumption 2.1 holds and $|\mathcal{X}| < \infty$,*

$$\mathbb{E}\left[\sum_{t\in\mathcal{T}} f(\boldsymbol{x}^*) - f(\boldsymbol{x}_t)\right] \leq 4\sqrt{C_1\beta_T\mathbb{E}[|\mathcal{T}|]\tilde{\gamma}_{\mathbb{E}[|\mathcal{T}|]}} + 1 \leq 4\sqrt{C_1\beta_T T_{\max}\tilde{\gamma}_{T_{\max}}} + 1. \tag{102}$$

  *where $\beta_t = \max\{1, 2\log(4|\mathcal{X}|T/\sqrt{2\pi})\}$ for all $t \in [T]$ and $T_{\max}$ is defined as in Theorem C.1.*

- *If Assumptions 2.1 and 2.2 hold,*

$$\mathbb{E}\left[\sum_{t\in\mathcal{T}} f(\boldsymbol{x}^*) - f(\boldsymbol{x}_t)\right] \leq 4 + 4\beta_T^{1/2} + 4\sqrt{C_1\beta_T\mathbb{E}[|\mathcal{T}|]\tilde{\gamma}_{\mathbb{E}[|\mathcal{T}|]}} \leq 4 + 4\beta_T^{1/2} + 4\sqrt{C_1\beta_T T_{\max}\tilde{\gamma}_{T_{\max}}}. \tag{103}$$

  *where $C_1 = 2/\log(1+\sigma^{-2})$, $\beta_t = \max\{1, 2\log(4\lceil drLT\rceil^d T/\sqrt{2\pi})\}$ for all $t \in [T]$, $L = \max\{L_\sigma, b(\sqrt{\log(ad)} + \sqrt{\pi}/2)\}$, $T_{\max}$ is defined as in Theorem C.1, and $L_\sigma$ is defined as in Lemma 2.3.*

*Proof.* Let $U_t(\boldsymbol{x}) = \mu_{t-1}(\boldsymbol{x}) + \beta_t^{1/2}\sigma_{t-1}(\boldsymbol{x})$ and $L_t(\boldsymbol{x}) = \mu_{t-1}(\boldsymbol{x}) - \beta_t^{1/2}\sigma_{t-1}(\boldsymbol{x})$. Moreover, we define $(c)_+ = \max\{0, c\}$.

Since $g_t(\boldsymbol{x}_t) \geq g_t(\boldsymbol{x}^*)$ and $\beta_t$ is monotonically increasing, we can derive

$$\mathbb{E}\left[\sum_{t\in\mathcal{T}} f(\boldsymbol{x}^*) - f(\boldsymbol{x}_t)\right] \tag{104}$$

$$\leq \mathbb{E}\left[\sum_{t\in\mathcal{T}}\left(f(\boldsymbol{x}^*) - U_t(\boldsymbol{x}^*) + 2\beta_t^{1/2}\sigma_{t-1}(\boldsymbol{x}^*) + L_t(\boldsymbol{x}^*) - g_t(\boldsymbol{x}^*) + g_t(\boldsymbol{x}_t) - U_t(\boldsymbol{x}_t) + 2\beta_t^{1/2}\sigma_{t-1}(\boldsymbol{x}_t) + L_t(\boldsymbol{x}_t) - f(\boldsymbol{x}_t)\right)\right] \tag{105}$$

$$\leq \underbrace{\mathbb{E}\left[\sum_{t=1}^T\left(\left(f(\boldsymbol{x}^*) - U_t(\boldsymbol{x}^*)\right)_+ + \left(L_t(\boldsymbol{x}^*) - g_t(\boldsymbol{x}^*)\right)_+ + \left(g_t(\boldsymbol{x}_t) - U_t(\boldsymbol{x}_t)\right)_+ + \left(L_t(\boldsymbol{x}_t) - f(\boldsymbol{x}_t)\right)_+\right)\right]}_{A_1} \tag{106}$$

$$+ \underbrace{2\beta_T^{1/2}\mathbb{E}\left[\sum_{t\in\mathcal{T}}\left(\sigma_{t-1}(\boldsymbol{x}^*) + \sigma_{t-1}(\boldsymbol{x}_t)\right)\right]}_{A_2}. \tag{107}$$

For the terms $A_1$ and $A_2$, by applying Lemma E.5 and Lemma E.7 respectively, we have

$$A_1 \leq \sum_{t=1}^T \frac{4}{4T} = 1, \tag{108}$$

$$A_2 \leq 4\sqrt{C_1\beta_T\mathbb{E}[|\mathcal{T}|]\tilde{\gamma}_{\mathbb{E}[|\mathcal{T}|]}}, \tag{109}$$

which is the desired result. Note that, for the terms $\left(L_t(\boldsymbol{x}^*) - g_t(\boldsymbol{x}^*)\right)_+$ and $\left(L_t(\boldsymbol{x}_t) - f(\boldsymbol{x}_t)\right)_+$, we apply Lemma E.5 to $-g_t$ and $-f$.

Next, we consider the case of $\mathcal{X} \subset [0, r]^d$. Purely for the sake of analysis, we consider equally divided grid points $\mathcal{X}_t$, where the grid size for each dimension is $\tau_t = \lceil drLT \rceil$ and $|\mathcal{X}_t| = \tau_t^d$, where $L = \max\{b(\sqrt{\log(ad)} + \sqrt{\pi}/2), L_\sigma\}$ with $L_\sigma$ defined in Lemma 2.3. We denote $[\boldsymbol{x}]_t = \arg\min_{\boldsymbol{x}'\in\mathcal{X}_t}\|\boldsymbol{x} - \boldsymbol{x}'\|_1$. Therefore, $\sup_{\boldsymbol{x}\in\mathcal{X}}\|\boldsymbol{x} - [\boldsymbol{x}]_t\|_1 \leq \frac{1}{LT}$.

Since $g_t(\boldsymbol{x}_t) \geq g_t(\boldsymbol{x}^*)$ and $\beta_t$ is monotonically increasing, we can derive

$$\mathbb{E}\left[\sum_{t\in\mathcal{T}} f(\boldsymbol{x}^*) - f(\boldsymbol{x}_t)\right] \tag{110}$$

$$\leq \mathbb{E}\left[\sum_{t\in\mathcal{T}}\left(f(\boldsymbol{x}^*) - f([\boldsymbol{x}^*]_t) + f([\boldsymbol{x}^*]_t) - U_t([\boldsymbol{x}^*]_t) + 2\beta_t^{1/2}\sigma_{t-1}([\boldsymbol{x}^*]_t) + L_t([\boldsymbol{x}^*]_t) - g_t([\boldsymbol{x}^*]_t)\right.\right. \tag{111}$$

$$\left.\left. + g_t(\boldsymbol{x}_t) - g_t([\boldsymbol{x}_t]_t) + g_t([\boldsymbol{x}_t]_t) - U_t([\boldsymbol{x}_t]_t) + 2\beta_t^{1/2}\sigma_{t-1}([\boldsymbol{x}_t]_t) + L_t([\boldsymbol{x}_t]_t) - f([\boldsymbol{x}_t]_t) + f([\boldsymbol{x}_t]_t) - f(\boldsymbol{x}_t)\right)\right] \tag{112}$$

$$\leq \underbrace{\mathbb{E}\left[\sum_{t=1}^T\left(|f(\boldsymbol{x}^*) - f([\boldsymbol{x}^*]_t)| + |g_t(\boldsymbol{x}_t) - g_t([\boldsymbol{x}_t]_t)| + |f([\boldsymbol{x}_t]_t) - f(\boldsymbol{x}_t)|\right)\right]}_{A_1} \tag{113}$$

$$+ \underbrace{\mathbb{E}\left[\sum_{t=1}^T\left(\left(f([\boldsymbol{x}^*]_t) - U_t([\boldsymbol{x}^*]_t)\right)_+ + \left(L_t([\boldsymbol{x}^*]_t) - g_t([\boldsymbol{x}^*]_t)\right)_+ + \left(g_t([\boldsymbol{x}_t]_t) - U_t([\boldsymbol{x}_t]_t)\right)_+ + \left(L_t([\boldsymbol{x}_t]_t) - f([\boldsymbol{x}_t]_t)\right)_+\right)\right]}_{A_2} \tag{114}$$

$$+ \underbrace{2\beta_T^{1/2}\mathbb{E}\left[\sum_{t=1}^T\left(|\sigma_{t-1}(\boldsymbol{x}^*) - \sigma_{t-1}([\boldsymbol{x}^*]_t)| + |\sigma_{t-1}(\boldsymbol{x}_t) - \sigma_{t-1}([\boldsymbol{x}_t]_t)|\right)\right]}_{A_3} + \underbrace{2\beta_T^{1/2}\mathbb{E}\left[\sum_{t\in\mathcal{T}}\left(\sigma_{t-1}(\boldsymbol{x}^*) + \sigma_{t-1}(\boldsymbol{x}_t)\right)\right]}_{A_4}. \tag{115}$$

Regarding the term $A_1$, from Lemma E.4, we obtain

$$A_1 \le 3 \sum_{t=1}^{T} \frac{1}{T} = 3. \tag{116}$$

Regarding the term $A_2$, from Lemma E.5, we derive

$$A_2 \le 4 \sum_{t=1}^{T} \frac{1}{4T} = 1. \tag{117}$$

Regarding the term $A_3$, from Lemma 2.3 and the construction of $\mathcal{X}_t$, we have

$$A_3 \le 4\beta_T^{1/2} \sum_{t=1}^{T} \frac{1}{T} = 4\beta_T^{1/2}. \tag{118}$$

Finally, regarding the term $A_4$, from Lemma E.7, we see that

$$A_4 \le 4\sqrt{C_1 \beta_T \mathbb{E}[|\mathcal{T}|] \tilde{\gamma}_{\mathbb{E}[|\mathcal{T}|]}}. \tag{119}$$

Consequently, we obtain the desired result:

$$\mathbb{E}\left[\sum_{t\in\mathcal{T}} f(\boldsymbol{x}^*) - f(\boldsymbol{x}_t)\right] \le 4 + 4\beta_T^{1/2} + 4\sqrt{C_1 \beta_T \mathbb{E}[|\mathcal{T}|] \tilde{\gamma}_{\mathbb{E}[|\mathcal{T}|]}}. \tag{120}$$

$\square$

Finally, we show the kernel-specific results shown below.

**Theorem 3.3.** *Fix* $\Delta > 0$ *and* $\delta \in (0,1)$ *and let* $\mathcal{T} = \{t \mid f(\boldsymbol{x}^*) - f(\boldsymbol{x}_t) \ge \Delta\}$. *Suppose that Assumption 2.1 and* $|\mathcal{X}| < \infty$ *hold or Assumptions 2.1 and 2.2 hold. Then, if Algorithm 1 runs, the following inequalities hold:*

$$\mathbb{E}[|\mathcal{T}|] \le T_{\max}, \tag{121}$$

$$\mathbb{E}\left[\sum_{t\in\mathcal{T}} f(\boldsymbol{x}^*) - f(\boldsymbol{x}_t)\right] = O\left(\sqrt{\beta_T T_{\max} \tilde{\gamma}_{T_{\max}}}\right). \tag{122}$$

*Here,* $T_{\max}$ *satisfies:*

$$T_{\max} = \begin{cases} O\left(\frac{\beta_T d \log T}{\Delta^2}\right) & \text{if } k = k_{\mathrm{Lin}}, \\ O\left(\frac{\beta_T \log^{d+1} T}{\Delta^2}\right) & \text{if } k = k_{\mathrm{SE}}, \\ O\left(\left(\frac{\beta_T \log^{\frac{4\nu+d}{2\nu+d}} T}{\Delta^2}\right)^{1+\frac{d}{2\nu}}\right) & \text{if } k = k_{\mathrm{Mat}}, \end{cases} \tag{123}$$

*where* $\beta_T = O\left(\log(|\mathcal{X}|T)\right)$ *and* $\nu > 1/2$ *for the case of* $|\mathcal{X}| < \infty$ *and* $\beta_T = O\left(d \log(dT)\right)$ *and* $\nu > 1$ *for the case of continuous* $\mathcal{X}$.

*Proof.* From Theorems C.1 and C.2, it suffices to show the upper bound of $T_{\max}$. For both cases of discrete and continuous input domains, we see that

$$T_{\max} = O\left(\frac{\beta_T \tilde{\gamma}_{T_{\max}}}{\Delta^2}\right). \tag{124}$$

For the linear and SE kernels, we obtain a crude upper bound by substituting $\gamma_T$ with $\tilde{\gamma}_{T_{\max}}$ as follows:

$$T_{\max} = O\left(\frac{\beta_T \gamma_T}{\Delta^2}\right). \tag{125}$$

Since $\gamma_T = O(d \log T)$ for the linear kernel and $\gamma_T = O(\log^{d+1} T)$ (Srinivas et al., 2010; Vakili et al., 2021a), we obtain the desired result. For the Matérn kernel, by substituting $\gamma_T = O(T^{\frac{d}{2\nu+d}} \log^{\frac{4\nu+d}{2\nu+d}} T)$ (Iwazaki, 2025b), we see that

$$T_{\max} = O\left(\frac{\beta_T T_{\max}^{\frac{d}{2\nu+d}} \log^{\frac{4\nu+d}{2\nu+d}} T}{\Delta^2}\right) \tag{126}$$

$$\Leftrightarrow T_{\max}^{\frac{2\nu}{2\nu+d}} = O\left(\frac{\beta_T \log^{\frac{4\nu+d}{2\nu+d}} T}{\Delta^2}\right) \tag{127}$$

$$\Leftrightarrow T_{\max} = O\left(\left(\frac{\beta_T \log^{\frac{4\nu+d}{2\nu+d}} T}{\Delta^2}\right)^{1+\frac{d}{2\nu}}\right), \tag{128}$$

which concludes the proof. $\qquad\square$

## D. Proof of Theorem 3.5

First, we show the following general result regarding the improved regret analysis, refined from (Iwazaki, 2025b):

**Lemma 3.4.** *Let $\mathcal{X} = [0, r]^d$. Suppose Assumptions 2.1 and 2.2 hold. In addition, assume the three conditions in Lemma 2.4 and the following event $E$ are true:*

$$E = \left\{\forall t \in \mathbb{N}, f(\boldsymbol{x}^*) - f(\boldsymbol{x}_t) \leq 2\beta_t^{1/2}\sigma_{t-1}(\boldsymbol{x}_t) + \frac{1}{t^2}\right\}, \tag{129}$$

*where $\{\beta_t\}_{t\in\mathbb{N}}$ is some monotonically increasing sequence with $\beta_t = O(\log t)$. Then, the following holds:*

$$\sum_{t=1}^{T} f(\boldsymbol{x}^*) - f(\boldsymbol{x}_t) = \begin{cases} O(\sqrt{T}\log T) & \text{if } k = k_{\mathrm{SE}}, \\ \tilde{O}(\sqrt{T}) & \text{if } k = k_{\mathrm{Mat}} \text{ with } \nu > 2, \end{cases} \tag{130}$$

*where the hidden constants may depend on $d, \nu, \ell, r, \sigma^2$, and the constants $c_{\sup}, c_{\mathrm{gap}}, \rho_{\mathrm{quad}}, c_{\mathrm{quad}}$ in Lemma 2.4.*

*Proof.* We consider the cases of SE and Matérn kernels separately.

**Case of SE kernels.**

We set $\Delta = \frac{\beta_T \gamma_T}{\sqrt{T}}$. If $\Delta > \min\{c_{\mathrm{gap}}, c_{\mathrm{quad}}\rho_{\mathrm{quad}}^2\}$, we can see that $T < \min\{c_{\mathrm{gap}}, c_{\mathrm{quad}}\rho_{\mathrm{quad}}^2\}^{-2}\beta_T^2\gamma_T^2$. Therefore, $\sum_{t=1}^{T} f(\boldsymbol{x}^*) - f(\boldsymbol{x}_t) \leq c_{\sup} \min\{c_{\mathrm{gap}}, c_{\mathrm{quad}}\rho_{\mathrm{quad}}^2\}^{-2}\beta_T^2\gamma_T^2 = O(\log^{2d+4} T)$. Hence, we consider the case of $\Delta \leq \min\{c_{\mathrm{gap}}, c_{\mathrm{quad}}\rho_{\mathrm{quad}}^2\}$ below.

We divide the index set $[T]$ as follows:

$$\mathcal{T} = \{t \in [T] \mid f(\boldsymbol{x}^*) - f(\boldsymbol{x}_t) \geq \Delta\}, \tag{131}$$
$$\mathcal{T}^c = \{t \in [T] \mid f(\boldsymbol{x}^*) - f(\boldsymbol{x}_t) < \Delta\}. \tag{132}$$

We will obtain the regret incurred regarding $\mathcal{T}$ and $\mathcal{T}^c$, respectively.

For the set $\mathcal{T}$, we can obtain the upper bound of $|\mathcal{T}|$ as follows:

$$|\mathcal{T}| = \sum_{t \in \mathcal{T}} \min\left\{1, \frac{f(\boldsymbol{x}^*) - f(\boldsymbol{x}_t)}{\Delta}\right\} \tag{133}$$

$$= \sum_{t \in \mathcal{T}} \min\left\{1, \frac{(f(\boldsymbol{x}^*) - f(\boldsymbol{x}_t))^2}{\Delta^2}\right\} \tag{134}$$

$$\overset{(a)}{\leq} \sum_{t \in \mathcal{T}} \frac{(2\beta_t^{1/2}\sigma_{t-1}(\boldsymbol{x}_t) + 1/t^2)^2}{\Delta^2} \tag{135}$$

$$\overset{(b)}{\leq} \frac{8\beta_T}{\Delta^2} \sum_{t \in \mathcal{T}} \sigma_{t-1}^2(\boldsymbol{x}_t) + \frac{\pi^2}{3\Delta^2} \tag{136}$$

$$\overset{(c)}{\leq} \frac{8C_1\beta_T\gamma_T}{\Delta^2} + \frac{\pi^2}{3\Delta^2} \tag{137}$$

$$= O\left(\frac{T}{\beta_T\gamma_T}\right), \tag{138}$$

where the inequalities hold because (a) the event $E$ holds and $\frac{f(\boldsymbol{x}^*) - f(\boldsymbol{x}_t)}{\Delta} > 1$, (b) $(a+b)^2 \leq 2a^2 + 2b^2$ for all $a, b \in \mathbb{R}$ and $2\sum_{t=1}^T 1/t^4 \leq 2\sum_{t=1}^T 1/t^2 \leq \pi^2/3$, and (c) $\sum_{t=1}^T \sigma_{t-1}^2(\boldsymbol{x}_t) \leq C_1\gamma_T$ with $C_1 = 2/\log(1 + \sigma^{-2})$ (Srinivas et al., 2010), respectively. Therefore, from Lemma E.3, we can obtain

$$\sum_{t \in \mathcal{T}} f(\boldsymbol{x}^*) - f(\boldsymbol{x}_t) \leq \sqrt{C_1\beta_T|\mathcal{T}|\gamma_T} + \pi^2/6 = O(\sqrt{T}). \tag{139}$$

For the set $\mathcal{T}^c$, due to $\Delta \leq \min\{c_{\text{gap}}, c_{\text{quad}}\rho_{\text{quad}}^2\}$ and the condition 3 in Lemma 2.4, we can obtain Lemma E.2, which is modified from the proof of Lemmas 4 and 20 of (Iwazaki, 2025b). Therefore, we see that $\boldsymbol{x}_t \in \mathcal{B}\left(\sqrt{c_{\text{quad}}^{-1}\Delta}; \boldsymbol{x}^*\right)$, where $\mathcal{B}\left(\sqrt{c_{\text{quad}}^{-1}\Delta}; \boldsymbol{x}^*\right)$ is the L2 ball whose center and radius are $\boldsymbol{x}^*$ and $\sqrt{c_{\text{quad}}^{-1}\Delta}$, respectively. Thus, by denoting the MIG over the domain $\mathcal{X}$ as $\gamma_T(\mathcal{X})$, we can obtain

$$\sum_{t \in \mathcal{T}^c} f(\boldsymbol{x}^*) - f(\boldsymbol{x}_t) \leq \sqrt{C_1\beta_T T\gamma_T\left(\mathcal{B}\left(\sqrt{c_{\text{quad}}^{-1}\Delta}; \boldsymbol{x}^*\right)\right)} + \pi^2/6. \tag{140}$$

From Corollary 8 of (Iwazaki, 2025b), we can choose sufficiently large $C_{\text{SE}}$ so that

$$\forall t \geq 2, \forall \eta \in \left(0, \sqrt{\frac{2\ell^2}{e^2 c_d}}\right), \gamma_t\left(\mathcal{B}\left(\eta; 0\right)\right) \leq C_{\text{SE}}\left(\frac{\log^{d+1} t}{\log^d\left(\frac{2\ell^2}{\eta^2 e c_d}\right)} + \log t\right), \tag{141}$$

where $c_d = \max\left\{1, \exp\left(\frac{1}{e}\left(\frac{d}{2} - 1\right)\right)\right\}$. If $\sqrt{c_{\text{quad}}^{-1}\Delta} \geq \sqrt{\frac{2\ell^2}{e^2 c_d}} \Leftrightarrow T \leq \left(\frac{e^2 c_d\beta_T\gamma_T}{2c_{\text{quad}}\ell^2}\right)^2$, we can see that $\sum_{t=1}^T f(\boldsymbol{x}^*) - f(\boldsymbol{x}_t) \leq c_{\text{sup}}\left(\frac{e^2 c_d\beta_T\gamma_T}{2c_{\text{quad}}\ell^2}\right)^2 = O(\log^{2d+4} T)$. Thus, we consider the case of $\sqrt{c_{\text{quad}}^{-1}\Delta} < \sqrt{\frac{2\ell^2}{e^2 c_d}}$ hereafter. Then, we see that

$$\gamma_T\left(\mathcal{B}\left(\sqrt{c_{\text{quad}}^{-1}\Delta}; \boldsymbol{x}^*\right)\right) \leq C_{\text{SE}}\left(\frac{\log^{d+1} T}{\log^d\left(\frac{2\ell^2}{c_{\text{quad}}^{-1} e c_d \Delta}\right)} + \log T\right) \tag{142}$$

$$= C_{\text{SE}}\left(\frac{\log^{d+1} T}{\left((1/2)\log T + \log\left(\frac{2c_{\text{quad}}\ell^2}{\beta_T\gamma_T e c_d}\right)\right)^d} + \log T\right). \tag{143}$$

Since $(1/2) \log T$ is dominant (in terms of $T$) compared with $\log \left( \frac{2c_{\text{quad}} \ell^2}{\beta_T \gamma_T e c_d} \right) = \Omega \left( -(d+2) \log \log T \right)$, we have $\left( (1/2) \log T + \log \left( \frac{2c_{\text{quad}} \ell^2}{\beta_T \gamma_T e c_d} \right) \right)^d = \Omega(\log^d T)$, and thus,

$$\gamma_T \left( \mathcal{B} \left( \sqrt{c_{\text{quad}}^{-1} \Delta}; \boldsymbol{x}^* \right) \right) = O(\log T). \tag{144}$$

Consequently, we obtain

$$\sum_{t \in \mathcal{T}^c} f(\boldsymbol{x}^*) - f(\boldsymbol{x}_t) \leq \sqrt{C_1 \beta_T T \gamma_T \left( \mathcal{B} \left( \sqrt{c_{\text{quad}}^{-1} \Delta}; \boldsymbol{x}^* \right) \right)} + \pi^2/6 \tag{145}$$

$$= O(\sqrt{T} \log T). \tag{146}$$

**Case of Matérn kernels.**

We set $\Delta_1 = T^{-\frac{\nu}{2(\nu+d)}}$. If $\Delta_1 > \min\{c_{\text{gap}}, c_{\text{quad}} \rho_{\text{quad}}^2\}$, we can see that $T < \min\{c_{\text{gap}}, c_{\text{quad}} \rho_{\text{quad}}^2\}^{-\frac{2(\nu+d)}{\nu}}$. Therefore, $\sum_{t=1}^{T} f(\boldsymbol{x}^*) - f(\boldsymbol{x}_t) \leq c_{\text{sup}} \min\{c_{\text{gap}}, c_{\text{quad}} \rho_{\text{quad}}^2\}^{-\frac{2(\nu+d)}{\nu}} = O(1)$. Hence, we consider the case of $\Delta_1 \leq \min\{c_{\text{gap}}, c_{\text{quad}} \rho_{\text{quad}}^2\}$ below.

We divide the index set $[T]$ as follows:

$$\mathcal{T}_0 = \{t \in [T] \mid f(\boldsymbol{x}^*) - f(\boldsymbol{x}_t) \geq \Delta_1\}, \tag{147}$$

$$\mathcal{T}_i = \{t \in [T] \mid \Delta_i \geq f(\boldsymbol{x}^*) - f(\boldsymbol{x}_t) \geq \Delta_{i+1}\}, \forall i \in [\bar{i} - 1], \tag{148}$$

$$\mathcal{T}_{\bar{i}} = \{t \in [T] \mid \Delta_{\bar{i}} \geq f(\boldsymbol{x}^*) - f(\boldsymbol{x}_t)\}, \tag{149}$$

where $\bar{i} \in \mathbb{N}$ and monotonically decreasing $\Delta_2, \ldots, \Delta_{\bar{i}} > 0$ will be chosen hereafter so that the desired result can be obtained.

First, we show the regret incurred for $\mathcal{T}_0$. As with the case of SE kernels, we have

$$|\mathcal{T}_0| = \sum_{t \in \mathcal{T}_0} \min \left\{ 1, \frac{f(\boldsymbol{x}^*) - f(\boldsymbol{x}_t)}{\Delta_1} \right\} \tag{150}$$

$$= \sum_{t \in \mathcal{T}_0} \min \left\{ 1, \frac{\left( f(\boldsymbol{x}^*) - f(\boldsymbol{x}_t) \right)^2}{\Delta_1^2} \right\} \tag{151}$$

$$\leq \frac{8\beta_T}{\Delta_1^2} \sum_{t \in \mathcal{T}_0} \sigma_{t-1}^2(\boldsymbol{x}_t) + \frac{\pi^2}{3\Delta_1^2} \tag{152}$$

$$\leq \frac{8 C_1 \beta_T \gamma_{|\mathcal{T}_0|}}{\Delta_1^2} + \frac{\pi^2}{3\Delta_1^2} \tag{153}$$

$$= \tilde{O} \left( |\mathcal{T}_0|^{\frac{d}{2\nu+d}} T^{\frac{\nu}{(\nu+d)}} \right). \tag{154}$$

Therefore, we see that

$$|\mathcal{T}_0| = \tilde{O} \left( T^{\frac{2\nu+d}{2(\nu+d)}} \right). \tag{155}$$

Note that, if $\frac{\pi^2}{3\Delta_1^2}$ is dominant compared with $\frac{8 C_1 \beta_T \gamma_{|\mathcal{T}_0|}}{\Delta_1^2}$, then $|\mathcal{T}_0| = \tilde{O} \left( T^{\frac{\nu}{(\nu+d)}} \right) = \tilde{O} \left( T^{\frac{2\nu+d}{2(\nu+d)}} \right)$. Hence, from

Lemma E.3, we obtain

$$\sum_{t \in \mathcal{T}_0} f(\boldsymbol{x}^*) - f(\boldsymbol{x}_t) \leq \sqrt{C_1 \beta_T |\mathcal{T}_0| \gamma_{|\mathcal{T}_0|}} \tag{156}$$

$$= \tilde{O}\left( T^{\frac{2\nu+d}{4(\nu+d)}} \left( T^{\frac{2\nu+d}{4(\nu+d)}} \right)^{\frac{d}{2\nu+d}} \right) \tag{157}$$

$$= \tilde{O}\left( \sqrt{T} \right). \tag{158}$$

Next, we consider $\mathcal{T}_i$ for all $i \in [\bar{i}]$. Due to $\Delta_1 \leq \min\{c_{\text{gap}}, c_{\text{quad}}\rho_{\text{quad}}^2\}$ and the condition 3 in Lemma 2.4, we can obtain Lemma E.2, which is modified from the proof of Lemmas 4 and 20 of (Iwazaki, 2025b). Therefore, we see that $\boldsymbol{x}_t \in \mathcal{B}\left( \sqrt{c_{\text{quad}}^{-1}\Delta_i}; \boldsymbol{x}^* \right)$ for all $t \in \mathcal{T}_i$ and $i \in [\bar{i}]$. Thus, from Lemma E.3, we can obtain

$$\forall i \in [\bar{i}], \sum_{t \in \mathcal{T}_i} f(\boldsymbol{x}^*) - f(\boldsymbol{x}_t) \leq \sqrt{C_1 \beta_T |\mathcal{T}_i| \gamma_{|\mathcal{T}_i|} \left( \mathcal{B}\left( \sqrt{c_{\text{quad}}^{-1}\Delta_i}; \boldsymbol{x}^* \right) \right)} + \pi^2/6. \tag{159}$$

From Corollary 8 of (Iwazaki, 2025b), we can choose sufficiently large $C_{\text{Mat}}$ so that

$$\forall t \geq 2, \forall \eta > 0, \gamma_t \left( \mathcal{B}\left( \eta; 0 \right) \right) \leq C_{\text{Mat}} \left( \eta^{\frac{2\nu d}{2\nu+d}} t^{\frac{d}{2\nu+d}} \log^{\frac{4\nu+d}{2\nu+d}} t + \log^2 t \right). \tag{160}$$

Therefore, we have

$$\gamma_{|\mathcal{T}_i|} \left( \mathcal{B}\left( \sqrt{c_{\text{quad}}^{-1}\Delta_i}; \boldsymbol{x}^* \right) \right) = O\left( \Delta_i^{\frac{\nu d}{2\nu+d}} |\mathcal{T}_i|^{\frac{d}{2\nu+d}} \log^{\frac{4\nu+d}{2\nu+d}} |\mathcal{T}_i| + \log^2 |\mathcal{T}_i| \right). \tag{161}$$

If the term $\log^2 |\mathcal{T}_i| = O(\log^2 T)$ is dominant compared with $\Delta_i^{\frac{\nu d}{2\nu+d}} |\mathcal{T}_i|^{\frac{d}{2\nu+d}} \log^{\frac{4\nu+d}{2\nu+d}} |\mathcal{T}_i|$, then the corresponding regret for $\mathcal{T}_i$ can be readily bounded from above by $\tilde{O}(\sqrt{T})$. Thus, hereafter, we focus on the set $\mathcal{I} \subset [\bar{i}]$, for which $\Delta_i^{\frac{\nu d}{2\nu+d}} |\mathcal{T}_i|^{\frac{d}{2\nu+d}} \log^{\frac{4\nu+d}{2\nu+d}} |\mathcal{T}_i|$ is dominant compared with $\log^2 T$. Therefore, we have

$$\forall i \in \mathcal{I}, \sum_{t \in \mathcal{T}_i} f(\boldsymbol{x}^*) - f(\boldsymbol{x}_t) \leq \tilde{O}\left( \sqrt{|\mathcal{T}_i|^{\frac{2\nu+2d}{2\nu+d}} \Delta_i^{\frac{\nu d}{2\nu+d}}} \right). \tag{162}$$

Hence, if we can set $\Delta_{\bar{i}} = \tilde{O}\left( T^{-\frac{1}{\nu}} \right)$ when $\bar{i} \in \mathcal{I}$, by the upper bound $|\mathcal{T}_{\bar{i}}| \leq T$, we can obtain

$$\sum_{t \in \mathcal{T}_{\bar{i}}} f(\boldsymbol{x}^*) - f(\boldsymbol{x}_t) \leq \tilde{O}\left( \sqrt{T} \right). \tag{163}$$

Hereafter, we will confirm the condition that we can choose $\Delta_2, \ldots, \Delta_{\bar{i}-1}$ so that the resulting regret over $\mathcal{T}_1, \ldots, \mathcal{T}_{\bar{i}-1}$ is $\tilde{O}(\sqrt{T})$ and $\Delta_{\bar{i}} = \tilde{O}\left( T^{-\frac{1}{\nu}} \right)$.

As with the previous derivations, for all $i \in \mathcal{I} \setminus \{\bar{i}\}$, we have

$$|\mathcal{T}_i| = \sum_{t \in \mathcal{T}_i} \min\left\{ 1, \frac{f(\boldsymbol{x}^*) - f(\boldsymbol{x}_t)}{\Delta_{i+1}} \right\} \tag{164}$$

$$= \sum_{t \in \mathcal{T}_i} \min\left\{ 1, \frac{(f(\boldsymbol{x}^*) - f(\boldsymbol{x}_t))^2}{\Delta_{i+1}^2} \right\} \tag{165}$$

$$\leq \frac{8\beta_T}{\Delta_{i+1}^2} \sum_{t \in \mathcal{T}_i} \sigma_{t-1}^2(\boldsymbol{x}_t) + \frac{\pi^2}{3\Delta_{i+1}^2} \tag{166}$$

$$\leq \frac{8 C_1 \beta_T \gamma_{|\mathcal{T}_i|} \left( \mathcal{B}\left( \sqrt{c_{\text{quad}}^{-1}\Delta_i}; \boldsymbol{x}^* \right) \right)}{\Delta_{i+1}^2} + \frac{\pi^2}{3\Delta_{i+1}^2}. \tag{167}$$

Then, we consider the following two cases, depending on which term is dominant.

**Case of** $\frac{8C_1\beta_T\gamma_{|\mathcal{T}_i|}\left(\mathcal{B}\left(\sqrt{c_{\text{quad}}^{-1}\Delta_i};\boldsymbol{x}^*\right)\right)}{\Delta_{i+1}^2} \geq \frac{\pi^2}{3\Delta_{i+1}^2}.$ In this case, we see that

$$|\mathcal{T}_i| = \tilde{O}\left(\Delta_i^{\frac{\nu d}{2\nu+d}}\Delta_{i+1}^{-2}|\mathcal{T}_i|^{\frac{d}{2\nu+d}}\right), \tag{168}$$

which immediately suggests that

$$|\mathcal{T}_i| = \tilde{O}\left(\left(\Delta_i^{\frac{\nu d}{2\nu+d}}\Delta_{i+1}^{-2}\right)^{\frac{2\nu+d}{2\nu}}\right). \tag{169}$$

Therefore, for all $i \in \mathcal{I}\backslash\{\bar{i}\}$, we can see that

$$\sum_{t\in\mathcal{T}_i} f(\boldsymbol{x}^*) - f(\boldsymbol{x}_t) \leq \sqrt{C_1\beta_T|\mathcal{T}_i|\gamma_{|\mathcal{T}_i|}\left(\mathcal{B}\left(\sqrt{c_{\text{quad}}^{-1}\Delta_i};\boldsymbol{x}^*\right)\right)} + \pi^2/6 \tag{170}$$

$$= \tilde{O}\left(\sqrt{\left(\left(\Delta_i^{\frac{\nu d}{2\nu+d}}\Delta_{i+1}^{-2}\right)^{\frac{2\nu+d}{2\nu}}\right)^{\frac{2\nu+2d}{2\nu+d}}\Delta_i^{\frac{\nu d}{2\nu+d}}}\right) \tag{171}$$

$$= \tilde{O}\left(\Delta_i^{\frac{d}{2}}\Delta_{i+1}^{-\frac{\nu+d}{\nu}}\right). \tag{172}$$

To obtain $\Delta_i^{\frac{d}{2}}\Delta_{i+1}^{-\frac{\nu+d}{\nu}} = \tilde{O}(\sqrt{T})$, we require the following condition:

$$\Delta_{i+1} = \tilde{\Omega}\left(T^{-\frac{\nu}{2(\nu+d)}}\Delta_i^{\frac{\nu d}{2(\nu+d)}}\right), \tag{173}$$

where $\tilde{\Omega}$ hides polylogarithmic factors.

**Case of** $\frac{8C_1\beta_T\gamma_{|\mathcal{T}_i|}\left(\mathcal{B}\left(\sqrt{c_{\text{quad}}^{-1}\Delta_i};\boldsymbol{x}^*\right)\right)}{\Delta_{i+1}^2} < \frac{\pi^2}{3\Delta_{i+1}^2}.$ In this case, we see that

$$|\mathcal{T}_i| = \tilde{O}\left(\Delta_{i+1}^{-2}\right). \tag{174}$$

Therefore, for all $i \in \mathcal{I}\backslash\{\bar{i}\}$, we can see that

$$\sum_{t\in\mathcal{T}_i} f(\boldsymbol{x}^*) - f(\boldsymbol{x}_t) \leq \sqrt{C_1\beta_T|\mathcal{T}_i|\gamma_{|\mathcal{T}_i|}\left(\mathcal{B}\left(\sqrt{c_{\text{quad}}^{-1}\Delta_i};\boldsymbol{x}^*\right)\right)} + \pi^2/6 \tag{175}$$

$$= \tilde{O}\left(\sqrt{\left(\Delta_{i+1}^{-2}\right)^{\frac{2(\nu+d)}{2\nu+d}}\Delta_i^{\frac{\nu d}{2\nu+d}}}\right) \tag{176}$$

$$= \tilde{O}\left(\Delta_i^{\frac{\nu d}{2(2\nu+d)}}\Delta_{i+1}^{-\frac{2(\nu+d)}{2\nu+d}}\right). \tag{177}$$

To obtain $\Delta_i^{\frac{\nu d}{2(2\nu+d)}}\Delta_{i+1}^{-\frac{2(\nu+d)}{2\nu+d}} = \tilde{O}(\sqrt{T})$, we require the following condition:

$$\Delta_{i+1} = \tilde{\Omega}\left(T^{-\frac{2\nu+d}{4(\nu+d)}}\Delta_i^{\frac{\nu d}{4(\nu+d)}}\right). \tag{178}$$

It suffices to consider the case of $\frac{8C_1\beta_T\gamma_{|\mathcal{T}_i|}\left(\mathcal{B}\left(\sqrt{c_{\text{quad}}^{-1}\Delta_i};\boldsymbol{x}^*\right)\right)}{\Delta_{i+1}^2} \geq \frac{\pi^2}{3\Delta_{i+1}^2}$ due to the following reason. Since we currently focus on $i < \bar{i}$ such that $\Delta_i = \tilde{\Omega}\left(T^{-\frac{1}{\nu}}\right)$, we have

$$T^{-\frac{2\nu+d}{4(\nu+d)}}\Delta_i^{\frac{\nu d}{4(\nu+d)}} = T^{-\frac{\nu}{2(\nu+d)}}\Delta_i^{\frac{\nu d}{2(\nu+d)}}T^{-\frac{d}{4(\nu+d)}}\Delta_i^{-\frac{\nu d}{4(\nu+d)}} = \tilde{O}\left(T^{-\frac{\nu}{2(\nu+d)}}\Delta_i^{\frac{\nu d}{2(\nu+d)}}\right).$$

Therefore, the required condition $\Delta_{i+1} = \tilde{\Omega}\left(T^{-\frac{\nu}{2(\nu+d)}}\Delta_i^{\frac{\nu d}{2(\nu+d)}}\right)$ is a strong condition. Hence, it is sufficient that we can prove for the case that $\frac{8C_1\beta_T\gamma_{|\mathcal{T}_i|}\left(\mathcal{B}\left(\sqrt{c_{\text{quad}}^{-1}\Delta_i};\boldsymbol{x}^*\right)\right)}{\Delta_{i+1}^2} \geq \frac{\pi^2}{3\Delta_{i+1}^2}$ holds for all $i \in \mathcal{I}\setminus\{\bar{i}\}$.

Thus, since $\Delta_1 = T^{-\frac{\nu}{2(\nu+d)}}$, we choose $\Delta_i$ as

$$\Delta_i = \tilde{\Theta}\left(T^{-\frac{\nu}{2(\nu+d)}\sum_{j=1}^{i}\left(\frac{\nu d}{2(\nu+d)}\right)^{j-1}}\right). \tag{179}$$

Under this setting of $\Delta_i$, the regret $\sum_{t\in\mathcal{T}_i} f(\boldsymbol{x}^*) - f(\boldsymbol{x}_t) = \tilde{O}(\sqrt{T})$ for all $i \in [\bar{i}]$. Finally, we will confirm the condition that above $\Delta_i$ can reach to $\Delta_{\bar{i}} = \tilde{O}\left(T^{-\frac{1}{\nu}}\right)$ with finite $\bar{i}$.

**Case of $\frac{\nu d}{2(\nu+d)} \geq 1$.** Due to the condition, we can obtain

$$\Delta_i = \tilde{O}\left(T^{-\frac{i\nu}{2(\nu+d)}}\right). \tag{180}$$

Thus, we can choose $\bar{i} = \left\lceil\frac{2(\nu+d)}{\nu^2}\right\rceil = O(1)$, by which $\Delta_{\bar{i}} = \tilde{O}\left(T^{-\frac{1}{\nu}}\right)$. The condition $\frac{\nu d}{2(\nu+d)} \geq 1$ can be transofrmed as $(\nu-2)d - 2\nu \geq 0$. Therefore, this condition holds if and only if $\nu > 2$ and $d \geq \frac{2\nu}{\nu-2}$ simultaneously hold.

**Case of $\frac{\nu d}{2(\nu+d)} < 1$.** From the above discussion, this condition holds if (i) $\nu \leq 2$ or (ii) $\nu > 2$ and $d < \frac{2\nu}{\nu-2}$ simultaneously hold. If $\sum_{j=1}^{\bar{i}}\left(\frac{\nu d}{2(\nu+d)}\right)^{j-1}$ with finite $\bar{i}$ can be $\frac{2(\nu+d)}{\nu^2}$, we can get $\Delta_{\bar{i}} = \tilde{O}\left(T^{-\frac{1}{\nu}}\right)$. From the formula for the sum of a geometric series, we obtain

$$\sum_{j=1}^{i}\left(\frac{\nu d}{2(\nu+d)}\right)^{j-1} = \frac{1-\left(\frac{\nu d}{2(\nu+d)}\right)^i}{1-\left(\frac{\nu d}{2(\nu+d)}\right)} \to \frac{2(\nu+d)}{2(\nu+d)-\nu d} \qquad \text{as } i \to \infty, \tag{181}$$

since $\left(\frac{\nu d}{2(\nu+d)}\right)^i \to 0$ due to the condition. Therefore, to get finite $\bar{i}$, we require the condition

$$\frac{2(\nu+d)}{2(\nu+d)-\nu d} > \frac{2(\nu+d)}{\nu^2}, \tag{182}$$

which can be rephrased as $(\nu+d)(\nu-2) > 0$. Hence, if $\nu > 2$, we can set $\bar{i}$ as the smallest integer such that

$$\frac{1-\left(\frac{\nu d}{2(\nu+d)}\right)^{\bar{i}}}{1-\left(\frac{\nu d}{2(\nu+d)}\right)} \geq \frac{2(\nu+d)}{\nu^2} \Leftrightarrow 1-\left(\frac{\nu d}{2(\nu+d)}\right)^{\bar{i}} \geq \frac{2(\nu+d)-\nu d}{\nu^2} \tag{183}$$

$$\Leftrightarrow 1 - \frac{2(\nu+d)-\nu d}{\nu^2} \geq \left(\frac{\nu d}{2(\nu+d)}\right)^{\bar{i}} \tag{184}$$

$$\Leftrightarrow \log\left(1-\frac{2(\nu+d)-\nu d}{\nu^2}\right) / \log\left(\frac{\nu d}{2(\nu+d)}\right) \leq \bar{i}. \tag{185}$$

Regarding the last transformation, note that $\left(\frac{\nu d}{2(\nu+d)}\right) < 1 \Leftrightarrow \log\left(\frac{\nu d}{2(\nu+d)}\right) < 0$ from the condition. Thus, we can set

$$\bar{i} = \left\lceil \log\left(1 - \frac{2(\nu+d) - \nu d}{\nu^2}\right) \Big/ \log\left(\frac{\nu d}{2(\nu+d)}\right) \right\rceil = O(1).$$

Summarizing the both cases, we can set $\bar{i} = O(1)$ if $\nu > 2$. Consequently, since the all regret incurred over $\mathcal{T}_0, \dots, \mathcal{T}_{\bar{i}}$ are $\tilde{O}(\sqrt{T})$, the cumulative regret is bounded from above as

$$\sum_{t=1}^{T} f(\boldsymbol{x}^*) - f(\boldsymbol{x}_t) = \sum_{i=0}^{\bar{i}} \tilde{O}(\sqrt{T}) = \tilde{O}(\sqrt{T}),$$

which is the desired result. □

**Theorem 3.5.** *Fix $\delta \in (0,1)$ and $\delta_{\mathrm{GP}} \in (0,1)$. Assume the same premise as in Lemma 2.4. Suppose Assumptions 2.1 and 2.2 hold. Then, if Algorithm 1 runs, the following holds with probability at least $1 - \delta - \delta_{\mathrm{GP}}$:*

$$R_T = \begin{cases} O(\sqrt{T}\log T) & \text{if } k = k_{\mathrm{SE}}, \\ \tilde{O}(\sqrt{T}) & \text{if } k = k_{\mathrm{Mat}} \text{ with } \nu > 2, \end{cases} \tag{186}$$

*where the hidden constants may depend on $1/\delta, d, \nu, \ell, r, \sigma^2$, and the constants $c_{\mathrm{sup}}, c_{\mathrm{gap}}, \rho_{\mathrm{quad}}, c_{\mathrm{quad}}$ corresponding with $\delta_{\mathrm{GP}}$.*

*Proof.* The overview of proof of this theorem is similar to that of Lemma 3.4. However, since the cumulative regret upper bound of GP-TS is shown via the expected regret bounds, we will transform the cumulative regret so that the proof of Lemma 3.4 can be applied. Let the index set $\mathcal{T}_i \subset [T]$ for all $i = 0, \dots, \bar{i}$ be

$$\mathcal{T}_i = \{t \in [T] \mid \Delta_i \geq f(\boldsymbol{x}^*) - f(\boldsymbol{x}_t) \geq \Delta_{i+1}\}, \tag{187}$$

with convenient definition of $\Delta_0 = 2c_{\mathrm{sup}}$ and $\Delta_{\bar{i}+2} = 0$. For the both cases of SE and Matérn kernels, we choose $\{\Delta_i\}_{i=1}^{\bar{i}}$ and $\bar{i}$ as with the proof of Lemma 3.4. Hereafter, we define $\beta_t = O(\log T)$ as the maximum of those defined in Lemmas C.1 and C.2 to apply both cases' proofs.

Assume the three conditions in Lemma 2.4 hold. Then, we can decompose the cumulative regret as follows:

$$\sum_{t=1}^{T} f(\boldsymbol{x}^*) - f(\boldsymbol{x}_t) = \sum_{i=0}^{\bar{i}} \sum_{t \in \mathcal{T}_i} f(\boldsymbol{x}^*) - f(\boldsymbol{x}_t). \tag{188}$$

Under the condition $\Delta_1 \leq \min\{c_{\mathrm{gap}}, c_{\mathrm{quad}}\rho_{\mathrm{quad}}^2\}$ and the condition 3 in Lemma 2.4, we can obtain Lemma E.2, which is modified from the proof of Lemmas 4 and 20 of (Iwazaki, 2025b). (Note that, if $\Delta_1 > \min\{c_{\mathrm{gap}}, c_{\mathrm{quad}}\rho_{\mathrm{quad}}^2\}$, we can get $R_T = \tilde{O}(1)$ as with the proof of Lemma 3.4.) Therefore, we see that $\Delta_i \geq f(\boldsymbol{x}^*) - f(\boldsymbol{x}_t) \Rightarrow \boldsymbol{x}_t \in \mathcal{B}\left(\sqrt{c_{\mathrm{quad}}^{-1}\Delta_i}; \boldsymbol{x}^*\right)$ for all $i \in [\bar{i}]$. Hence, we have the upper bound below:

$$\sum_{t=1}^{T} f(\boldsymbol{x}^*) - f(\boldsymbol{x}_t) \leq \sum_{i=0}^{\bar{i}} \sum_{t \in \widetilde{\mathcal{T}}_i} f(\boldsymbol{x}^*) - f(\boldsymbol{x}_t), \tag{189}$$

with $\widetilde{\mathcal{T}}_i = \left\{t \in [T] \mid f(\boldsymbol{x}^*) - f(\boldsymbol{x}_t) \geq \Delta_{i+1} \wedge \boldsymbol{x}_t \in \mathcal{B}\left(\sqrt{c_{\mathrm{quad}}^{-1}\Delta_i}; \boldsymbol{x}^*\right)\right\}$. We will show upper bounds of $\mathbb{E}\left[\sum_{t \in \widetilde{\mathcal{T}}_i} f(\boldsymbol{x}^*) - f(\boldsymbol{x}_t)\right]$.

At first, for any $\widetilde{\mathcal{T}}_i$, the proof of Theorem C.1 derives

$$\mathbb{E}\left[\left|\widetilde{\mathcal{T}}_i\right|\right] = O\left(\frac{\beta_T \mathbb{E}\left[\sum_{t \in \widetilde{\mathcal{T}}_i}\{\sigma_{t-1}^2(\boldsymbol{x}^*) + \sigma_{t-1}^2(\boldsymbol{x}_t)\}\right]}{\Delta_{i+1}^2}\right) + O\left(\frac{\beta_T}{\Delta_{i+1}^2}\right). \tag{190}$$

Furthermore, the proof of Theorem C.2 derives

$$\mathbb{E}\left[\sum_{t\in\widetilde{\mathcal{T}}_i} f(\boldsymbol{x}^*) - f(\boldsymbol{x}_t)\right] \leq O(\log T) + 2\beta_T^{1/2}\left(\mathbb{E}\left[\sum_{t\in\widetilde{\mathcal{T}}_i}\sigma_{t-1}(\boldsymbol{x}^*)\right] + \mathbb{E}\left[\sum_{t\in\widetilde{\mathcal{T}}_i}\sigma_{t-1}(\boldsymbol{x}_t)\right]\right).$$

By applying Lemma E.8, we obtain[2]

$$\mathbb{E}\left[\left|\widetilde{\mathcal{T}}_i\right|\right] = O\left(\max\left\{\frac{\beta_T\gamma_{\mathbb{E}[|\widetilde{\mathcal{T}}_i|]}\left(\mathcal{B}\left(\sqrt{c_{\mathrm{quad}}^{-1}\Delta_i};0\right)\right)}{\Delta_{i+1}^2}, \frac{\beta_T}{\Delta_{i+1}^2}\right\}\right), \tag{191}$$

and

$$\mathbb{E}\left[\sum_{t\in\widetilde{\mathcal{T}}_i} f(\boldsymbol{x}^*) - f(\boldsymbol{x}_t)\right] = O\left(\sqrt{\beta_T\mathbb{E}\left[\left|\widetilde{\mathcal{T}}_i\right|\right]\tilde{\gamma}_{\mathbb{E}[|\widetilde{\mathcal{T}}_i|]}\left(\mathcal{B}\left(\sqrt{c_{\mathrm{quad}}^{-1}\Delta_i};0\right)\right)}\right). \tag{192}$$

By taking the union bound regarding the conditions in Lemma 2.4 and the Markov's inequality for $\mathbb{E}\left[\sum_{t\in\widetilde{\mathcal{T}}_i} f(\boldsymbol{x}^*) - f(\boldsymbol{x}_t)\right]$ for all $i = 0, \ldots, \bar{i}$, the three conditions in Lemma 2.4 and the following inequalities hold simultaneously with probability at least $1 - \delta - \delta_{\mathrm{GP}}$:

$$\sum_{t\in\widetilde{\mathcal{T}}_i} f(\boldsymbol{x}^*) - f(\boldsymbol{x}_t) = O\left(\frac{\bar{i}+1}{\delta}\sqrt{\beta_T\mathbb{E}\left[\left|\widetilde{\mathcal{T}}_i\right|\right]\tilde{\gamma}_{\mathbb{E}[|\widetilde{\mathcal{T}}_i|]}\left(\mathcal{B}\left(\sqrt{c_{\mathrm{quad}}^{-1}\Delta_i};0\right)\right)}\right). \tag{193}$$

The remainder of the proof is the same as that of Lemma 3.4, although the dependence on $1/\delta$ and $\bar{i}$ is worse. $\qquad\square$

## E. Supporting Lemmas

**Lemma E.1** (Lemma 5.1 of (Srinivas et al., 2010)). *Suppose that Assumption 2.1 holds. Fix $\delta \in (0,1)$. Then, for any $\mathcal{X}$ such that $|\mathcal{X}| < \infty$, the following inequality holds with probability at least $1 - \delta$:*

$$\Pr\left(\forall t \geq 1, \forall \boldsymbol{x} \in \mathcal{X}, |f(\boldsymbol{x}) - \mu_{t-1}(\boldsymbol{x})| \leq \beta_t^{1/2}\sigma_{t-1}(\boldsymbol{x})\right) \geq 1 - \delta, \tag{194}$$

*where $\beta_t = 2\log(|\mathcal{X}|t(t+1)/\delta)$.*

**Lemma E.2** (Modified from the proof of Lemmas 4 and 20 of (Iwazaki, 2025b)). *Suppose $f$ is continuous. Then, under conditions 1 and 3 in Lemma 2.4, $\boldsymbol{x} \in \mathcal{B}\left(\sqrt{c_{\mathrm{quad}}^{-1}\epsilon};\boldsymbol{x}^*\right) \subset \mathcal{B}(\rho_{\mathrm{quad}};\boldsymbol{x}^*)$ holds for any $\boldsymbol{x} \in \mathcal{X}$ such that $f(\boldsymbol{x}^*) - f(\boldsymbol{x}) \leq \epsilon$ with $\epsilon \leq \min\{c_{\mathrm{gap}}, c_{\mathrm{quad}}\rho_{\mathrm{quad}}^2\}$.*

*Proof.* From Lemma 20 of (Iwazaki, 2025b), we see that

$$\boldsymbol{x} \in \mathcal{B}(\rho_{\mathrm{quad}};\boldsymbol{x}^*). \tag{195}$$

Hereafter, we follow the proof of Lemma 4 in (Iwazaki, 2025b). From the condition 3 in Lemma 2.4 and $\boldsymbol{x} \in \mathcal{B}(\rho_{\mathrm{quad}};\boldsymbol{x}^*)$, we obtain

$$\forall \boldsymbol{x} \in \{\boldsymbol{x} \in \mathcal{X} \mid f(\boldsymbol{x}^*) - f(\boldsymbol{x}) \leq \epsilon\}, f(\boldsymbol{x}^*) - f(\boldsymbol{x}) \geq c_{\mathrm{quad}}\|\boldsymbol{x}^* - \boldsymbol{x}\|_2^2. \tag{196}$$

Since $f(\boldsymbol{x}^*) - f(\boldsymbol{x}) \leq \epsilon$, we have

$$c_{\mathrm{quad}}\|\boldsymbol{x}^* - \boldsymbol{x}\|_2^2 \leq \epsilon \Leftrightarrow \|\boldsymbol{x}^* - \boldsymbol{x}\|_2 \leq \sqrt{c_{\mathrm{quad}}^{-1}\epsilon}, \tag{197}$$

which shows the desired result $\boldsymbol{x} \in \mathcal{B}\left(\sqrt{c_{\mathrm{quad}}^{-1}\epsilon};\boldsymbol{x}^*\right)$. $\qquad\square$

---

[2]For the case of Matérn kernels, although the upper bound $\mathbb{E}\left[\left|\widetilde{\mathcal{T}}_i\right|\right] = O(\beta_T/\Delta_{i+1}^2)$ is worse by $\log T$ factor than that in Lemma 3.4, we can apply the same proof since the logarithmic factor does not change the result. Alternatively, we can remove $\beta_T$ by a finer discretization.

**Lemma E.3** (Modified from Lemma 21 of (Iwazaki, 2025b)). *Fix any index set $\mathcal{T} \subset [T]$. Then, we have the following inequality:*

$$\sum_{t \in \mathcal{T}} 2\beta_t^{1/2} \sigma_{t-1}(\boldsymbol{x}_t) \leq 2\sqrt{C_1 \beta_T |\mathcal{T}| \gamma_{|\mathcal{T}|}(\mathcal{X}_{\mathcal{T}})}, \tag{198}$$

*where $C_1 = 2/\log(1 + \sigma^{-2})$, $\mathcal{X}_{\mathcal{T}} = \{\boldsymbol{x}_t\}_{t \in \mathcal{T}}$, and $\gamma_{|\mathcal{T}|}(\mathcal{X}_{\mathcal{T}})$ is the MIG defined over $\mathcal{X}_{\mathcal{T}}$.*

*Proof.* See the proof of Lemma 21 of (Iwazaki, 2025b). $\square$

**Lemma E.4** (Discretized error, Lemma H.2 of (Takeno et al., 2023)). *Suppose Assumptions 2.1 and 2.2 hold. Let $\mathcal{X}_t \subset \mathcal{X}$ be a finite set with each dimension equally divided into $\tau_t = bdru_t\left(\sqrt{\log(ad)} + \sqrt{\pi}/2\right)$ for any $t \geq 1$. Then, for all $t \geq 1$, the following inequality holds:*

$$\mathbb{E}\left[\sup_{\boldsymbol{x} \in \mathcal{X}} |f(\boldsymbol{x}) - f([\boldsymbol{x}]_t)|\right] \leq \frac{1}{u_t}, \tag{199}$$

*where $[\boldsymbol{x}]_t$ is the nearest point in $\mathcal{X}_t$ of $\boldsymbol{x} \in \mathcal{X}$.*

**Lemma E.5.** *Suppose Assumption 2.1 holds. Let $\mathcal{X}_t \subseteq \mathcal{X}$ be a finite set. Then, the following inequality holds:*

$$\mathbb{E}\left[\sup_{\boldsymbol{x} \in \mathcal{X}_t} \left\{ \left(f(\boldsymbol{x}) - U_t(\boldsymbol{x})\right)_+ \right\}\right] \leq \frac{1}{u_t}, \tag{200}$$

$$\mathbb{E}\left[\sup_{\boldsymbol{x} \in \mathcal{X}_t} \left\{ \left(f(\boldsymbol{x}) - U_t(\boldsymbol{x})\right)_+^2 \right\}\right] \leq \frac{2\beta_t^{1/2}}{u_t}, \tag{201}$$

*where $(c)_+ = \max\{c, 0\}$ and $U_t(\boldsymbol{x}) = \mu_{t-1}(\boldsymbol{x}) + \beta_t^{1/2}\sigma_{t-1}(\boldsymbol{x})$ with $\beta_t = \max\{1, 2\log(|\mathcal{X}_t|u_t/\sqrt{2\pi})\}$.*

*Proof.* As with (Russo & Van Roy, 2014), we use the following fact:

$$\mathbb{E}[Z_+] \leq \frac{s}{\sqrt{2\pi}} \exp\left\{-\frac{\alpha^2}{2}\right\}, \tag{202}$$

$$\mathbb{E}[Z_+^2] = (m^2 + s^2)\Phi(\alpha) + ms\phi(\alpha) \leq \frac{s^2(\alpha^2 + 1)}{\sqrt{2\pi}|\alpha|} \exp\left\{-\frac{\alpha^2}{2}\right\}, \tag{203}$$

where $Z \sim \mathcal{N}(m, s^2)$ with $m \leq 0$ and $\alpha = m/s$. The former inequality is used, e.g., in (Russo & Van Roy, 2014; Takeno et al., 2023). Both inequalities can be obtained, e.g., by the moments of the truncated normal distribution.

For the first inequality, following the proof from (Russo & Van Roy, 2014), we can see that

$$\mathbb{E}\left[\sup_{\boldsymbol{x} \in \mathcal{X}_t} (f(\boldsymbol{x}) - U_t(\boldsymbol{x}))_+\right] \leq \mathbb{E}\left[\sum_{\boldsymbol{x} \in \mathcal{X}_t} (f(\boldsymbol{x}) - U_t(\boldsymbol{x}))_+\right] \tag{204}$$

$$\leq \sum_{\boldsymbol{x} \in \mathcal{X}_t} \mathbb{E}_{\mathcal{D}_{t-1}}\left[\mathbb{E}\left[(f(\boldsymbol{x}) - U_t(\boldsymbol{x}))_+ \mid \mathcal{D}_{t-1}\right]\right], \tag{205}$$

where $f(\boldsymbol{x}) - U_t(\boldsymbol{x}) \mid \mathcal{D}_{t-1}$ follows $\mathcal{N}(-\beta_t^{1/2}\sigma_{t-1}(\boldsymbol{x}), \sigma_{t-1}(\boldsymbol{x}))$. Therefore, we can obtain

$$\mathbb{E}_{\mathcal{D}_{t-1}}\left[\sup_{\boldsymbol{x} \in \mathcal{X}_t} \{f(\boldsymbol{x}) - U_t(\boldsymbol{x})\}\right] \leq \mathbb{E}\left[\sum_{\boldsymbol{x} \in \mathcal{X}_t} \frac{\sigma_{t-1}(\boldsymbol{x})}{\sqrt{2\pi}} \exp\left\{-\frac{\beta_t}{2}\right\}\right] \tag{206}$$

$$\leq \frac{|\mathcal{X}_t|}{\sqrt{2\pi}} \exp\left\{-\frac{\beta_t}{2}\right\} \tag{207}$$

$$\leq \frac{1}{u_t}, \tag{208}$$

where the second and third inequalities holds because of $\sigma_{t-1}(\boldsymbol{x}) \leq 1$ and the definition of $\beta_t$, respectively.

For the second inequality, as with the above derivation, we see that

$$\mathbb{E}\left[\sup_{\boldsymbol{x} \in \mathcal{X}_t}\left\{(f(\boldsymbol{x}) - U_t(\boldsymbol{x}))_+^2\right\}\right] \leq \sum_{\boldsymbol{x} \in \mathcal{X}_t} \mathbb{E}_{\mathcal{D}_{t-1}}\left[\mathbb{E}\left[(f(\boldsymbol{x}) - U_t(\boldsymbol{x}))_+^2 \mid \mathcal{D}_{t-1}\right]\right] \tag{209}$$

$$\leq \sum_{\boldsymbol{x} \in \mathcal{X}_t} \mathbb{E}_{\mathcal{D}_{t-1}}\left[\frac{\sigma_{t-1}^2(\boldsymbol{x})(\beta_t + 1)}{\sqrt{2\pi}\beta_t^{1/2}}\exp\left\{-\frac{\beta_t}{2}\right\}\right] \tag{210}$$

$$\leq \frac{2\beta_t^{1/2}|\mathcal{X}_t|}{\sqrt{2\pi}}\exp\left\{-\frac{\beta_t}{2}\right\} \tag{211}$$

$$\leq \frac{2\beta_t^{1/2}}{u_t}, \tag{212}$$

where the second and third inequalities hold because of $\sigma_{t-1}^2(\boldsymbol{x}) \leq 1$ and the definition of $\beta_t(\geq 1)$, respectively. $\square$

**Lemma E.6** (Squared discretized error). *Suppose Assumptions 2.1 and 2.2 hold. Let $\mathcal{X}_t \subset \mathcal{X}$ be a finite set with each dimension equally divided into $\tau_t = \left\lceil bdr\sqrt{u_t\left(\log(ad) + 1\right)}\right\rceil$ for any $t \geq 1$. Then, for all $t \geq 1$, the following inequality holds:*

$$\mathbb{E}\left[\sup_{\boldsymbol{x} \in \mathcal{X}}\left(f(\boldsymbol{x}) - f([\boldsymbol{x}]_t)\right)^2\right] \leq \frac{1}{u_t}, \tag{213}$$

*where $[\boldsymbol{x}]_t$ is the nearest point in $\mathcal{X}_t$ of $\boldsymbol{x} \in \mathcal{X}$.*

*Proof.* Let $L_f$ be the Lipschitz constant of $f$. From Assumption 2.2, we see that

$$\Pr(L_f^2 > c) \leq ad\exp\left(-\frac{c}{b^2}\right). \tag{214}$$

Therefore, we can obtain

$$\mathbb{E}\left[L_f^2\right] = \int_0^\infty \Pr(L_f^2 > c)\mathrm{d}c \tag{215}$$

$$\leq \int_0^\infty \min\{1, ade^{-c/b^2}\}\mathrm{d}c \tag{216}$$

$$= b^2\log(ad) + \int_{b^2\log(ad)}^\infty ade^{-c/b^2}\mathrm{d}c \tag{217}$$

$$= b^2\log(ad) + b^2\int_{\log(ad)}^\infty ade^{-c}\mathrm{d}c \tag{218}$$

$$= b^2\left(\log(ad) + 1\right). \tag{219}$$

Then, we see that

$$\mathbb{E}\left[\sup_{\boldsymbol{x} \in \mathcal{X}}\left(f(\boldsymbol{x}) - f([\boldsymbol{x}]_t)\right)^2\right] \leq \mathbb{E}\left[L_f^2\sup_{\boldsymbol{x} \in \mathcal{X}}\|\boldsymbol{x} - [\boldsymbol{x}]_t\|_1^2\right]. \tag{220}$$

From the construction of $\mathcal{X}_t$, we see that

$$\sup_{\boldsymbol{x} \in \mathcal{X}}\|\boldsymbol{x} - [\boldsymbol{x}]_t\|_1^2 \leq \frac{1}{b^2u_t\left(\log(ad) + 1\right)}. \tag{221}$$

Therefore, we derive

$$\mathbb{E}\left[\sup_{\boldsymbol{x} \in \mathcal{X}}\left(f(\boldsymbol{x}) - f([\boldsymbol{x}]_t)\right)^2\right] \leq \frac{\mathbb{E}\left[L_f^2\right]}{b^2u_t\left(\log(ad) + 1\right)} \tag{222}$$

$$\leq \frac{1}{u_t}. \tag{223}$$

$\square$

**Lemma E.7.** *Suppose Assumption 2.1 holds. Fix $\Delta > 0$ and let $\mathcal{T} = \{t \mid f(\boldsymbol{x}^*) - f(\boldsymbol{x}_t) \geq \Delta\}$. Then, if Algorithm 1 runs, the following inequality holds:*

$$\mathbb{E}\left[\sum_{t \in \mathcal{T}} \sigma_{t-1}^2(\boldsymbol{x}_t)\right] \leq C_1 \tilde{\gamma}_{\mathbb{E}[|\mathcal{T}|]}, \tag{224}$$

$$\mathbb{E}\left[\sum_{t \in \mathcal{T}} \sigma_{t-1}(\boldsymbol{x}_t)\right] \leq \sqrt{C_1 \mathbb{E}[|\mathcal{T}|] \tilde{\gamma}_{\mathbb{E}[|\mathcal{T}|]}}, \tag{225}$$

*and*

$$\mathbb{E}\left[\sum_{t \in \mathcal{T}} \sigma_{t-1}^2(\boldsymbol{x}^*)\right] \leq C_1 \tilde{\gamma}_{\mathbb{E}[|\mathcal{T}|]}, \tag{226}$$

$$\mathbb{E}\left[\sum_{t \in \mathcal{T}} \sigma_{t-1}(\boldsymbol{x}^*)\right] \leq \sqrt{C_1 \mathbb{E}[|\mathcal{T}|] \tilde{\gamma}_{\mathbb{E}[|\mathcal{T}|]}}, \tag{227}$$

*where $C_1 = 2/\log(1 + \sigma^{-2})$.*

*Proof.* First, we can see that

$$\mathbb{E}\left[\sum_{t \in \mathcal{T}} \sigma_{t-1}^2(\boldsymbol{x}_t)\right] \overset{(a)}{\leq} C_1 \mathbb{E}[\gamma_{|\mathcal{T}|}] \tag{228}$$

$$\overset{(b)}{=} C_1 \mathbb{E}[\tilde{\gamma}_{|\mathcal{T}|}] \tag{229}$$

$$\overset{(c)}{\leq} C_1 \tilde{\gamma}_{\mathbb{E}[|\mathcal{T}|]}. \tag{230}$$

The inequalities hold because of (a) $\sum_{t \in \mathcal{T}} \sigma_{t-1}^2(\boldsymbol{x}_t) \leq C_1 \gamma_{|\mathcal{T}|}$ from Lemma E.3, (b) $\gamma_T = \tilde{\gamma}_T$ for any $T \in \mathbb{N}$, and (c) Jensen's inequality and the concavity of $\tilde{\gamma}_T$.

Second, we can obtain

$$\mathbb{E}\left[\sum_{t \in \mathcal{T}} \sigma_{t-1}(\boldsymbol{x}_t)\right] \overset{(a)}{\leq} \mathbb{E}\left[\sqrt{|\mathcal{T}| \sum_{t \in \mathcal{T}} \sigma_{t-1}^2(\boldsymbol{x}_t)}\right] \tag{231}$$

$$\overset{(b)}{\leq} \mathbb{E}\left[\sqrt{C_1 |\mathcal{T}| \tilde{\gamma}_{|\mathcal{T}|}}\right] \tag{232}$$

$$\overset{(c)}{\leq} \sqrt{C_1 \mathbb{E}[|\mathcal{T}|] \mathbb{E}\left[\tilde{\gamma}_{|\mathcal{T}|}\right]} \tag{233}$$

$$\overset{(d)}{\leq} \sqrt{C_1 \mathbb{E}[|\mathcal{T}|] \tilde{\gamma}_{\mathbb{E}[|\mathcal{T}|]}}. \tag{234}$$

The inequalities hold because of (a) Cauchy–Schwarz inequality (b) $\sum_{t \in \mathcal{T}} \sigma_{t-1}^2(\boldsymbol{x}_t) \leq C_1 \gamma_{|\mathcal{T}|}$ from Lemma E.3 and $\gamma_T = \tilde{\gamma}_T$ for any $T \in \mathbb{N}$, (c) Cauchy–Schwarz inequality ($|\mathbb{E}[XY]| \leq \sqrt{\mathbb{E}[X^2]\mathbb{E}[Y^2]}$ for any random variables $X$ and $Y$), and (d) Jensen's inequality and the concavity of $\tilde{\gamma}_t$.

For the proof with respect to $\sigma_{t-1}^2(\boldsymbol{x}^*)$, we use the property of $\mathcal{T}_g = \{t \mid g_t(\boldsymbol{x}_t) - g_t(\boldsymbol{x}^*) \geq \Delta\}$. Let $\mathbb{1}_E$ be the indicator function that is 1 if $E$ is true, and 0 otherwise. Since $\left(\mathbb{1}_{f(\boldsymbol{x}^*) - f(\boldsymbol{x}_t) \geq \Delta} \mid \mathcal{D}_{t-1}\right)$ and $\left(\mathbb{1}_{g_t(\boldsymbol{x}_t) - g_t(\boldsymbol{x}^*) \geq \Delta} \mid \mathcal{D}_{t-1}\right)$ follow the

identical distribution, we can see that

$$\mathbb{E}[|\mathcal{T}|] = \mathbb{E}_{\mathcal{D}_{t-1}}\left[\sum_{t=1}^{T}\mathbb{E}\left[\mathbb{1}_{f(\boldsymbol{x}^*)-f(\boldsymbol{x}_t)\geq\Delta} \mid \mathcal{D}_{t-1}\right]\right] \tag{235}$$

$$= \mathbb{E}_{\mathcal{D}_{t-1}}\left[\sum_{t=1}^{T}\mathbb{E}\left[\mathbb{1}_{g_t(\boldsymbol{x}_t)-g_t(\boldsymbol{x}^*)\geq\Delta} \mid \mathcal{D}_{t-1}\right]\right] \tag{236}$$

$$= \mathbb{E}\left[\sum_{t=1}^{T}\mathbb{1}_{g_t(\boldsymbol{x}_t)-g_t(\boldsymbol{x}^*)\geq\Delta}\right] \tag{237}$$

$$= \mathbb{E}[|\mathcal{T}_g|]. \tag{238}$$

Then, we can show the results regarding $\sigma_{t-1}^2(\boldsymbol{x}^*)$ as follows:

$$\mathbb{E}\left[\sum_{t\in\mathcal{T}}\sigma_{t-1}^2(\boldsymbol{x}^*)\right] = \mathbb{E}_{\mathcal{D}_{t-1}}\left[\sum_{t\in[T]}\mathbb{E}\left[\sigma_{t-1}^2(\boldsymbol{x}^*)\mathbb{1}_{f(\boldsymbol{x}^*)-f(\boldsymbol{x}_t)\geq\Delta} \mid \mathcal{D}_{t-1}\right]\right] \tag{239}$$

$$\overset{(a)}{=} \mathbb{E}_{\mathcal{D}_{t-1}}\left[\sum_{t\in[T]}\mathbb{E}\left[\sigma_{t-1}^2(\boldsymbol{x}_t)\mathbb{1}_{g_t(\boldsymbol{x}_t)-g_t(\boldsymbol{x}^*)\geq\Delta} \mid \mathcal{D}_{t-1}\right]\right] \tag{240}$$

$$= \mathbb{E}\left[\sum_{t\in[T]}\sigma_{t-1}^2(\boldsymbol{x}_t)\mathbb{1}_{g_t(\boldsymbol{x}_t)-g_t(\boldsymbol{x}^*)\geq\Delta}\right] \tag{241}$$

$$\overset{(b)}{\leq} C_1\mathbb{E}[\gamma_{|\mathcal{T}_g|}] \tag{242}$$

$$\overset{(c)}{\leq} C_1\tilde{\gamma}_{\mathbb{E}[|\mathcal{T}_g|]} \tag{243}$$

$$= C_1\tilde{\gamma}_{\mathbb{E}[|\mathcal{T}|]}. \tag{244}$$

The equalities and inequalities hold because (a) $\left(\sigma_{t-1}^2(\boldsymbol{x}^*)\mathbb{1}_{f(\boldsymbol{x}^*)-f(\boldsymbol{x}_t)\geq\Delta} \mid \mathcal{D}_{t-1}\right)$ and $\left(\sigma_{t-1}^2(\boldsymbol{x}_t)\mathbb{1}_{g_t(\boldsymbol{x}_t)-g_t(\boldsymbol{x}^*)\geq\Delta} \mid \mathcal{D}_{t-1}\right)$ follow the identical distribution, (b) $\sum_{t\in\mathcal{T}_g}\sigma_{t-1}^2(\boldsymbol{x}_t) \leq C_1\gamma_{|\mathcal{T}_g|}$ from Lemma E.3, and (c) $\gamma_T = \tilde{\gamma}_T$ for any $T \in \mathbb{N}$ and $\tilde{\gamma}_T$ is a concave function.

Furthermore, by the same reasons shown above, we can see that

$$\mathbb{E}\left[\sum_{t\in\mathcal{T}}\sigma_{t-1}(\boldsymbol{x}^*)\right] = \mathbb{E}_{\mathcal{D}_{t-1}}\left[\sum_{t\in[T]}\mathbb{E}\left[\sigma_{t-1}(\boldsymbol{x}^*)\mathbb{1}_{f(\boldsymbol{x}^*)-f(\boldsymbol{x}_t)\geq\Delta} \mid \mathcal{D}_{t-1}\right]\right] \tag{245}$$

$$= \mathbb{E}_{\mathcal{D}_{t-1}}\left[\sum_{t\in[T]}\mathbb{E}\left[\sigma_{t-1}(\boldsymbol{x}_t)\mathbb{1}_{g_t(\boldsymbol{x}_t)-g_t(\boldsymbol{x}^*)\geq\Delta} \mid \mathcal{D}_{t-1}\right]\right] \tag{246}$$

$$= \mathbb{E}\left[\sum_{t\in[T]}\sigma_{t-1}(\boldsymbol{x}_t)\mathbb{1}_{g_t(\boldsymbol{x}_t)-g_t(\boldsymbol{x}^*)\geq\Delta}\right] \tag{247}$$

$$\leq \mathbb{E}\left[\sqrt{C_1|\mathcal{T}_g|\gamma_{|\mathcal{T}_g|}}\right] \tag{248}$$

$$\leq \sqrt{C_1\mathbb{E}[|\mathcal{T}_g|]\gamma_{\mathbb{E}[|\mathcal{T}_g|]}} \tag{249}$$

$$\leq \sqrt{C_1\mathbb{E}[|\mathcal{T}|]\gamma_{\mathbb{E}[|\mathcal{T}|]}}. \tag{250}$$

$\square$

**Lemma E.8.** *Suppose Assumption 2.1 holds and $k = k_{\mathrm{SE}}$ or $k = k_{\mathrm{Mat}}$ with $\nu > 1/2$. Fix $\Delta, \rho > 0$ and let $\mathcal{T} =$*

$\{t \in [T] \mid f(\boldsymbol{x}^*) - f(\boldsymbol{x}_t) \geq \Delta \wedge \boldsymbol{x}_t \in \mathcal{B}(\rho; \boldsymbol{x}^*)\}$. *Then, if Algorithm 1 runs, the following inequality holds:*

$$\mathbb{E}\left[\sum_{t \in \mathcal{T}} \sigma_{t-1}^2(\boldsymbol{x}_t)\right] \leq C_1 \tilde{\gamma}_{\mathbb{E}[|\mathcal{T}|]}(\mathcal{B}(\rho; 0)), \tag{251}$$

$$\mathbb{E}\left[\sum_{t \in \mathcal{T}} \sigma_{t-1}(\boldsymbol{x}_t)\right] \leq \sqrt{C_1 \mathbb{E}[|\mathcal{T}|] \tilde{\gamma}_{\mathbb{E}[|\mathcal{T}|]}(\mathcal{B}(\rho; 0))}, \tag{252}$$

*and*

$$\mathbb{E}\left[\sum_{t \in \mathcal{T}} \sigma_{t-1}^2(\boldsymbol{x}^*)\right] \leq C_1 \tilde{\gamma}_{\mathbb{E}[|\mathcal{T}|]}(\mathcal{B}(\rho; 0)), \tag{253}$$

$$\mathbb{E}\left[\sum_{t \in \mathcal{T}} \sigma_{t-1}(\boldsymbol{x}^*)\right] \leq \sqrt{C_1 \mathbb{E}[|\mathcal{T}|] \tilde{\gamma}_{\mathbb{E}[|\mathcal{T}|]}(\mathcal{B}(\rho; 0))}, \tag{254}$$

*where $C_1 = 2/\log(1 + \sigma^{-2})$ and $\tilde{\gamma}_T(\mathcal{X})$ is the MIG over the domain $\mathcal{X}$.*

*Proof.* First, we can see that

$$\mathbb{E}\left[\sum_{t \in \mathcal{T}} \sigma_{t-1}^2(\boldsymbol{x}_t)\right] \overset{(a)}{\leq} C_1 \mathbb{E}[\gamma_{|\mathcal{T}|}(\mathcal{B}(\rho; \boldsymbol{x}^*))] \tag{255}$$

$$\overset{(b)}{=} C_1 \mathbb{E}[\tilde{\gamma}_{|\mathcal{T}|}(\mathcal{B}(\rho; 0))] \tag{256}$$

$$\overset{(c)}{\leq} C_1 \tilde{\gamma}_{\mathbb{E}[|\mathcal{T}|]}(\mathcal{B}(\rho; 0)). \tag{257}$$

The inequalities hold because of (a) $\sum_{t \in \mathcal{T}} \sigma_{t-1}^2(\boldsymbol{x}_t) \leq C_1 \gamma_{|\mathcal{T}|}(\mathcal{B}(\rho; \boldsymbol{x}^*))$ from Lemma E.3 and $\boldsymbol{x}_t \in \mathcal{B}(\rho; \boldsymbol{x}^*)$, (b) $\gamma_T(\mathcal{B}(\rho; \boldsymbol{x}^*)) = \tilde{\gamma}_T(\mathcal{B}(\rho; 0))$ for any $T \in \mathbb{N}$ due to the definition of $\tilde{\gamma}$ and the shift-invariant property of the SE and Matérn kernels, and (c) Jensen's inequality and the concavity of $\tilde{\gamma}_T$.

Second, we can obtain

$$\mathbb{E}\left[\sum_{t \in \mathcal{T}} \sigma_{t-1}(\boldsymbol{x}_t)\right] \overset{(a)}{\leq} \mathbb{E}\left[\sqrt{|\mathcal{T}| \sum_{t \in \mathcal{T}} \sigma_{t-1}^2(\boldsymbol{x}_t)}\right] \tag{258}$$

$$\overset{(b)}{\leq} \mathbb{E}\left[\sqrt{C_1 |\mathcal{T}| \gamma_{|\mathcal{T}|}(\mathcal{B}(\rho; \boldsymbol{x}^*))}\right] \tag{259}$$

$$\overset{(c)}{\leq} \mathbb{E}\left[\sqrt{C_1 |\mathcal{T}| \tilde{\gamma}_{|\mathcal{T}|}(\mathcal{B}(\rho; 0))}\right] \tag{260}$$

$$\overset{(d)}{\leq} \sqrt{C_1 \mathbb{E}[|\mathcal{T}|] \mathbb{E}\left[\tilde{\gamma}_{|\mathcal{T}|}(\mathcal{B}(\rho; 0))\right]} \tag{261}$$

$$\overset{(e)}{\leq} \sqrt{C_1 \mathbb{E}[|\mathcal{T}|] \tilde{\gamma}_{\mathbb{E}[|\mathcal{T}|]}(\mathcal{B}(\rho; 0))}. \tag{262}$$

The inequalities hold because of (a) Cauchy–Schwarz inequality (b) $\sum_{t \in \mathcal{T}} \sigma_{t-1}^2(\boldsymbol{x}_t) \leq C_1 \gamma_{|\mathcal{T}|}(\mathcal{B}(\rho; \boldsymbol{x}^*))$ from Lemma E.3 and $\boldsymbol{x}_t \in \mathcal{B}(\rho; \boldsymbol{x}^*)$, (c) $\gamma_T(\mathcal{B}(\rho; \boldsymbol{x}^*)) = \tilde{\gamma}_T(\mathcal{B}(\rho; 0))$ for any $T \in \mathbb{N}$ due to the definition of $\tilde{\gamma}$ and the shift-invariant property of the SE and Matérn kernels, (d) Cauchy–Schwarz inequality ($|\mathbb{E}[XY]| \leq \sqrt{\mathbb{E}[X^2]\mathbb{E}[Y^2]}$ for any random variables $X$ and $Y$), and (e) Jensen's inequality and the concavity of $\tilde{\gamma}_t$.

For the proof with respect to $\sigma_{t-1}^2(\boldsymbol{x}^*)$, we use the property of $\mathcal{T}_g = \left\{t \in [T] \mid g_t(\boldsymbol{x}_t) - g_t(\boldsymbol{x}^*) \geq \Delta \wedge \|\boldsymbol{x}_t - \boldsymbol{x}^*\|_2 \leq \rho\right\}$.

Let $\mathbb{1}_E$ be the indicator function that is 1 if $E$ is true, and 0 otherwise. We can see that

$$
\begin{aligned}
\mathbb{E}\left[|\mathcal{T}|\right] &= \mathbb{E}\left[\sum_{t=1}^{T}\mathbb{E}\left[\mathbb{1}_{f(\boldsymbol{x}^*)-f(\boldsymbol{x}_t)\geq\Delta\wedge\|\boldsymbol{x}_t-\boldsymbol{x}^*\|_2\leq\rho}\mid\mathcal{D}_{t-1}\right]\right] \\
&= \mathbb{E}\left[\sum_{t=1}^{T}\mathbb{E}\left[\mathbb{1}_{g_t(\boldsymbol{x}_t)-g_t(\boldsymbol{x}^*)\geq\Delta\wedge\|\boldsymbol{x}_t-\boldsymbol{x}^*\|_2\leq\rho}\mid\mathcal{D}_{t-1}\right]\right] \\
&= \mathbb{E}\left[|\mathcal{T}_g|\right],
\end{aligned}
$$

where the second equality holds since $\left(\mathbb{1}_{f(\boldsymbol{x}^*)-f(\boldsymbol{x}_t)\geq\Delta\wedge\|\boldsymbol{x}_t-\boldsymbol{x}^*\|_2\leq\rho}\mid\mathcal{D}_{t-1}\right)$ and $\left(\mathbb{1}_{g_t(\boldsymbol{x}_t)-g_t(\boldsymbol{x}^*)\geq\Delta\wedge\|\boldsymbol{x}_t-\boldsymbol{x}^*\|_2\leq\rho}\mid\mathcal{D}_{t-1}\right)$ are identically distributed.

Then, we have

$$
\mathbb{E}\left[\sum_{t\in\mathcal{T}}\sigma_{t-1}^2(\boldsymbol{x}^*)\right] = \mathbb{E}\left[\sum_{t\in[T]}\mathbb{E}\left[\sigma_{t-1}^2(\boldsymbol{x}^*)\mathbb{1}_{f(\boldsymbol{x}^*)-f(\boldsymbol{x}_t)\geq\Delta\wedge\boldsymbol{x}_t\in\mathcal{B}(\rho;\boldsymbol{x}^*)}\;\middle|\;\mathcal{D}_{t-1}\right]\right] \tag{263}
$$

$$
\stackrel{(a)}{=} \mathbb{E}\left[\sum_{t\in[T]}\mathbb{E}\left[\sigma_{t-1}^2(\boldsymbol{x}_t)\mathbb{1}_{g_t(\boldsymbol{x}_t)-g_t(\boldsymbol{x}^*)\geq\Delta\wedge\|\boldsymbol{x}_t-\boldsymbol{x}^*\|_2\leq\rho}\;\middle|\;\mathcal{D}_{t-1}\right]\right] \tag{264}
$$

$$
= \mathbb{E}\left[\sum_{t\in\mathcal{T}_g}\sigma_{t-1}^2(\boldsymbol{x}_t)\right] \tag{265}
$$

$$
\stackrel{(b)}{\leq} C_1\mathbb{E}\left[\gamma_{|\mathcal{T}_g|}\left(\mathcal{B}\left(\rho;\boldsymbol{x}^*\right)\right)\right] \tag{266}
$$

$$
\stackrel{(c)}{\leq} C_1\mathbb{E}\left[\tilde{\gamma}_{|\mathcal{T}_g|}\left(\mathcal{B}\left(\rho;0\right)\right)\right] \tag{267}
$$

$$
\stackrel{(d)}{\leq} C_1\tilde{\gamma}_{\mathbb{E}[|\mathcal{T}_g|]}\left(\mathcal{B}\left(\rho;0\right)\right) \tag{268}
$$

$$
= C_1\tilde{\gamma}_{\mathbb{E}[|\mathcal{T}|]}\left(\mathcal{B}\left(\rho;0\right)\right) \tag{269}
$$

where the equality and inequality hold because (a) $\left(\sigma_{t-1}^2(\boldsymbol{x}^*)\mathbb{1}_{f(\boldsymbol{x}^*)-f(\boldsymbol{x}_t)\geq\Delta\wedge\boldsymbol{x}_t\in\mathcal{B}(\rho;\boldsymbol{x}^*)}\mid\mathcal{D}_{t-1}\right)$ and $\left(\sigma_{t-1}^2(\boldsymbol{x}_t)\mathbb{1}_{g_t(\boldsymbol{x}_t)-g_t(\boldsymbol{x}^*)\geq\Delta\wedge\|\boldsymbol{x}_t-\boldsymbol{x}^*\|_2\leq\rho}\mid\mathcal{D}_{t-1}\right)$ are identically distributed, (b) $\sum_{t\in\mathcal{T}}\sigma_{t-1}^2(\boldsymbol{x}_t)\leq C_1\gamma_{|\mathcal{T}|}\left(\mathcal{B}\left(\rho;\boldsymbol{x}^*\right)\right)$ from Lemma E.3 and $\boldsymbol{x}_t\in\mathcal{B}\left(\rho;\boldsymbol{x}^*\right)$, (c) $\gamma_T\left(\mathcal{B}\left(\rho;\boldsymbol{x}^*\right)\right)=\tilde{\gamma}_T\left(\mathcal{B}\left(\rho;0\right)\right)$ for any $T\in\mathbb{N}$ due to the definition of $\tilde{\gamma}$ and the shift-invariant property of the SE and Matérn kernels, and (d) Jensen's inequality and the concavity of $\tilde{\gamma}_T$. By almost the same arguments shown above and Lemma E.7, we further obtain

$$
\mathbb{E}\left[\sum_{t\in\mathcal{T}}\sigma_{t-1}(\boldsymbol{x}^*)\right] \leq \sqrt{C_1\mathbb{E}[|\mathcal{T}|]\tilde{\gamma}_{\mathbb{E}[|\mathcal{T}|]}\left(\mathcal{B}\left(\rho;0\right)\right)},
$$

which concludes the proof. $\qquad\square$

