# OpenReview forum: "On Regret Bounds of Thompson Sampling for Bayesian Optimization"
_ICML.cc/2026/Conference — ICML 2026 regular_

### Official Review · Reviewer_sL9y · 2026-02-25

**Soundness:** 4
**Presentation:** 4
**Significance:** 3
**Originality:** 4
**Overall Recommendation:** 5
**Confidence:** 3

**Summary:**

The paper analyzes the regret of Gaussian process Thompson Sampling (GP-TS) in a Bayesian setting and provides 4 main results. First, the paper provides an instance where GP-TS obtains $O(1/\delta^c)$ regret with probability $\delta$ which implies that general $O(\log(1/\delta))$ regret upper bounds cannot be obtained. Second, the paper provides an upper bound of $\mathbb{E}[R^2_T]$ that leads to a $O(1/\sqrt{\delta})$ high probability upper bound of the regret, improving upon previous $O(1/\delta)$ results. Third, the paper provides the first bound of the expected lenient regret for GP-TS, matching existing high-probability results for GP-UCB. Fourth, the paper provides a high-probability regret bound for GP-TS using different techniques compared to previous work. Notably, this final result has looser requirements on the smoothness parameter $\nu$ for Matérn kernels.

**Compliance With Llm Reviewing Policy:**

Affirmed.

**Final Justification:**

The original review did not raise any major weaknesses nor did the reviews of the other reviewers. A minor error was discovered later (see rebuttal acknowledgement) but the authors agreed with the proposed fix. As such, I maintain the recommendation of acceptance.

**Key Questions For Authors:**

- **Q1:** Connected to W1, can the techniques and results of Theorem 3.1 be used to derive a regret lower bound that holds for all algorithms?
- **Q2:** The role and impact of $\bar i$ in Lemma 3.4 and Theorem 3.5 is not clear to me. Could you elaborate on how $\bar i$ is used and its importance?
- **Q3:** L29-30 in the abstract, L68-69, L190, and the title of subsection 3.4 reference that Theorem 3.5 is "improved" or "tighter" with respect to $T$ . The discussion below Theorem 3.5 compares against the result of Iwazaki (2025b) with respect to $\delta$ , not $T$ . Could you elaborate on how Theorem 3.5 improves or tightens regret bounds for GP-TS with respect to $T$ ?
- **Q4:** Lenient regret is perhaps less studied than standard regret and expected regret. However, are there any lower bounds for lenient regret that one could use to understand the tightness of the bounds in Theorem 3.3?

**Limitations:**

Yes.

**Strengths And Weaknesses:**

- **S1:** The paper improves upon existing results for GP-TS in multiple aspects.
- **S2:** The paper is overall well-written and the theoretical work appears to be sound (although the proofs have not been verified in detail). Each result is compared against relevant work allowing the reader to put it into context. Additionally, high-level explanations of the proofs are also provided.
- **W1:** In the introduction, it is noted that Scarlett (2018) showed a $\Omega(\sqrt{T})$ regret lower bound for conditional expected regret in one dimension and that rigorous high-probability regret lower bounds have not been established. The regret lower bound of Theorem 3.1 does not hold for any algorithm but only for GP-TS, unlike the result of Scarlett (2018).

---

> ### Author Rebuttal · Authors · 2026-03-30
>
> We appreciate the detailed and constructive feedback.
> We will revise the paper to improve clarity depending on the comments.
>
> >Q1: Connected to W1, can the techniques and results of Theorem 3.1 be used to derive a regret lower bound that holds for all algorithms?
>
> Extending this result to all algorithms appears difficult.
> However, Theorem 3.1 may generalize to algorithms that do not have tuning parameters depending on $t$ and $\delta$.
> Indeed, our proof is inspired by Theorem 4.6 in Takeno et al. (2025a), which shows GP-UCB with a fixed constant $\beta\_t = c$ incurs linear cumulative regret with strictly positive probability.
>
>
> >Q2: The role and impact of $\bar{i}$ in Lemma 3.4 and Theorem 3.5 is not clear to me. Could you elaborate on how $\bar{i}$ is used and its importance?
>
> In the proof, we tune $\Delta\_i$ so that the cumulative regret over the index set $\mathcal{T}\_i$ can be bounded from above by $\tilde{O}(\sqrt{T})$.
> This imposes a constraint on how fast $\Delta\_i$ can decrease: if $\Delta\_{i+1}$ is too small, the regret over $\mathcal{T}\_i$ may exceed $\tilde{O}(\sqrt{T})$.
> On the other hand, to obtain $\tilde{O}(\sqrt{T})$ regret upper bound over $\mathcal{T}\_{\bar{i}}$, we require $\Delta\_{\bar{i}}$ to be sufficiently small.
> Thus, our proof needs the existence of $\bar{i} = \tilde{O}(1)$ such that $\Delta\_{\bar{i}}$ is sufficiently small while respecting the decay constraint of $\Delta\_i$.
> We have shown that such $\bar{i}$ exists for SE kernels or Mat\'ern kernels with $\nu > 2$.
> As a result, Lemma 3.4 gives $R\_T = \tilde{O}(\bar{i} \sqrt{T})$, since we sum $\tilde{O}(\sqrt{T})$ cumulative regrets for all index sets.
> Furthermore, Theorem 3.5 gives $R\_T = \tilde{O}(\bar{i}^2 \sqrt{T})$ since we further consider the union bound for all $i \in [\bar{i}]$.
>
>
> >Q3: L29-30 in the abstract, L68-69, L190, and the title of subsection 3.4 reference that Theorem 3.5 is "improved" or "tighter" with respect to $T$. The discussion below Theorem 3.5 compares against the result of Iwazaki (2025b) with respect to $\delta$, not $T$. Could you elaborate on how Theorem 3.5 improves or tightens regret bounds for GP-TS with respect to $T$?
>
> The known-best high-probability cumulative regret upper bound for GP-TS are $O(\sqrt{T \gamma\_T \log T} / \delta)$ (Russo and Van Roy, 2014; Takeno et al., 2024) or $O(\sqrt{T \gamma\_T \log T/ \delta})$ (Theorem 3.2).
> Thus, Theorem 3.5 improves the dependence on $T$, not on $\delta$.
> The discussion below Theorem 3.5 compares GP-TS and GP-UCB (Iwazaki, 2025b) in terms of $\delta$-dependence.
>
>
> >Q4: Lenient regret is perhaps less studied than standard regret and expected regret. However, are there any lower bounds for lenient regret that one could use to understand the tightness of the bounds in Theorem 3.3?
>
> To our knowledge, lower bounds for lenient regret are only known in the frequentist setting (Cai et al., 2021).
> Establishing such bounds in the Bayesian setting remains open.

---

> > ### Author Rebuttal · Reviewer_sL9y · 2026-04-01
> >
> > The original review review only raised relatively minor concerns. Reading the other reviews and responses did not raise any major additional concerns. As such, I maintain the recommendation of acceptance.
> >
> > An additional minor comment, on line 187 column 2, it is stated that $\gamma_T$ is submodular and therefore has decreasing increments in $T$. First, the mutual information $I(f; y_A)$ is submodular w.r.t. $A$ but $\gamma_T$ cannot be submodular since it is not a set function. Second, submodularity of $I(f; y_A)$ does not immediately imply that the maximum information gain has diminishing returns, i.e. that $\gamma_{t} - \gamma_{t-1} \leq \gamma_{t+1} - \gamma_t$, since the maximally informative set of size $t$ is not necessarily a subset of the maximally informative set of size $t+1$. Consequently, the piecewise linear interpolation is not necessarily concave. Although, I believe this can be resolved simply by considering a concave function $\tilde{\gamma}_T$ that minimally *upper bounds* $\gamma_T$ as compared to linearly interpolating it. The known upper bounds for common kernels should satisfy these requirements.

---

> > > ### Author Response · Authors · 2026-04-03
> > >
> > > We sincerely appreciate the prompt feedback and insightful comments.
> > >
> > > >An additional minor comment
> > >
> > > We fully agree with the reviewer's comments on both points, and we apologize for our carelessness.
> > >
> > >
> > > >Although, I believe this can be resolved simply by considering a concave function $\tilde{\gamma}\_T$ that minimally upper bounds $\gamma\_T$ as compared to linearly interpolating it. The known upper bounds for common kernels should satisfy these requirements.
> > >
> > > We agree on this point too.
> > > We will revise the paper regarding this point.

---

### Official Review · Reviewer_QExa · 2026-02-28

**Soundness:** 3
**Presentation:** 2
**Significance:** 2
**Originality:** 2
**Overall Recommendation:** 4
**Confidence:** 2

**Summary:**

This paper studies theoretical regret analysis for Gaussian Process Thompson Sampling (GP-TS) in Bayesian Optimization (BO), which presents the characterization of cumulative regret and lenient regret bounds for GP-TS under the Bayesian setting where the objective function is sampled from a Gaussian process (GP).

The paper aims to close the theoretical gap between GP-TS and GP-UCB. While GP-UCB enjoys well-established high-probability cumulative regret and lenient regret bounds, GP-TS has historically been analyzed mostly in expectation, with weaker high-probability guarantees and suboptimal dependence on the failure probability parameter δ.

**Compliance With Llm Reviewing Policy:**

Affirmed.

**Final Justification:**

My main takeaway is that the paper’s contribution primarily lies in establishing a lower bound for $\delta$ and improving the corresponding upper bound. While such upper- and lower-bound results are certainly meaningful, the bandit literature already contains a large number of results of this type under different settings. As a result, I have been uncertain whether the level of contribution here is strong enough for this conference. That said, after further consideration, I view this paper more positively than work that merely introduces a new setting without obtaining any substantive improvement. For this reason, I have increased my score, though I lowered my confidence to reflect my remaining uncertainty about the overall significance of the contribution.

**Key Questions For Authors:**

1. What is the conjectured optimal dependence on $\delta$ for GP-TS? Can the authors clarify whether their $1/\sqrt{\delta}$ improvement (via second-moment bounds) is believed to be minimax optimal?
2. Are all three structural properties (unique maximizer, sup-norm bound, quadratic local behavior) essential for the $\tilde{O}(\sqrt{T})$ result? Could weaker assumptions suffice?

**Limitations:**

Yes

**Strengths And Weaknesses:**

### Strengths

1. The paper precisely articulates what is missing in the GP-TS literature (e.g., lack of high-probability bounds, absence of lenient regret analysis) and systematically addresses these gaps.
2. The two-armed construction in Theorem 3.1 is conceptually simple yet powerful. It convincingly shows that logarithmic dependence on $1/\delta$ is impossible in general for GP-TS.
3. The paper provides a comprehensive discussion of both Bayesian and frequentist BO analyses and clearly situates its contributions relative to GP-UCB and recent refinements.

### Weaknesses

1. While primarily theoretical, even minimal experiments illustrating concentration behavior of GP-TS regret could strengthen practical relevance.
2. While improving δ-dependence is technically interesting, it is unclear whether this refinement fundamentally changes our understanding of GP-TS.

---

> ### Author Rebuttal · Authors · 2026-03-30
>
> We appreciate the detailed and constructive feedback.
>
> >(Weakness 2) While improving $\delta$-dependence is technically interesting, it is unclear whether this refinement fundamentally changes our understanding of GP-TS.
>
> Theorem 3.2 shows that the cumulative regret is more tightly concentrated around zero than previously suggested by existing results.
> Although minimax optimality is not established, improving the dependence on $\delta$ is important for demonstrating the practical stability of GP-TS.
> For example, for a practical choice such as $\delta = 0.01$, the gap between $1/\sqrt{\delta} = 10$ and $\log(1/\delta) \approx 4.6$ is relatively modest, whereas the scale of $1/\delta = 100$ remains substantial and cannot be neglected.
> In this sense, we believe that the improvement to a $1/\sqrt{\delta}$ dependence is a meaningful step forward.
>
>
> >1. What is the conjectured optimal dependence on $\delta$ for GP-TS? Can the authors clarify whether their $1 / \sqrt{\delta}$ improvement (via second-moment bounds) is believed to be minimax optimal?
>
> We do not consider our result to be minimax optimal, as discussed in Section 4.
> Establishing tighter lower or upper bounds remains an important direction for future work.
>
>
> >2. Are all three structural properties (unique maximizer, sup-norm bound, quadratic local behavior) essential for the $\tilde{O}(\sqrt{T})$ result? Could weaker assumptions suffice?
>
> Our proof relies on all the properties.
> However, we do not show that they are necessary.
> We believe that the sup-norm bound is already sufficiently mild since it can be shown by, e.g., Dudley's inequality under mild kernel regularity assumptions.
> We conjecture that the unique maximizer condition can be relaxed to the condition that the number of maximizers is bounded from above by some variable with $\tilde{O}(1)$.
> On the other hand, the quadratic local behavior condition is essential to our proof and appears to be the main bottleneck.
> Deriving $\tilde{O}(\sqrt{T})$ cumulative regret upper bound under weaker assumptions is an important future direction.

---

> > ### Author Rebuttal · Reviewer_QExa · 2026-04-03
> >
> > I have no further major concerns. However, I am still somewhat uncertain whether the significance of this work is sufficient for presentation at this conference. I will use the discussion phase to further assess its significance and then consider whether to raise my score.

---

> > > ### Author Response · Authors · 2026-04-07
> > >
> > > We sincerely appreciate the prompt feedback.
> > > We would like to clarify our contributions again.
> > > We believe that our work addresses known theoretical gaps and supports the theoretical effectiveness of the widely used and analyzed GP-TS algorithm.
> > >
> > > Our contributions are not limited to Theorems 3.1 and 3.2.
> > > While these theorems characterize the $\delta$-dependence in the cumulative regret bounds of GP-TS, we also provide Theorems 3.3 and 3.5, Lemma 3.4, and related lemmas.
> > > In particular, Theorems 3.3 and 3.5 derive lenient regret bounds, and the high-probability cumulative regret bound improved with respect to the dependence on $T$, respectively.
> > >
> > > Furthermore, our proof techniques are generalizable to other algorithms.
> > > We expect that our theoretical results can be extended to other BO algorithms for which only Bayesian expected regret bounds have been established, as discussed in the right column of page 2.
> > > Furthermore, the proof technique of Theorem 3.3 can be applied to the expected lenient regret analysis of GP-UCB, for which only the high-probability analysis was available (see the last paragraph of Section 1).
> > > Finally, Lemma 3.5 can directly provide an improved upper bound on the cumulative regret of GP-UCB with respect to $T$ under a weaker assumption on $\nu$ for Mat\'ern kernels, thereby allowing for the widely used parameter value $\nu = 5/2$.

---

### Official Review · Reviewer_Rcrd · 2026-03-12

**Soundness:** 3
**Presentation:** 3
**Significance:** 3
**Originality:** 3
**Overall Recommendation:** 4
**Confidence:** 3

**Summary:**

This paper investigates the cumulative regret of Gaussian Process Thompson Sampling (GP-TS). The authors first provide a lower bound (Theorem 3.1) to motivate the need for a tighter analysis of the $\delta$-dependence in the regret bound. They then propose an improved upper bound by replacing the conventional Markov’s inequality with a second-order moment analysis, achieving a $1/\sqrt{\delta}$ dependence. Finally, they extend these results to the more practical Matérn kernel setting (Theorem 3.5). While the objective is clear, there are significant concerns regarding the validity of the problem setting used for the lower bound proof.

**Compliance With Llm Reviewing Policy:**

Affirmed.

**Final Justification:**

Initially, I had significant concerns regarding the motivation of the paper, specifically the seemingly contrived problem setting in Theorem 3.1. However, through the rebuttal and subsequent discussion phase, the authors successfully clarified that this specific lower-bound construction is essential for identifying a fundamental theoretical limitation of GP-TS—namely, its inability to incorporate the failure probability $\delta$ into its sampling mechanism—which corrects technical flaws in existing literature (e.g., Bayrooti et al., 2025).

The authors have committed to revising the manuscript to explicitly discuss the astronomical rarity of the event $O(T^{-8})$ and to clearly distinguish this theoretical "gap-filling" from practical empirical stability. While the technical rigor in refining the $\delta$-dependence and the extension to Matérn kernels ($\nu > 2$) are commendable, a degree of uncertainty remains regarding the overall significance and impact of addressing such rare failure scenarios within the broader Bayesian Optimization community.

Reflecting these considerations, I have raised my scores for Soundness and Significance to acknowledge the technical contributions, while maintaining a balanced Overall Recommendation that reflects both the solid theoretical correction and the specialized nature of the findings.

**Key Questions For Authors:**

[1] Justification of the condition in Theorem 3.1
 In your proof for the lower bound, you assume $f(x^{(1)}) \ge 4\sqrt{\log T}$. In a standard Bayesian Optimization setting, the true function $f$ is a realization from a GP and should be independent of the number of steps $T$. Could you explain the rationale for making the function's properties dependent on $T$? If $f$ is fixed before $T$ is chosen, does the $1/\delta^c$ regret still hold?

[2] Comparison with GP-UCB
It is widely accepted that GP-UCB achieves $O(\log(1/\delta))$ dependence. Under the exact same problem instance and conditions used in Theorem 3.1, does GP-UCB also suffer from the $1/\delta^c$ regret? Please clarify whether the difficulty arises specifically from the Thompson Sampling mechanism or the problem instance itself.

**Limitations:**

yes

**Strengths And Weaknesses:**

Soundness
The soundness of the motivation is questionable due to the construction of Theorem 3.1. The condition $f(x^{(1)}) \ge 4\sqrt{\log T}$ used in the proof implies that the difficulty of the problem instance (the "target" function) scales with the number of observation steps $T$. This makes it unclear whether the failure to achieve $\log(1/\delta)$ is a fundamental flaw of GP-TS or simply a result of an unnaturally difficult problem setting.

Persentation
The paper is generally well-structured. However, the connection between the "pessimistic" lower bound in Section 3.1 and the "improved" upper bound in Section 3.2 is not fully convincing. The paper would benefit from a more rigorous justification of why the $1/\delta$ dependence is unavoidable in a standard, fixed-objective setting.

Significance
The extension of the regret analysis to Matérn kernels ($\nu > 2$) is highly significant. Since many real-world phenomena do not satisfy the infinite smoothness of the SE kernel, providing theoretical guarantees for Matérn kernels increases the practical utility of GP-TS theory.

Originality
The authors move beyond the standard 0-th order (Markov) inequality by employing a second-order moment analysis. This technical transition is a creative way to refine the $\delta$-dependence in the regret bound for GP-TS.

---

> ### Author Rebuttal · Authors · 2026-03-30
>
> We appreciate the detailed and constructive feedback.
>
> >[1] Justification of the condition in Theorem 3.1 In your proof for the lower bound, you assume $f(x^{(1)}) \geq 4 \sqrt{\log T}$. In a standard Bayesian Optimization setting, the true function $f$ is a realization from a GP and should be independent of the number of steps. Could you explain the rationale for making the function's properties dependent on $T$? If $f$ is fixed before $T$ is chosen, does the $1 / \delta^c$ regret still hold?
>
> We apologize for any confusion on this point.
> We assume that $f$ follows a GP and do not impose that $f(x^{(1)}) \geq 4 \sqrt{\log T}$.
> Instead, we consider the (probabilistic) event $f(x^{(1)}) \geq 4 \sqrt{\log T}$, and explicitly account for its probability in our lower bound.
> That is, we also showed that, under our assumption that $f$ follows a GP, the event $f(x^{(1)}) \geq 4 \sqrt{\log T}$ occurs with probability at least $1 / T^c$ with some constant $c$.
> Thus, we believe that our proof rigorously shows that GP-TS cannot avoid $1 / \delta^c$ dependence with some constant $c$ under the common GP assumption.
> We will revise the manuscript to clarify this point.
>
>
> >[2] Comparison with GP-UCB It is widely accepted that GP-UCB achieves $O(\log (1 / \delta))$ dependence. Under the exact same problem instance and conditions used in Theorem 3.1, does GP-UCB also suffer from the $1 / \delta^c$ regret? Please clarify whether the difficulty arises specifically from the Thompson Sampling mechanism or the problem instance itself.
>
> No.
> By setting $\beta\_t = O(\log(t / \delta))$, GP-UCB can achieve $O(\log (1 / \delta))$ dependence.
> The difficulty arises because GP-TS lacks tuning parameters that depend on $\delta$ and $t$.
> Therefore, inversely, GP-TS with variance inflation (Chowdhury and Gopalan, 2017) using $\beta\_t = O(\log(t / \delta))$ can achieve the $O(\log (1 / \delta))$ dependence for the problem instance in Theorem 3.1.

---

> > ### Author Rebuttal · Reviewer_Rcrd · 2026-04-03
> >
> > I appreciate the authors' detailed responses and the proposed revisions to the manuscript. I now understand that Theorem 3.1 rigorously demonstrates the structural difference between GP-UCB and GP-TS by focusing on a specific event where GP-TS fails due to its inability to incorporate the failure probability $\delta$ into its sampling mechanism.
> >
> > However, from a practical perspective, I remain concerned about the significance of this "rare case" scenario. Based on the condition $f(x^{(1)}) \ge 4\sqrt{\log T}$, the probability of this event occurring under a standard GP prior is approximately $O(T^{-8})$. For instance, even at a small $T=10$, this probability is as low as $10^{-8}$, and it vanishes extremely rapidly as $T$ increases.
> >
> > While I acknowledge that the second half of the paper (e.g., Section 3.3) addresses this by relaxing the regret definition to mitigate these rare events, I believe the following points should be addressed to ensure the paper's contribution is not "oversold":
> >
> > 1. Quantifying the Rarity: Could the authors explicitly discuss or provide an upper bound for the probability of the event $f(x^{(1)}) \ge 4\sqrt{\log T}$ in the manuscript? This would help readers understand that the "failure" of the existing $\log(1/\delta)$ analysis only occurs in astronomically rare instances.
> >
> > 2. Impact on Practical Stability: Does this theoretical "unreliability" in $O(T^{-8})$ cases actually influence the practical stability of GP-TS in real-world benchmarks, or is it purely a theoretical "gap-filling" exercise to correct the technical flaws in prior works (e.g., Bayrooti et al.)?
> >
> > I believe clarifying these points will significantly strengthen the paper’s integrity and help properly position its theoretical contributions.
> >
> > Regarding the scores for Soundness, Presentation, Significance, Originality, and Overall Recommendation, I plan to revise them upward. However, I will determine the specific numerical values based on our ongoing discussion and your final clarifications.

---

> > > ### Author Response · Authors · 2026-04-06
> > >
> > > We sincerely appreciate the prompt feedback and constructive suggestions.
> > >
> > > >1. Quantifying the Rarity: Could the authors explicitly discuss or provide an upper bound for the probability of the event $f(x^{(1)}) \geq 4 \sqrt{\log T}$ in the manuscript? This would help readers understand that the "failure" of the existing $\log (1 / \delta)$ analysis only occurs in astronomically rare instances.
> > >
> > > We will add a discussion following Theorem 3.1 to clarify that the event $f(x^{(1)}) \geq 4 \sqrt{\log T}$ is highly rare.
> > > In addition, we will explicitly state that this result addresses a purely theoretical limitation rather than empirical behavior.
> > >
> > > >2. Impact on Practical Stability: Does this theoretical "unreliability" in $O(T^{-8})$ cases actually influence the practical stability of GP-TS in real-world benchmarks, or is it purely a theoretical "gap-filling" exercise to correct the technical flaws in prior works (e.g., Bayrooti et al.)?
> > >
> > >
> > > Theorem 3.1 does not imply practical instability of GP-TS.
> > > It identifies a purely theoretical limitation that $\log(1/\delta)$ bounds cannot be obtained without additional assumptions.
> > > However, in practice, GP-TS is often stable and effective.
> > > Thus, Theorem 3.1 served as motivation for improvement of the exponent of $1 / \delta$ in the regret upper bound presented in Theorem 3.2.
> > > Furthermore, to support the empirical stability, we derived Theorems 3.3 and 3.5.
> > > We will revise the manuscript to explicitly clarify the distinction between these theoretical limitations and practical performance.

---

### Official Review · Reviewer_NcQT · 2026-03-12

**Soundness:** 4
**Presentation:** 3
**Significance:** 4
**Originality:** 3
**Overall Recommendation:** 5
**Confidence:** 3

**Summary:**

This paper studies regret guarantees for Gaussian process Thompson sampling in Bayesian optimization under a Gaussian process prior with Gaussian noise. The main claims are a lower bound showing that GP-TS cannot generally achieve logarithmic dependence on failure probability, a second moment bound for cumulative regret that improves the high probability dependence to $1 / \sqrt{\delta}$, an expected lenient regret bound, and an improved high probability cumulative regret bound near $\sqrt{T}$ up to polylog terms for squared exponential and Matern kernels. The paper is fully theoretical, and the claims are supported by new theorem statements, proof sketches in the main text, and appendices.

**Compliance With Llm Reviewing Policy:**

Affirmed.

**Final Justification:**

The authors' answers have fully resolved my concerns. I think my score is sufficiently high and I would like to see the acceptance of this paper at ICML 2026.

**Key Questions For Authors:**

- Can you make the dependence of the constants from Lemma 2.4 on $\delta_{GP}$ explicit, or at least characterize whether Theorem 3.5 preserves a meaningful improvement in failure probability after these constants are unpacked?
- Can you clarify more concretely which step in the recent Bayrooti analysis fails in the Bayesian setting, and whether a modified GP-TS rule with variance inflation would recover a sharper dependence on $\delta$?
- Can you add a short discussion of how posterior sampling approximation error would enter the new bounds, even if a full theorem is left to future work?

**Limitations:**

Yes

**Strengths And Weaknesses:**

## Strengths

- The problem is well motivated. The paper targets a real gap between GP-UCB and GP-TS theory, especially for high probability and lenient regret guarantees. Theorem 3.1 and Theorem 3.2 directly address this gap.
- The contribution set is coherent. The lower bound, second moment analysis, lenient regret result, and improved $R_T$ bound fit together well rather than reading like disconnected lemmas.
- Theorem 3.3 seems especially interesting. A polylogarithmic expected lenient regret guarantee for GP-TS appears new and could be useful beyond the final theorem in this paper.
- The refined analysis for Matern kernels looks meaningful. Relaxing the prior condition to $\nu > 2$ strengthens the scope of the improved cumulative regret result.
- The paper is generally careful about limitations. Section 4 is candid about what remains open, especially the gap on $\delta$ and the missing approximation error analysis.

## Weaknesses

- The practical relevance is limited by the modeling and implementation assumptions. The paper assumes the objective is an exact Gaussian process sample path with known kernel and Gaussian noise, and it explicitly ignores posterior sampling approximation error, even though GP-TS usually needs approximate sampling in practice.
- The probability dependence story is still incomplete. Theorem 3.2 improves the bound to $1 / \sqrt{\delta}$, but the lower and upper bounds are still far apart. For Theorem 3.5, the hidden constants depend on quantities from Lemma 2.4 through $\delta_{GP}$, so the final dependence on failure probability is not very explicit.
- The paper would be easier to evaluate with a clearer summary table of prior and new bounds. Several claims about what is new versus what follows from existing analyses are spread across the introduction and theorem discussions.

---

> ### Author Rebuttal · Authors · 2026-03-30
>
> We appreciate the detailed and constructive feedback.
> We will revise the manuscript to clarify the points raised in the comments.
>
> >Can you make the dependence of the constants from Lemma 2.4 on $\delta\_{\rm GP}$ explicit, or at least characterize whether Theorem 3.5 preserves a meaningful improvement in failure probability after these constants are unpacked?
>
> The explicit dependence of the constants in Lemma 2.4 on $\delta\_{\rm GP}$ has not been shown at least by (De Freitas et al., 2012; Scarlett, 2018; Iwazaki, 2025b).
> Theorem 3.5, based on our lenient regret analysis (Theorem 3.3), does not improve the order of $\delta$ but improves the order of $T$ compared with the known-best GP-TS cumulative regret upper bound $O(\sqrt{T \gamma\_T \log T} / \delta)$ (Russo and Van Roy, 2014; Takeno et al., 2024) or $O(\sqrt{T \gamma\_T \log T/ \delta})$ (Theorem 3.2).
>
>
> >Can you clarify more concretely which step in the recent Bayrooti analysis fails in the Bayesian setting?
>
> See the arguments around Eqs. (17-18) of (Bayrooti et al., 2025a).
> In the Bayesian setting, the saturated set $\mathcal{S}\_{h,k}$ is a random variable depending on the target function $f$.
> Therefore, the action $b\_{h,k} = {\rm argmin}\_{a \in \mathcal{A} \backslash \mathcal{S}\_{h,k}} \xi\_{k-1}(s\_{h,k}, a)$ is also random.
> Hence, the realization $\xi\_{k-1}(s\_{h,k}, b\_{h,k})$ is not necessarily smaller than the conditional expectation $\mathbb{E}[\xi\_{k-1}(s\_{h,k}, a\_{h,k}) \mid a\_{h,k} \notin \mathcal{S}\_{h,k}]$, yet this inequality is used in Eq. (18).
> This issue does not occur when $f$ and $\mathcal{S}\_{h,k}$ are fixed constants as in the original analysis of (Chowdhury and Gopalan, 2017).
>
>
> >Whether a modified GP-TS rule with variance inflation would recover a sharper dependence on $\delta$?
>
> We believe that the variance-inflated (modified) GP-TS achieves an $O(\sqrt{T \gamma\_T} \log(T/\delta))$ cumulative regret upper bound in our setting.
> On the other hand, for the problems considered in (Bayrooti et al., 2025a), it remains necessary to verify whether the proof technique of Chowdhury and Gopalan (2017) extends consistently.
>
>
> >Can you add a short discussion of how posterior sampling approximation error would enter the new bounds, even if a full theorem is left to future work?
>
> One naive approach is bounding $f(x^*) - f(x\_t) \leq {\rm max}\_{x \in \mathcal{X}} |f(x) - g\_t(x)| + g\_t(x\_t) - f(x\_t)$, where $g\_t(x)$ is an approximated posterior sample path.
> We conjecture that this naive upper bound (and its square) can be bounded by combining our analysis with the proof by Mutny and Krause (2018).
> However, a more refined argument may exist.

---

> > ### Author Rebuttal · Reviewer_NcQT · 2026-03-31
> >
> > The authors' answers have fully resolved my concerns.  I think my score is sufficiently high and I would like to see the acceptance of this paper at ICML 2026.

---

> > > ### Author Response · Authors · 2026-04-03
> > >
> > > We sincerely appreciate the prompt feedback.

---

### Decision · Program_Chairs · 2026-04-30

**Decision:**

Accept (regular)

**Comment:**

The paper studies regret guarantees for Gaussian process Thompson sampling in Bayesian optimization, with a focus on high-probability and lenient regret bounds under a GP prior. Reviewers generally viewed the paper as a technically strong and well-motivated contribution that addresses a gap in the theory of GP-TS. In particular, the lower bound ruling out logarithmic dependence on the failure probability, the second-moment analysis leading to improved concentration, and the extensions to lenient regret and Matérn kernels were seen as meaningful advances. The main concerns were about the practical significance of the rare-event lower-bound construction, the strength of some modeling assumptions, and aspects of presentation and positioning.

The rebuttal addressed most of these concerns well. In particular, the authors clarified the role of the lower bound as identifying a theoretical limitation rather than practical instability, and committed to revising the paper to better explain the rarity of the bad event and sharpen the exposition. Overall, the technical contributions appear sound, novel, and likely to be useful to researchers working on Bayesian optimization theory. The strengths outweigh the remaining concerns, and I recommend acceptance.